# CAPTURING UNCERTAINTY IN REGRESSION VIA CONDITIONAL DIFFUSION MODELS

## ABSTRACT

Quantifying uncertainty is a fundamental problem in both statistics and machine learning. Existing uncertainty quantification (UQ) approaches often suffer from significant limitations, including high computational cost, restrictive parametric assumptions, and overly conservative prediction intervals. In this paper, we propose a UQ framework based on diffusion models. In regression tasks, we learn the full conditional distribution of the response variable given the input features. Our method enables flexible, nonparametric modeling of complex conditional data distributions. We construct prediction intervals from the learned conditional distribution and establish theoretical guarantees on their coverage probabilities. Empirically, we conduct experiments on both synthetic and real-world regression tasks to evaluate the effectiveness of our approach. The results demonstrate that our method achieves competitive or superior performance in predictive uncertainty estimation compared to a range of established baselines, offering a powerful, efficient and theoretically grounded alternative for uncertainty quantification.

## 1 INTRODUCTION

Uncertainty quantification (UQ) is a foundational objective in statistical estimation and machine learning, concerned with characterizing the reliability of model predictions (Smith, 2013; Sullivan, 2015; Abdar et al., 2021; Nemani et al., 2023; He et al., 2025). Beyond point estimates, effective UQ seeks to capture the range of plausible outcomes given observed data, providing critical insight into model confidence and robustness. This is particularly vital in high-stakes applications, such as medical diagnosis, weather forecasting, and industrial automation, where quantifying predictive uncertainty supports risk-aware decision-making and enhances system safety (Kwon et al., 2020; Begoli et al., 2019; Grenyer et al., 2021; Mehdiyev et al., 2024; Bülte et al., 2025). More broadly, UQ plays a central role across scientific and engineering disciplines, where understanding uncertainty is essential for model validation, decision analysis, and the design of systems (Ghanem et al., 2017).

A wide range of methods have been developed to quantify uncertainty in predictive modeling, each balancing statistical validity, computational efficiency, and modeling flexibility (MacKay, 1992; Koenker, 2005; Yu & Moyeed, 2001; Smith, 2013; Sullivan, 2015; Nemani et al., 2023; Gawlikowski et al., 2023; He et al., 2025). Classical statistical approaches, such as quantile estimation and parametric modeling, offer interpretability and theoretical guarantees under strong assumptions, but often lack flexibility in complex or high-dimensional settings (Waldmann, 2018; Berk et al., 2021). Bootstrapping methods are broadly applicable and easy to implement, yet can be computationally intensive and sensitive to data variability (Kleiner et al., 2014). Bayesian approaches provide a coherent framework for uncertainty quantification by modeling posterior distributions over parameters or predictions, but they typically rely on priors and incur substantial computational costs in practice (Neal, 2012; Gal & Ghahramani, 2016c; Gal et al., 2022; Gawlikowski et al., 2023). More recently, conformal prediction has gained attention for its ability to produce distribution-free prediction sets with finite-sample validity guarantees (Shafer & Vovk, 2008; Angelopoulos & Bates, 2023). However, it tends to be conservative and insufficiently responsive to input-specific uncertainty (Romano et al., 2019; Ho et al., 2020b; Kong et al., 2024). Despite the diversity of existing techniques, achieving a practical balance between coverage accuracy, adaptivity, and scalability remains a fundamental challenge in modern UQ.

Recent advances in machine learning have led to the development of more flexible and data-driven approaches to UQ (MacKay, 1992; Lakshminarayanan et al., 2017b; Romano et al., 2019; Wenzel et al., 2020; Olivier et al., 2021; Karimi & Samavi, 2023). In particular, deep ensembles (Lakshminarayanan et al., 2017b; Laurent et al., 2023; Wenzel et al., 2020; Hu et al., 2019; Salem et al., 2020), Bayesian neural network based methods (Neal, 2012; Wang & Yeung, 2021; Gal & Ghahramani, 2016c; MacKay, 1992), and quantile regression with neural networks (Cannon, 2011; Zhang et al., 2018; Romano et al., 2019) have shown promise in capturing complex predictive uncertainty without relying on restrictive parametric assumptions. More recently, generative models such as variational autoencoders (VAEs) and generative adversarial networks (GANs) have been leveraged to approximate predictive distributions and generate uncertainty estimates in high-dimensional settings (Sohn et al., 2015; Goodfellow et al., 2020; Edupuganti et al., 2021; Gao & Ng, 2022; Mo et al., 2022; Kingma & Welling, 2022). These models offer appealing expressiveness and adaptivity to data, helping to overcome the limitations of traditional statistical techniques (Khursheed et al., 2020; Chakraborty et al., 2024; Garay-Maestre et al., 2019). However, their empirical performance often suffers from issues such as mode collapse and training instability. Moreover, limited theoretical guarantees are established for these methods, which hinder their reliability in applications.

Diffusion models have recently emerged as a powerful class of generative models capable of capturing intricate and high-dimensional data distributions (Lu et al., 2024; Finzi et al., 2023; Song & Ermon, 2019; Ho et al., 2020b; Song et al., 2020b; Ho et al., 2020c; Song et al., 2020a). These models achieve state-of-the-art performance in image synthesis and other generative tasks, with stable training and strong sample fidelity (Kong et al., 2020; Mittal et al., 2021; Jeong et al., 2021; Huang et al., 2022; Avrahami et al., 2022; Ulhaq & Akhtar, 2022). More importantly, conditional diffusion models allow the sample generation process to be guided by auxiliary information such as inputs, contexts, or covariates, dramatically enhancing their versatility and applicability to prediction tasks. Motivated by these strengths, we propose a new framework for UQ based on conditional diffusion models; see a demonstration of performance in

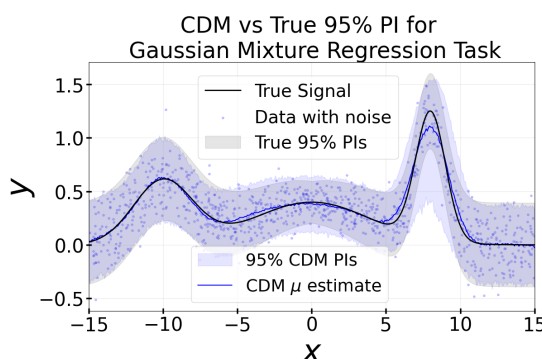

Figure 1: Diffusion-based UQ on a synthetic regression task with data-dependent noise. Blue dots are noisy observations, the black curve is the true function, and the light-gray band denotes the true 95% prediction interval under $P(Y|X = x)$. The light-blue band is the estimated 95% interval, and the solid blue line is the conditional diffusion model (CDM)'s point estimate of the mean.

Figure 1. Our approach leverages their expressiveness and sampling-based nature to approximate predictive distributions and construct confidence intervals in a principled and data-driven manner. For clarity and self-containment, we include a review of diffusion model theory in Appendix A.

We conduct comprehensive investigations of our diffusion-based UQ method. Theoretically, we focus on nonparametric regression problems with data-dependent noise. We address the following central theoretical question:

*Can conditional diffusion models effectively learn predictive distributions and construct uncertainty intervals with desired coverage?*

Empirically, we compare our method against a range of baseline UQ approaches, including conformal prediction, deep ensembles, Bayesian methods, and quantile regression. Across diverse datasets, our method consistently produces accurate and adaptive prediction intervals, achieving competitive or superior performance in terms of interval coverage and computational efficiency.

In summary, our main contributions are three-fold: 1) we present a diffusion-based UQ framework for nonparametric regression with data-dependent noise in Algorithm 1; 2) we establish theoretical guarantees of our algorithm in terms of the convergence of coverage probability to the nominal level in Theorem 4.5; 3) we demonstrate strong empirical performance of our method.

We are aware that Han et al. (2022) are one of the earliest efforts to apply diffusion methods for UQ in classification and regression, adopting a DDPM parameterization and emphasizing algorithmic

design and predictive performance. By comparison, our work prioritizes theoretical analysis and proves rigorous guarantees for our diffusion-based UQ algorithm.

**Notation**  For a vector $x \in \mathbb{R}^d$, we denote its $\ell_2$-norm by $\|x\|$, its $\ell_\infty$-norm by $\|x\|_\infty = \max_{1 \le i \le d} |x_i|$, and its $\ell_1$-norm by $\|x\|_1 = \sum_{i=1}^d |x_i|$. When describing the forward process of diffusion models, we use $\phi_t$ to denote the Gaussian transition kernel at time $t$.

## 2    EXTENDED RELATED WORK ON UNCERTAINTY QUANTIFICATION

In recent decades, a wide array of techniques has been developed for uncertainty quantification in regression problems. These include Bayesian approaches (MacKay, 1992; Neal, 2012), ensemble techniques (Hu et al., 2019), direct interval estimation methods (Koenker & Hallock, 2001), and conformal prediction methods (Romano et al., 2019). We refer readers to surveys for a more comprehensive discussion (Abdar et al., 2021; Nemani et al., 2023; He et al., 2025).

Among Bayesian methods, Bayesian Deep Learning (BDL) (Wang & Yeung, 2021) and Bayesian Neural Networks (BNNs) (MacKay, 1992; Neal, 2012; Wang et al., 2018) are two prominent examples. These models typically employ Markov Chain Monte Carlo (MCMC) (Salakhutdinov & Mnih, 2008; Neal, 2012), variational inference (VI) (Graves, 2011; Hinton & van Camp, 1993), or Monte Carlo (MC) dropout to approximate Bayesian inference (Gal & Ghahramani, 2016c; Brach et al., 2020). They are generally robust against overfitting and adaptable to both small and large datasets (Abdar et al., 2021). Nevertheless, MCMC and VI methods often entail significant computational costs and exhibit limited scalability (Neal, 2012). To alleviate this, MC dropout offers a more scalable alternative. However, its effectiveness is sensitive to several hyperparameters, such as the dropout probability and the number of dropout layers, which can substantially influence the quality of uncertainty estimates (Osband, 2016; Alarab et al., 2021; Caldeira & Nord, 2020).

Ensemble techniques are a widely adopted approach for improving predictive performance and capturing model uncertainty in supervised learning tasks (Lakshminarayanan et al., 2017b; Laurent et al., 2023; Hu et al., 2019; Wenzel et al., 2020). Ensemble models aggregate predictions of multiple neural networks to form an output distribution, with the variability among predictions serving as an indicator of model uncertainty. This approach also tends to produce higher uncertainty estimates for out-of-distribution examples (Lakshminarayanan et al., 2017b). This makes them particularly useful in uncertainty prediction problems where test data distributions are often unknown (Jain et al., 2020).

Direct interval estimation methods, such as quantile regression, are powerful tools for constructing prediction intervals in regression tasks (Koenker & Hallock, 2001; Romano et al., 2019). Unlike traditional regression techniques, quantile regression focuses on estimating conditional quantiles, providing a more comprehensive view of the potential outcomes (Koenker & Hallock, 2001).

Conformal Prediction is a distribution-free framework for uncertainty quantification that constructs valid prediction intervals based on a chosen nonconformity measure. Gammerman et al. (1998); Saunders et al. (1999); Vovk et al. (1999); Angelopoulos & Bates (2023). Unlike methods that rely on strong probabilistic assumptions, conformal prediction provides finite-sample coverage guarantees under minimal assumptions. However, the original form of conformal prediction involved considerable computational cost due to the need to retrain models for each prediction. To address this limitation, split-conformal prediction was proposed (Papadopoulos et al., 2002; 2007). It separates the model training phase from the conformalization process, significantly improving computational efficiency while preserving the theoretical validity of the intervals.

Distinguishing which sources of uncertainty each class of methods primarily addresses is also crucial in UQ. Following Nemani et al. (2023), two main types of uncertainty are typically considered in UQ: *epistemic* uncertainty and *aleatoric* uncertainty. Epistemic uncertainty stems from limited data and model misspecification, whereas aleatoric uncertainty arises from inherent randomness in the observations and cannot be reduced even with more data. For Bayesian deep models such as Bayesian Neural Networks (BNNs) (MacKay, 1992; Neal, 2012; Wang et al., 2018), variability across posterior samples of the network (obtained via MCMC, VI, or MC dropout) is mainly a proxy for *epistemic* uncertainty (Salakhutdinov & Mnih, 2008; Neal, 2012; Graves, 2011; Hinton & van Camp, 1993; Gal & Ghahramani, 2016c; Brach et al., 2020). Ensemble techniques similarly cap-

ture epistemic uncertainty through training an ensemble (Lakshminarayanan et al., 2017b; Laurent et al., 2023; Hu et al., 2019; Wenzel et al., 2020). By contrast, direct interval estimation methods and conformal prediction methods focus on the conditional distribution of the response given the covariates and thus primarily target aleatoric (or overall predictive) uncertainty, without explicitly disentangling epistemic and aleatoric components (Koenker & Hallock, 2001; Romano et al., 2019; Gammerman et al., 1998; Saunders et al., 1999; Vovk et al., 1999; Angelopoulos & Bates, 2023). More recent work seeks to bridge this gap by designing models that jointly account for, and in some cases explicitly decompose, both types of uncertainty within a single framework (Kendall & Gal, 2017a; Depeweg et al., 2018; Depeweg, 2019; Tagasovska & Lopez-Paz, 2019; Chan et al., 2024).

## 3 CONDITIONAL DIFFUSION-BASED UNCERTAINTY QUANTIFICATION FOR REGRESSION

In this section, we introduce nonparametric regression model with data-dependent noise and our diffusion-based UQ method.

Let $X \in \mathbb{R}^d$ denote the feature vector, and $Y \in \mathbb{R}$ the corresponding response variable. We consider a nonparametric regression model with data-dependent noise, formally specified as

$$Y = f(X) + g(X)\epsilon,$$

where $\epsilon$ is assumed to follow a standard normal distribution, i.e., $\epsilon \sim \mathcal{N}(0,1)$. When $g(X)$ is a constant for any $X$, we recover the standard nonparametric regression model.

We collect i.i.d. samples $\mathcal{D} = \{(x_1, y_1), (x_2, y_2), ..., (x_n, y_n)\}$ from the regression model. Instead of explicitly learning the regression function, our proposal is to capture the conditional distribution of $Y$ given features $x$. This method possesses several advantages: 1) by estimating statistics of the learned conditional distribution, e.g., mean or median, we can form point estimates of $Y$ given $x$ and 2) the conditional distribution provides off-the-shelf estimates of prediction intervals.

Conditional Diffusion Models (CDMs) emerged as a powerful class of generative models that capture complex and high-dimensional distributions. In this paper, we focus on estimating $P(Y|X = x)$ using CDMs. In a nutshell, CDMs learn conditional distributions by two coupled processes, where one corrupts response $Y$ by Gaussian noise and the other attempts to denoise. More specifically, we try to use conditional diffusion model to estimate the full distribution of $Y$ conditioning on $x$.

The forward diffusion process adds noise to $y$ according to an Ornstein–Uhlenbeck (OU) process:

$$dY_t = -\frac{1}{2}Y_t\, dt + dW_t, \quad Y_0 \sim P(\cdot|x), \tag{3.1}$$

where $W_t$ is a standard Wiener process. As $t \to \infty$, the distribution of $Y_t$ converges to a standard Gaussian. We denote the conditional distribution at time $t$ as $P_t(\cdot|x)$. It is worth mentioning that the conditional information $x$ is kept throughout the forward process.

To sample from $P(Y|X = x)$, we reverse the time of the forward SDE in equation 3.1. The reverse-time dynamics are given by

$$dY_t^{\leftarrow} = \left[\frac{1}{2}Y_t^{\leftarrow} + \partial_y \log p_{T-t}(Y_t^{\leftarrow}|x)\right] dt + d\overline{W}_t, \quad Y_0^{\leftarrow} \sim P_T(\cdot|x), \tag{3.2}$$

where $\overline{W}_t$ is another independent Wiener process. The term $\partial_y \log p_{T-t}(Y_t^{\leftarrow}|x)$ is the conditional score function, which is typically unknown and must be estimated.

In practice, the conditional score is parameterized by a neural network. Given an estimated score function $\widehat{s}(x, y, t)$, also replacing the unknown $P_T(\cdot|x)$ by $\mathcal{N}(0,1)$, a tractable reverse process reads

$$d\widetilde{Y}_t^{\leftarrow} = \left[\frac{1}{2}\widetilde{Y}_t^{\leftarrow} + \widehat{s}(x, \widetilde{Y}_t^{\leftarrow}, T-t)\right] dt + d\overline{W}_t, \quad \widetilde{Y}_0^{\leftarrow} \sim \mathcal{N}(0,1). \tag{3.3}$$

Process equation 3.3 is typically terminated at $T - t_0$ for a small early-stopping time (Song & Ermon, 2020; Vahdat et al., 2021b). We denote the distribution of $\widetilde{Y}_{T-t_0}^{\leftarrow}$ as $\widetilde{P}_{T-t_0}(\cdot|x)$, which serves as an approximation to the true distribution $P(\cdot|x)$.

We estimate the score function by minimizing the denoising score matching objective:

$$\mathcal{L}(s) = \frac{1}{T - t_0} \int_{t_0}^{T} \mathbb{E}_{(X,Y_0)} \mathbb{E}_{Y_t \sim \mathcal{N}(\alpha_t Y_0, \sigma_t^2)} \left[ |s(X, Y_t, t) - \partial_{y_t} \log \phi_t(Y_t|Y_0)|^2 \right] dt, \qquad (3.4)$$

where $\phi_t(Y_t|Y_0)$ is the Gaussian transition kernel given by $\partial_{y_t} \log \phi_t(Y_t|Y_0) = -\frac{Y_t - \alpha_t Y_0}{\sigma_t^2}$, with $\alpha_t = e^{-t/2}$ and $\sigma_t^2 = 1 - e^{-t}$.

The empirical risk is computed over data $\{(x_i, y_i)\}_{i=1}^{n}$ and discretized time steps:

$$\widehat{\mathcal{L}}(s) = \frac{1}{n} \sum_{i=1}^{n} \ell(x_i, y_i; s) \qquad (3.5)$$

with $\ell(x, y; s) = \frac{1}{T - t_0} \int_{t_0}^{T} \mathbb{E}_{Y_t \sim \mathcal{N}(\alpha_t y, \sigma_t^2)} \left[ |s(x, Y_t, t) - \partial_{y_t} \log \phi_t(Y_t|y)|^2 \right] dt.$

Given a score network class $\mathcal{F}$, the empirical risk minimizer is denoted by $\widehat{s} \in \arg\min_{s \in \mathcal{F}} \widehat{\mathcal{L}}(s)$. The quality of $\widehat{s}$ is evaluated via its mean squared error:

$$\mathcal{R}(\widehat{s}) = \frac{1}{T - t_0} \int_{t_0}^{T} \mathbb{E}_{(X,Y_t)} \left[ |\widehat{s}(X, Y_t, t) - \partial_{y_t} \log p_t(Y_t|X)|^2 \right] dt. \qquad (3.6)$$

We focus on scalar response $Y$ in the study, however, extension to vector valued response imposes no substantial challenge. When a conditional diffusion model is well-trained on the dataset, we independently draw responses conditioned on $x_{\text{new}}$. From the generated samples, we extract the $\alpha/2$ and $1 - \alpha/2$ quantiles to form a prediction interval. We summarize our conditional diffusion-based uncertainty quantification procedure in the following algorithm.

---

**Algorithm 1** Constructing Prediction Interval via Conditional Diffusion Model

---

**Require:** Training data $\mathcal{D} = \{(x_i, y_i)\}_{i=1}^{n}$, test point $x_{\text{new}}$, significance level $\alpha$
  1: Train a conditional diffusion model to approximate $P(Y|X)$.
  2: Given $x_{\text{new}}$, sample $M$ outputs $\{\widehat{y}_j\}_{j=1}^{M}$ from the trained model.
  3: Compute empirical quantiles:

$$\widehat{q}_{\frac{\alpha}{2}} = \text{empirical } (\alpha/2) \times 100\% \text{ quantile of } \{\widehat{y}_j\}_{j=1}^{M},$$

$$\widehat{q}_{1 - \frac{\alpha}{2}} = \text{empirical } (1 - \alpha/2) \times 100\% \text{ quantile of } \{\widehat{y}_j\}_{j=1}^{M}.$$

  4: **return** Prediction interval $[\widehat{q}_{\frac{\alpha}{2}}, \widehat{q}_{1 - \frac{\alpha}{2}}]$.

---

## 4 THEORY

We analyze the coverage probability of the constructed prediction interval in Section 3. We adopt the common setting in nonparametric regression by imposing Hölder regularity on $f$ and $g$ (Tsybakov, 2008). Before we state the formal assumption, we provide a brief introduction to Hölder spaces.

**Definition 4.1** (Hölder Space). Let $k \in \mathbb{N}_0$ be a non-negative integer, $\alpha \in (0, 1]$ and let $\Omega \subseteq \mathbb{R}^d$ be an open set. A function $f : \Omega \to \mathbb{R}$ is $\alpha$-*Hölder continuous* if there exists a constant $C > 0$ such that

$$|f(x) - f(y)| \leq C \|x - y\|^{\alpha} \quad \text{for all } x, y \in \Omega.$$

Let $\beta = k + \alpha$. The *Hölder space* $\mathcal{H}^{\beta}(\Omega)$ consists of all functions $f : \Omega \to \mathbb{R}$ that are $k$-times continuously differentiable and whose partial derivatives of order $k$ are $\alpha$-Hölder continuous. That is, for every multi-index $\boldsymbol{\gamma} \in \mathbb{N}_0^d$ with $|\boldsymbol{\gamma}| = \sum_{i=1}^{d} \gamma_i = k$, there exists a constant $C_{\gamma} > 0$ such that

$$|D^{\boldsymbol{\gamma}} f(x) - D^{\boldsymbol{\gamma}} f(y)| \leq C_{\gamma} \|x - y\|^{\alpha} \quad \text{for all } x, y \in \Omega.$$

For any $f \in \mathcal{H}^{\beta}(\Omega)$, its *Hölder norm* is defined as

$$\|f\|_{\mathcal{H}^{\beta}(\Omega)} := \sum_{|\boldsymbol{\gamma}| \leq k} \sup_{x \in \mathbb{R}^d} |D^{\boldsymbol{\gamma}} f(x)| + \sum_{|\boldsymbol{\gamma}| = k} \sup_{x \neq y} \frac{|D^{\boldsymbol{\gamma}} f(x) - D^{\boldsymbol{\gamma}} f(y)|}{\|x - y\|^{\alpha}}.$$

We define the *Hölder ball* of radius $B > 0$ for some constant $B$ as

$$\mathcal{H}^\beta(\Omega, B) = \left\{ f : \Omega \to \mathbb{R} \mid \|f\|_{\mathcal{H}^\beta(\Omega)} < B \right\}.$$

For notational convenience, we write $\mathcal{H}^\beta$ instead of $\mathcal{H}^\beta(\Omega)$ whenever the domain is understood from context. We impose the following assumption on $f$ and $g$.

**Assumption 4.2** (Hölder Regularity). Let $f, g : \mathbb{R}^d \to \mathbb{R}$ be functions belonging to the Hölder space $\mathcal{H}^{\beta_f}(\mathbb{R}^d)$ and $\mathcal{H}^{\beta_g}(\mathbb{R}^d)$ for indices $\beta_f, \beta_g \geq 1$. There exist constants $B_f, B_g > 0$ such that

$$\|f\|_{\mathcal{H}^{\beta_f}(\mathbb{R}^d)} \leq B_f \quad \text{and} \quad \|g\|_{\mathcal{H}^{\beta_g}(\mathbb{R}^d)} \leq B_g.$$

Meanwhile, it holds that $\inf_x g(x) \geq g_{\min} > 0$ for a constant $g_{\min}$.

The Hölder continuity on $f$ in Assumption 4.2 is widely studied in regression analysis (Bassanini & Elcrat, 1997). The additional assumption on $g$ implies that the data-dependent noise is typically not negligible. Otherwise, the response $Y$ is nearly deterministic for a given covariate $x$, rendering uncertainty quantification of less importance. A crucial consequence of Assumption 4.2 is that the conditional distribution $P(Y|x)$ has a Hölder continuous density as detailed in the following lemma.

**Lemma 4.3.** Suppose Assumption 4.2 holds. For any constant $B_y > 0$, the restriction of conditional density $p(y|x)$ to $\mathbb{R}^d \times (-B_y, B_y)$ belongs to the Hölder space $\mathcal{H}^\beta(\mathbb{R}^d \times (-B_y, B_y))$, where

$$\beta := \min\left(\lfloor \beta_f \rfloor, \lfloor \beta_g \rfloor\right) + \min\left(\beta_f - \lfloor \beta_f \rfloor, \beta_g - \lfloor \beta_g \rfloor\right). \tag{4.1}$$

Meanwhile, its Hölder norm $\|p(y|x)\|_{\mathcal{H}^\beta(\mathbb{R}^d \times (-B_y, B_y))}$ is bounded by a constant that only depends on $B_y, B_f, B_g, \beta_f, \beta_g, g_{\min}$. Moreover, there exist positive constants $C_1, C_2$ such that for all $y \in \mathbb{R}$, the density function

$$p(y|x) \leq C_1 \exp(-C_2 y^2/2).$$

The proof is provided in Appendix B. Lemma 4.3 says that the conditional density inherits the regularity from the regression functions. Moreover, the response $Y$ is light-tailed. We assume an additional assumption on the marginal distribution of the features.

**Assumption 4.4.** There exist constants $C_{x_1}, C_{x_2}$ such that density $p(x)$ has a sub-Gaussian tail, i.e.,

$$p(x) \leq C_{x_1} \exp(-C_{x_2} \|x\|^2/2).$$

Assumption 4.4 is mild and encodes commonly studied regression settings, e.g., $x$ belongs to a compact region. Notably, this extends the bounded guidance assumption in Fu et al. (2024) to a broader light-tailed setting.

## 4.1 COVERAGE GUARANTEE

The following theorem establishes a theoretical guarantee for the coverage of the prediction intervals generated by our method.

**Theorem 4.5.** Under Assumptions 4.2 and 4.4, for any fixed confidence level $\alpha \in (0, 1)$, with probability $1 - \delta$, for any $x_{\text{new}}$, it holds that

$$\mathbb{E}_{\mathcal{D}}\left[\left|\mathbb{P}[Y \in [\hat{q}_{\frac{\alpha}{2}}, \hat{q}_{1-\frac{\alpha}{2}}]] - (1 - \alpha)\right|\right]$$

$$\leq \sqrt{\frac{2}{n_1} \log \frac{4}{\delta}} + \mathcal{T}(x_{\text{new}}) \cdot \mathcal{O}\left(n^{-\frac{\beta}{4(d+\beta+1)}} (\log n)^{\max\left(9, \frac{3}{2} + \frac{\beta}{4}\right)}\right),$$

with probability $1 - \delta$, where $\beta$ is defined in equation 4.1, $Y \sim P(\cdot|x_{\text{new}})$, $n_1$ denotes the number of samples drawn from the generated distribution $\widetilde{P}_{T-t_0}(Y|X = x_{\text{new}})$, and $\mathcal{T}(x_{\text{new}})$ is a distribution-shift term, defined as

$$\mathcal{T}(x_{\text{new}}) = \sup_{s \in \mathcal{F}} \frac{\sqrt{\mathbb{E}_{Y \sim P(\cdot|x_{\text{new}})}\left[\ell(x_{\text{new}}, Y; s)\right]}}{\sqrt{\mathbb{E}_{X,Y}\left[\ell(X, Y; s)\right]}}.$$

This result asserts that the prediction interval constructed from the empirical quantiles of the learned distribution $\widetilde{P}_{T-t_0}(Y|X = x_{\text{new}})$ ensures a coverage level close to the nominal level $1 - \alpha$. The probability $1 - \delta$ refers to the randomness in the $n_1$ generated samples used to estimate the quantiles. The coverage deviation is pointwise in $x_{\text{new}}$, and bounded by the sum of two terms. The first term stems from estimating quantiles using finitely $n_1$ generated responses. When $n_1$ is sufficiently large, this error term vanishes. The second term corresponds to the error of the learned conditional diffusion model. It scales depending on the smoothness of the conditional density and the data dimensionality.

The coverage of the constructed prediction interval is also influenced by a distribution shift coefficient $\mathcal{T}(x_{\text{new}})$. Roughly speaking, $\mathcal{T}(x_{\text{new}})$ quantifies the relative degradation in score approximation quality at the test input $x_{\text{new}}$ compared to the average over the training distribution. If $x_{\text{new}}$ lies in a region where the score function is poorly estimated, $\mathcal{T}(x_{\text{new}})$ becomes large, amplifying the overall error. Consequently, the coverage guarantee becomes sensitive to distributional shifts between the training and test inputs, with more severe shifts leading to degraded coverage performance; see Figure 2.

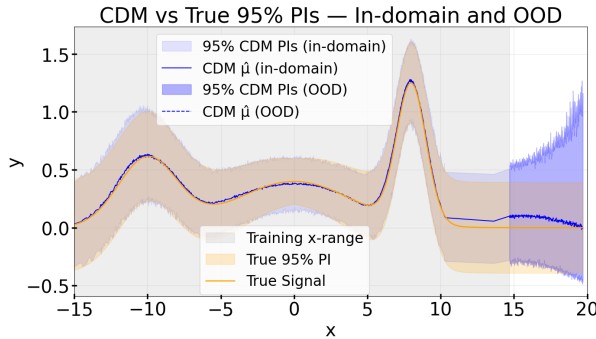

Figure 2: Predictive intervals on the same synthetic regression task in Figure 1. The intervals deviate from ground-truth when the tested instance is out of the range of the training data, due to a large distribution shift.

## 4.2 Proof Sketch

Proof of Theorem 4.5 consists of four main steps, leveraging neural network approximation theory, statistical learning bounds, diffusion process analysis, and probabilistic concentration arguments. We overview the four steps in below.

**Step 1: Choose a score network—approximation of the conditional score.** We rewrite the conditional score function as $\partial_y \log p_t(y|x) = \frac{\partial_y p_t(y|x)}{p_t(y|x)}$. Then we approximate the numerator and denominator separately. This leads to the following score approximation theory.

**Lemma 4.6.** Suppose Assumptions 4.2 and 4.4 hold. Given a sufficiently large integer $N$, for some constants $C_\sigma, C_\alpha$ such that by taking the early-stopping time $t_0 = N^{-C_\sigma}$ and $T = C_\alpha \log N$, there exists a network architecture $\mathcal{F}(M_t, W, \kappa, L, K)$ giving rise to a $\widetilde{s} \in \mathcal{F}$ with

$$\mathbb{E}_{X \sim P_x} \left[ \mathbb{E}_{Y \sim P_t(\cdot|X)} \left[ |\widetilde{s}(X, Y, t) - \partial_y \log p_t(Y|X)|^2 \right] \right] = \mathcal{O}\left( \frac{1}{\sigma_t^4} \cdot N^{-\frac{\beta}{d+1}} \cdot (\log N)^{2+\beta/2} \right).$$

The hyperparameters in the network class $\mathcal{F}$ satisfy

$$M_t = \mathcal{O}\left( \frac{\sqrt{\log N}}{\sigma_t^2} \right), \quad W = \mathcal{O}(N \log^7 N),$$

$$\log \kappa = \mathcal{O}(\log^4 N), \quad L = \mathcal{O}(\log^4 N), \quad K = \mathcal{O}(N \log^9 N),$$

where $\mathcal{O}$ hides polynomial factors depending on $d, C_1, C_2, C_{x_1}, C_{x_2}, B_f, B_g, \beta_f, \beta_g, g_{\min}, C_\alpha$ and $C_\sigma$.

The proof of Lemma 4.6 is provided in Appendix C. The approximation guarantee depends on the time $t$. As can be seen, when $t$ is small, $\sigma_t$ is small, which leads to large approximation errors. This is consistent with the score blow up phenomenon in practical implementation of diffusion models (Song & Ermon, 2020; Vahdat et al., 2021a). A common remedy is to introduce the early-stopping time $t_0$. Lemma 4.6 also suggests that to achieve an $\epsilon/\sigma_t^4$ approximation error for all $t$, the network size scales as $\epsilon^{-\frac{d+1}{\beta}}$, indicating that the smoother the density is, the easier the approximation becomes.

**Step 2: Estimating conditional score—sample complexity bound.** Given the network architecture in **Step 1**, we establish statistical complexities of score estimation. By choosing the network size depending on the training sample size, we show the following generalization bound.

**Lemma 4.7.** Suppose Assumptions 4.2 and 4.4 hold, and choose the score network $\mathcal{F}(M_t, W, \kappa, L, K)$ constructed as in Lemma 4.6 with $N = n^{\frac{d+1}{d+\beta+1}}$. Consider $\widehat{s}$ as the empirical risk minimizer of equation 3.5. It holds that

$$\mathbb{E}_{\mathcal{D}} [\mathcal{R}(\widehat{s})] = \mathcal{O}\left( \frac{1}{t_0} \cdot n^{-\frac{\beta}{d+\beta+1}} (\log n)^{\max(17, 2+\beta/2)} \right).$$

The proof of Lemma 4.7 is provided in Appendix D. This result is the core of the proof, as the accuracy of the score estimation dictates the learning of the conditional distribution as well as the performance of uncertainty quantification.

**Step 3: Learning the conditional distribution—sample complexity and distribution shift.** Using the well trained score function $\widehat{s}$, we can generate responses by specifying a testing feature $x_{\text{new}}$. Recall that we denote the true distribution of response as $P(\cdot|x_{\text{new}})$ and we let $\widetilde{P}_{T-t_0}(Y|X = x_{\text{new}})$ be the generated response distribution. The following lemma bounds the total variation distance between them.

**Lemma 4.8.** Suppose Assumptions 4.2 and 4.4 hold. Set the early-stopping time and terminal time as $t_0 = n^{-\frac{\beta}{2(d+\beta+1)}}$ and $T = \frac{\beta \log n}{4(d+\beta+1)}$, respectively. Then, for any $x_{\text{new}}$, it holds that

$$\mathbb{E}_{\mathcal{D}}\left[\mathrm{TV}\left(\widetilde{P}_{T-t_0}(\cdot|x_{\text{new}}), P(\cdot|x_{\text{new}})\right)\right] = \mathcal{T}(x_{\text{new}}) \cdot \mathcal{O}\left(n^{-\frac{\beta}{4(d+\beta+1)}} (\log n)^{\max\left(9, \frac{3}{2} + \frac{\beta}{4}\right)}\right).$$

The proof of Lemma 4.8 is provided in Appendix E. We observe the presence of the distribution shift coefficient, as $x_{\text{new}}$ may not be encountered during the training.

**Step 4: Quantifying predictive uncertainty—coverage guarantee.** The last step translates the distribution convergence in **Step 3** to coverage guarantees. By the triangle inequality, we have

$$\left|\mathbb{P}\left[Y \in [\widehat{q}_{\frac{\alpha}{2}}, \widehat{q}_{1-\frac{\alpha}{2}}]\right] - (1-\alpha)\right| \leq \left|\mathbb{P}\left[Y \in [\widehat{q}_{\frac{\alpha}{2}}, \widehat{q}_{1-\frac{\alpha}{2}}]\right] - \mathbb{P}\left[\widetilde{Y} \in [\widehat{q}_{\frac{\alpha}{2}}, \widehat{q}_{1-\frac{\alpha}{2}}]\right]\right|$$
$$+ \left|\mathbb{P}\left[\widetilde{Y} \in [\widehat{q}_{\frac{\alpha}{2}}, \widehat{q}_{1-\frac{\alpha}{2}}]\right] - (1-\alpha)\right|,$$

where $\widetilde{Y}$ follows the distribution of $\widetilde{P}_{T-t_0}(Y|X = x_{\text{new}})$.

The first term can be bounded using Lemma 4.8, which asserts the closeness of $\widetilde{P}_{T-t_0}$ and $P$ for a given $x_{\text{new}}$. The second term concerns the convergence of cumulative probability at empirical quantiles to the true cumulative probability, which is bounded by applying Dvoretzky–Kiefer–Wolfowitz inequality (Dvoretzky et al., 1956). The detailed proof is provided in Appendix F. Combining all the steps, we complete the proof of Theorem 4.5.

## 5 EXPERIMENTS

In this section, we assess the effectiveness of our uncertainty quantification methods on ten regression tasks sourced from the UCI Machine Learning Repository (Dua & Graff, 2019). Throughout, we compare CDM against four representative UQ baselines: Split–Conformal Prediction (Split–CP) (Papadopoulos et al., 2002; 2007), Conformal Quantile Regression (CQR) (Romano et al., 2019), Monte Carlo Dropout (MC Dropout) (Gal & Ghahramani, 2016b), and Deep Ensemble (DE) (Lakshminarayanan et al., 2017c). More details about these datasets and baselines are provided in Appendix G.1. The implementation details of our experiments are provided in Appendix G.2.

In these datasets, evaluating the coverage probability for a given $x_{\text{new}}$ is prohibitive, as repeated responses corresponding to the same feature are lacking. Therefore, we report an alternative measure of the coverage probability—prediction interval coverage probability (PICP)—at an $\alpha = 5\%$ level. For each test data $(x_i, y_i)$, we construct a predictive interval $[\widehat{q}_{\frac{\alpha}{2}}^{(i)}, \widehat{q}_{1-\frac{\alpha}{2}}^{(i)}]$ and check if the collected response falls into the interval. PICP is calculated by counting the frequency of $y$ falling into the predictive interval, which has been described in Yao et al. (2019); Han et al. (2022). Formally, we define PICP as

$$\mathrm{PICP} := \frac{1}{n} \sum_{i=1}^{n} \mathbb{1}\{y_i \geq \widehat{q}_{\frac{\alpha}{2}}^{(i)}\} \cdot \mathbb{1}\{y_i \leq \widehat{q}_{1-\frac{\alpha}{2}}^{(i)}\}$$

Alongside PICP, we report root mean squared error (RMSE) and negative log-likelihood (NLL). RMSE quantifies the accuracy of point predictions relative to observed responses. NLL is computed under a Gaussian likelihood assumption, i.e., it evaluates the conditional predictive distribution $p(y|x)$ as Gaussian.

In the remainder of the paper, we denote our proposed Algorithm 1 as the CDM method. Table 1 reports point estimates RMSE and NLL (mean ± s.e.) for each dataset and method, while Table 2

reports the PICP for each dataset. These results demonstrate that CDM delivers consistently competitive and often superior predictive performance across a wide range of regression benchmarks, In terms of NLL, CDM achieves the best values in 8 of 10 datasets, while for RMSE and PICP, it achieves the best in 5 of 10 datasets. These results highlight CDM's ability to provide well-calibrated uncertainty estimates, particularly important in settings with higher dimensions like Protein and Year. It is important to note that in datasets that are very high-dimensional and heavy-tailed with few observations, the NLL Gaussian assumption may be violated, leading to overly penalized likelihood values even when predictive intervals are well calibrated. To further supplement this analysis, we ran additional experiments on a heavy-tailed dataset, with results reported in Appendix G.3. Across datasets, CDM has a similar running-time cost to Deep Ensembles and a substantially lower cost than the most expensive baseline, MC Dropout. Full timing and resource statistics are reported in Appendix G.4.

Table 1: Reported RMSE (↓) and NLL (↓) (mean ± s.e.) for all regression tasks.

| Dataset | Split-CP | | CQR | | CDM | | Deep Ensemble | | MC Dropout | |
|---|---|---|---|---|---|---|---|---|---|---|
| | RMSE | NLL | RMSE | NLL | RMSE | NLL | RMSE | NLL | RMSE | NLL |
| Housing | **2.99±0.82** | 2.56±0.3 | 3.31±0.92 | 2.58±0.27 | 3.01±0.78 | **2.41±0.22** | 3.21±1.06 | 2.45±0.29 | 3.02±0.85 | 2.50±0.28 |
| Concrete | 5.25±0.59 | 3.10±0.1 | 6.04±0.53 | 3.16±0.12 | **4.98±0.64** | **2.91±0.18** | 5.14±0.43 | 2.99±0.21 | 5.26±0.52 | 3.07±0.10 |
| Energy | 0.6±0.07 | 0.93±0.10 | 2.4±0.22 | 1.96±0.52 | **0.55±0.06** | **0.76±0.13** | 2.23±0.26 | 1.48±0.23 | 1.20±0.28 | 1.63±0.19 |
| Kin8nm | **6.62±0.2** | -1.29±0.03 | 6.97±0.25 | -1.3±0.03 | 6.83±0.23 | **−1.31±0.03** | 8.88±0.39 | -1.17±0.03 | 7.05±0.34 | -1.12±0.03 |
| Naval | **0.14±0.11** | -5.72±0.48 | 0.36±0.2 | 1.81±0.27 | **0.14±0.10** | -4.57±0.73 | 0.1±0.01 | **-6.1±0.08** | 0.16±0.04 | -4.41±0.02 |
| Power | 4.08±0.16 | 2.83±0.04 | 4.32±0.14 | 2.87±0.04 | **3.48±0.91** | 2.96±0.15 | 4.04±0.15 | **2.80±0.04** | 4.04 ±0.18 | 2.81±0.04 |
| Protein | 4.21±0.02 | 2.86±0.01 | 5.28±0.05 | 3.22±0.07 | **4.03±0.02** | **2.66±0.03** | 4.46±0.03 | 2.80±0.02 | 4.32±0.03 | 2.88±0.01 |
| Wine | 0.71±0.07 | 1.11±0.07 | 0.66±0.05 | 1.09±0.12 | 0.67±0.07 | **0.89±0.19** | 0.63±0.04 | 0.98±0.16 | **0.62±0.04** | 0.94±0.06 |
| Yacht | 0.61±0.19 | 1.4±0.33 | **1.44±0.46** | 1.47±0.48 | 1.47±0.25 | **0.57±0.09** | 2.21±0.96 | 1.02±0.22 | 1.85±0.64 | 1.87±0.27 |
| Year | 9.72±NA | 3.7±NA | 11.77±NA | 3.46±NA | 9.21±NA | **3.16±NA** | **8.82±NA** | 3.35±NA | 8.83±NA | 3.56±NA |

Table 2: Reported PICP (↑) (mean ± s.e.) for all regression tasks.

| Dataset | Split-CP | CQR | CDM | DE | MC Dropout |
|---|---|---|---|---|---|
| Housing | 95.59±3.56 | 95.98±3.74 | 93.40±3.90 | 89.21±6.39 | **96.08±2.70** |
| Concrete | 95.68±1.80 | 95.87±2.40 | **97.83±1.13** | 91.07±2.33 | 97.52±2.43 |
| Energy | 95.58±3.03 | 96.04±2.58 | 96.10±2.00 | 96.08±2.40 | **99.03±1.08** |
| Kin8nm | 94.88±1.02 | 95.39±0.80 | 94.10±1.50 | **96.36±0.48** | 95.37±2.24 |
| Naval | 95.73±1.21 | 94.85±2.10 | 98.40±0.69 | 93.46±2.19 | **99.77±0.33** |
| Power | 94.97±0.91 | 95.23±1.20 | 94.00±1.00 | 94.29±0.92 | **95.78±1.24** |
| Protein | 95.12±0.67 | 95.09±0.71 | **95.76±0.70** | 93.21±0.65 | 94.96±0.76 |
| Wine | 95.80±2.10 | 95.87±2.16 | **96.80±2.00** | 94.17±2.15 | 95.60±2.21 |
| Yacht | 97.62±2.45 | 97.00±2.10 | **100.00±0.00** | 95.91±2.25 | 98.55±1.46 |
| Year | 94.78±NA | 95.16±NA | **96.01±NA** | 94.05±NA | 94.92±NA |

Looking at RMSE, CDM matches or outperforms competing methods in several benchmarks. In Concrete, Energy, Naval, Power, and Protein, CDM yields lower RMSE compared to both Split-CP and CQR, confirming that its predictive mean estimates remain accurate even without the explicit calibration steps of conformal approaches. Although Deep Ensembles occasionally achieve slightly lower RMSE, CDM achieves consistently comparable performance. Meanwhile, CDM demonstrates excellent performance on the NLL metric. Finally, the PICP results underline CDM's strength in achieving reliable coverage without requiring post hoc conformal adjustments. We observe that CDM consistently surpasses or matches the performance of other baselines. Notably, CDM attains perfect coverage (100%) in Yacht while maintaining competitive RMSE and NLL. Even in cases where MC Dropout achieves slightly higher coverage (e.g., Energy and Naval), it comes at the cost of much weaker NLL and RMSE. Taken together, these results demonstrate that CDM possesses strong uncertainty quantification capabilities for regression tasks, offering a compelling alternative to both ensemble-based and conformal methods in real-world settings.

## 6 CONCLUSION AND LIMITATION

In this paper, we propose a novel uncertainty quantification framework based on conditional diffusion models for nonparametric regression with data-dependent noise. Our approach leverages the expressive power of diffusion models to approximate conditional distributions and construct prediction intervals. Theoretically, we establish convergence guarantees for the coverage probability of the constructed intervals, providing a upper bound that contains score approximation error and quantile estimation error. Empirically, we conduct comprehensive experiments on synthetic and real-world regression tasks, demonstrating that our method consistently achieves accurate and adaptive prediction intervals, outperforming established baselines.

While our CDM method demonstrates strong empirical performance and enjoys theoretical guarantees, one limitation lies in its sensitivity to distribution shift between training and test data. A promising direction for future work is to incorporate weighted score matching to emphasize regions less covered by the training data, potentially mitigating distribution shift and enhancing interval robustness. Another direction is to develop fine-grained analysis for CDM on more complicated UQ tasks, such as dynamical systems and multi-modal predictive distributions.

ETHICS STATEMENT

This paper does not involve human subjects, practices to data set releases, and sensitive applications. We do not anticipate any direct ethical concerns. We have adhered to the ICLR Code of Ethics and confirm that all aspects of this research were conducted in accordance with the principles of fairness, transparency, and research integrity.

REPRODUCIBILITY STATEMENT

We have made all code and scripts available in an anonymized GitHub repository included with the supplementary materials. Full implementation details, including preprocessing steps, model architectures, and training procedures, are described in Appendix G.2 and Appendix G.2.2, with additional dataset-specific settings and baselines outlined in Appendix G.2.1. Evaluation metrics (RMSE, NLL, PICP) are defined in Section 5, and all reported results correspond directly to the tables in Section 5 and Appendix G.3. Together, these materials and references provide a complete pipeline for reproducing both the empirical results and theoretical analyses presented in this paper.

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

## A    EXTENDED RELATED WORK ON DIFFUSION MODELS

Diffusion models are a class of generative models that consist of a forward and a reverse process (Sohl-Dickstein et al., 2015; Song & Ermon, 2019; Song et al., 2020a;b; Ho et al., 2020b;c). The forward process gradually adds noise to the data, transforming it into a standard Gaussian distribution. The reverse process begins with samples from this Gaussian and performs denoising steps to recover samples from the original data distribution. These models are typically trained using score matching objectives (Hyvärinen, 2005; Vincent, 2011; Song et al., 2021) or denoising-based losses (Ho et al., 2020b). Diffusion models have demonstrated remarkable success in generating high-fidelity and diverse data, particularly in the domain of image synthesis (Dhariwal & Nichol, 2021). Despite their generative capabilities, the application of Diffusion models to uncertainty quantification remains relatively underexplored, with only a few recent efforts addressing this direction (Finzi et al., 2023; Han et al., 2022; Berry et al., 2024; Chan et al., 2024).

From the perspective of statistical learning theory, recent works have increasingly focused on the score approximation, distribution estimation, and sampling theory of diffusion models. Theoretical studies on score approximation and estimation aim to establish upper bounds on the sample complexity required to accurately recover the score function. Block et al. (2020) is the first work that rigorously bounds the score estimation error in the $\ell_2$ norm. Building on this, two recent studies from a nonparametric statistics standpoint further enriched the theoretical framework. Chen et al. (2023) developed the approximation theory based on local Taylor expansions, while Oko et al. (2023) proposed an alternative approach using "diffused basis" functions. In both cases, the convergence rates are determined by the smoothness of the score function and the ambient dimension. These studies also contributed to the distribution estimation theory, demonstrating that diffusion models achieve minimax optimality for Besov densities and exhibit adaptivity to linear subspace structures. Extending this line of research, Wibisono et al. (2024) recently applied empirical Bayes theory to investigate score estimation via kernel methods.

In parallel, emerging sampling theories have established that diffusion models are capable of generating distributions that closely approximate the true data distribution, assuming the score estimator achieves sufficient accuracy. Notably, De Bortoli et al. (2021) and Albergo et al. (2023) prove that, for diffusion Schrödinger bridges, the distribution estimation error can be controlled by the score estimation error. A more detailed characterization of the distribution estimation error is provided by Block et al. (2020); Lee et al. (2022); Chen et al. (2022); Lee et al. (2023). Moving beyond Euclidean settings, De Bortoli (2022) establish Wasserstein guarantees on low-dimensional manifolds, albeit with an exponential dependence on the manifold's diameter. Furthermore, diffusion-based techniques have enabled polynomial-time sampling for spiked and Gibbs models (Montanari & Wu, 2023; Alaoui et al., 2023).

## B    PROOF OF LEMMA 4.3

*Proof.* We define $B'_g = \max(B_g, \sqrt{2\pi})$, $g'_{\min} = \min(1, g_{\min})$, $B'_f = \max(B_f, 1)$, $k_f = \lfloor \beta_f \rfloor$, $k_g = \lfloor \beta_g \rfloor$, $\alpha_f = \beta_f - \lfloor \beta_f \rfloor$, $\alpha_g = \beta_g - \lfloor \beta_g \rfloor$.

We begin with a lemma in Hardy (2006).

**Lemma B.1** (Proposition 6 in Hardy (2006)). *Let* $u(x_1, \dots, x_n)$, $v(x_1, \dots, x_n) \in C^\infty(\mathbb{R}^n)$, *and let* $\mathbf{k} = (k_1, \dots, k_n) \in \mathbb{N}^n$ *be a multi-index with total order* $|\mathbf{k}| = k_1 + \cdots + k_n$. *Then*

$$\frac{\partial^{|\mathbf{k}|}(uv)}{\partial x_1^{k_1} \cdots \partial x_n^{k_n}} = \sum_{\ell_1=0}^{k_1} \cdots \sum_{\ell_n=0}^{k_n} \binom{k_1}{\ell_1} \cdots \binom{k_n}{\ell_n} \frac{\partial^{\ell_1 + \cdots + \ell_n} u}{\partial x_1^{\ell_1} \cdots \partial x_n^{\ell_n}} \frac{\partial^{(k_1-\ell_1)+\cdots+(k_n-\ell_n)} v}{\partial x_1^{k_1-\ell_1} \cdots \partial x_n^{k_n-\ell_n}}.$$

For $x \in \mathbb{R}^d$, $y \in \mathbb{R}$, we have the density function of the conditional distribution $p(y|x)$ is

$$p(y|x) = \frac{1}{\sqrt{2\pi} g(x)} \exp\left( -\frac{1}{2} \left( \frac{y - f(x)}{g(x)} \right)^2 \right).$$

We define $u(x) = \frac{1}{\sqrt{2\pi} g(x)}$, and $v(x, y) = \exp\left( -\frac{1}{2} \left( \frac{y-f(x)}{g(x)} \right)^2 \right)$, so that the conditional density can be written as $p(y|x) = u(x)v(x, y)$.

Using Lemma B.1, for a multi-index $\boldsymbol{\gamma} = (\gamma_1, \ldots, \gamma_{d+1}) \in \mathbb{N}^{d+1}$, we have

$$\frac{\partial^{\|\boldsymbol{\gamma}\|_1} p(y|x)}{\partial x_1^{\gamma_1} \cdots \partial x_d^{\gamma_d} \partial y^{\gamma_{d+1}}}$$

$$= \sum_{\ell_1=0}^{\gamma_1} \cdots \sum_{\ell_{d+1}=0}^{\gamma_{d+1}} \binom{\gamma_1}{\ell_1} \cdots \binom{\gamma_{d+1}}{\ell_{d+1}} \frac{\partial^{\ell_1+\cdots+\ell_{d+1}} u(x)}{\partial x_1^{\ell_1} \cdots \partial x_d^{\ell_d} \partial y^{\ell_{d+1}}} \frac{\partial^{(\gamma_1-\ell_1)+\cdots+(\gamma_{d+1}-\ell_{d+1})} v(x,y)}{\partial x_1^{\gamma_1-\ell_1} \cdots \partial x_d^{\gamma_d-\ell_d} \partial y^{\gamma_{d+1}-\ell_{d+1}}}$$

$$= \sum_{\ell_1=0}^{\gamma_1} \cdots \sum_{\ell_d=0}^{\gamma_d} \binom{\gamma_1}{\ell_1} \cdots \binom{\gamma_d}{\ell_d} \frac{\partial^{\ell_1+\cdots+\ell_d} u(x)}{\partial x_1^{\ell_1} \cdots \partial x_d^{\ell_d}} \frac{\partial^{(\gamma_1-\ell_1)+\cdots+(\gamma_d-\ell_d)+\gamma_{d+1}} v(x,y)}{\partial x_1^{\gamma_1-\ell_1} \cdots \partial x_d^{\gamma_d-\ell_d} \partial y^{\gamma_{d+1}}}.$$

The second equation holds because when $\ell_{d+1} \geq 1$, $\frac{\partial^{\ell_1+\cdots+\ell_{d+1}} u(x)}{\partial x_1^{\ell_1} \cdots \partial x_d^{\ell_d} \partial y^{l_{d+1}}} = 0$.

For

$$\frac{\partial^{\ell_1+\cdots+\ell_d} u(x)}{\partial x_1^{\ell_1} \cdots \partial x_d^{\ell_d}} \quad \text{and} \quad \frac{\partial^{(\gamma_1-\ell_1)+\cdots+(\gamma_d-\ell_d)+\gamma_{d+1}} v(x,y)}{\partial x_1^{\gamma_1-\ell_1} \cdots \partial x_d^{\gamma_d-\ell_d} \partial y^{\gamma_{d+1}}},$$

we compute them using the following lemma, which provides the multivariate chain rule for higher-order mixed partial derivatives.

**Lemma B.2** (Corollary to Propositions 1 & 2 in Hardy (2006)). Fix integers $k_1, \ldots, k_n \geq 0$ and set

$$\tau = \{\underbrace{1, \ldots, 1}_{k_1}, \underbrace{2, \ldots, 2}_{k_2}, \ldots, \underbrace{n, \ldots, n}_{k_n}\}, \qquad |\tau| := k_1 + \cdots + k_n.$$

For a smooth scalar field $y : \mathbb{R}^n \to \mathbb{R}$ and a function $f \in C^\infty(\mathbb{R})$ write

$$\partial_\tau := \frac{\partial^{\|\tau\|_1}}{\partial x_1^{k_1} \ldots \partial x_n^{k_n}}.$$

Then

$$\partial_\tau f\big(g(x)\big) = \sum_{\tau=m_1\tau_1+m_2\tau_2+\cdots+m_t\tau_t} M_{\{m_1\tau_1, m_2\tau_2, \cdots, m_t\tau_t\}} f^m\big(g(x)\big) \prod_{i=1}^{t} \prod_{j=1}^{m_i} \partial_{\tau_i} g(x),$$

where $m = \sum_{i=1}^{t} m_i$ is the number of blocks in the partition, hence the order of the derivative $f^{(m)}$, and the sum runs over all multiset-partitions $\tau = m_1\tau_1 + m_2\tau_2 + \cdots + m_t\tau_t$. $\partial_{\tau_j}$ denotes the mixed-derivative operator associated with the sub-multiset $\tau_j$, $M_{\{m_1\tau_1, m_2\tau_2, \cdots, m_t\tau_t\}}$ is the number of ordinary set-partitions of $\{1, 2, \ldots, |\tau|\}$ that collapse to the given multiset-partition of $\tau$.

We define the multi-index $\boldsymbol{\ell} = (\ell_1, \ldots, \ell_{n+1}) \in \mathbb{N}^{d+1}$, and set $\boldsymbol{h} = \boldsymbol{\gamma} - \boldsymbol{\ell}$, so that each component satisfies $h_i = \gamma_i - \ell_i$.

Additionally, we introduce the following auxiliary functions for notational convenience:

$$f_u(x) := \frac{1}{\sqrt{2\pi}\, x}, \qquad g_u(x) := g(x)$$

$$f_v(x) := \exp\left(-\frac{1}{2}x^2\right), \qquad g_v(x,y) := \frac{y - f(x)}{g(x)},$$

so that $u(x) = f_u(g_u(x))$ and $v(x,y) = f_v(g_v(x,y))$.

From Lemma B.2, we have

$$\frac{\partial^{\ell_1+\cdots+\ell_d} u(x)}{\partial x_1^{\ell_1} \cdots \partial x_d^{\ell_d}}$$

$$= \sum_{\tau=m_1\tau_1+m_2\tau_2+\cdots+m_t\tau_t} M_{\{m_1\tau_1, m_2\tau_2, \cdots, m_t\tau_t\}} f_u^{(m)}\big(g_u(x)\big) \prod_{i=1}^{t} \prod_{j=1}^{m_i} \partial_{\tau_i} g_u(x)$$

$$= \sum_{\tau=m_1\tau_1+m_2\tau_2+\cdots+m_t\tau_t} M_{\{m_1\tau_1, m_2\tau_2, \cdots, m_t\tau_t\}} \frac{m!(-1)^m}{\sqrt{2\pi} g(x)^{m+1}} \prod_{i=1}^{t} \prod_{j=1}^{m_i} \partial_{\tau_i} g(x). \tag{B.1}$$

and

$$\frac{\partial^{(\gamma_1-\ell_1)+\cdots+(\gamma_d-\ell_d)+\gamma_{d+1}}v(x,y)}{\partial x_1^{\gamma_1-\ell_1}\cdots\partial x_d^{\gamma_d-\ell_d}\partial y^{\beta_{d+1}}}$$

$$=\frac{\partial^{h_1+\cdots+h_d+\gamma_{d+1}}v(x,y)}{\partial x_1^{h_1}\cdots\partial x_d^{h_d}\partial y^{\gamma_{d+1}}}$$

$$=\sum_{\tau=m_1\tau_1+m_2\tau_2+\cdots+m_t\tau_t}M_{\{m_1\tau_1,m_2\tau_2,\cdots,m_t\tau_t\}}f_v^{(m)}\big(g_v(x,y)\big)\prod_{i=1}^{t}\prod_{j=1}^{m_i}\partial_{\tau_i}g_v(x,y). \quad \text{(B.2)}$$

We define $k_p=\min(k_f,k_g)$, and $\alpha_p=\min(\alpha_f,\alpha_g)$.

For $|D^{\gamma}p(y|x)|=\left|\frac{\partial^{|\gamma|}p(y|x)}{\partial x_1^{\gamma_1}\cdots\partial x_d^{\gamma_d}\partial y^{\gamma_{d+1}}}\right|$, we have

$$\left|\frac{\partial^{|\gamma|}p(y|x)}{\partial x_1^{\gamma_1}\cdots\partial x_d^{\gamma_d}\partial y^{\gamma_{d+1}}}\right|$$

$$\leq\sum_{\ell_1=0}^{\gamma_1}\cdots\sum_{\ell_d=0}^{\gamma_d}\binom{\gamma_1}{\ell_1}\cdots\binom{\gamma_d}{\ell_d}\left|\frac{\partial^{\ell_1+\cdots+\ell_d}u(x)}{\partial x_1^{\ell_1}\cdots\partial x_d^{\ell_d}}\right|\left|\frac{\partial^{(\gamma_1-\ell_1)+\cdots+(\gamma_d-\ell_d)+\gamma_{d+1}}v(x,y)}{\partial x_1^{\gamma_1-\ell_1}\cdots\partial x_d^{\gamma_d-\ell_d}\partial y^{\gamma_{d+1}}}\right|. \quad \text{(B.3)}$$

From equation B.1, we have

$$\left|\frac{\partial^{\ell_1+\cdots+\ell_d}u(x)}{\partial x_1^{\ell_1}\cdots\partial x_d^{\ell_d}}\right|$$

$$\leq\sum_{\tau=m_1\tau_1+m_2\tau_2+\cdots+m_t\tau_t}M_{\{m_1\tau_1,m_2\tau_2,\cdots,m_t\tau_t\}}\frac{m!}{\sqrt{2\pi}g(x)^{m+1}}\prod_{i=1}^{t}(|\partial_{\tau_i}g(x)|)^{m_i}$$

$$\leq\sum_{\tau=m_1\tau_1+m_2\tau_2+\cdots+m_t\tau_t}M_{\{m_1\tau_1,m_2\tau_2,\cdots,m_t\tau_t\}}\frac{m!}{\sqrt{2\pi}g(x)^{m+1}}\left(B_g\right)^m$$

$$\leq\frac{(k_p!)B_{k_p}(B_g)^{k_p}}{\sqrt{2\pi}(g'_{\min})^{k_p+1}}. \quad \text{(B.4)}$$

In the final inequality, we invoke the Bell number $B_{k_p}$, which denotes the number of ways to partition a set of $k_p$ elements into non-empty subsets. Specifically, we use the identity

$$\sum_{\tau=m_1\tau_1+m_2\tau_2+\cdots+m_t\tau_t}M_{\{m_1\tau_1,m_2\tau_2,\ldots,m_t\tau_t\}}=B_{\|\boldsymbol{\ell}\|_1},$$

where $B_{\|\boldsymbol{\ell}\|_1}$ is the Bell number corresponding to $\|\boldsymbol{\ell}\|_1$, the total number of derivatives taken. Since $\|\boldsymbol{\ell}\|_1\leq k_p$, the inequality follows from the monotonicity of Bell numbers:

$$B_{\|\boldsymbol{\ell}\|_1}\leq B_{k_p}.$$

From equation B.2, we have

$$\left|\frac{\partial^{(\beta_1-\ell_1)+\cdots+(\beta_d-\ell_d)+\beta_{d+1}}v(x,y)}{\partial x_1^{\beta_1-\ell_1}\cdots\partial x_d^{\beta_d-\ell_d}\partial y^{\beta_{d+1}}}\right|$$

$$\leq\sum_{\tau=m_1\tau_1+m_2\tau_2+\cdots+m_t\tau_t}M_{\{m_1\tau_1,m_2\tau_2,\cdots,m_t\tau_t\}}\left|f_v^{(m)}\big(g_v(x,y)\big)\right|\prod_{i=1}^{t}\prod_{j=1}^{m_i}|\partial_{\tau_i}g_v(x,y)|$$

$$\leq\sum_{\tau=m_1\tau_1+m_2\tau_2+\cdots+m_t\tau_t}M_{\{m_1\tau_1,m_2\tau_2,\cdots,m_t\tau_t\}}\left|f_v^{(m)}\big(g_v(x,y)\big)\right|\prod_{i=1}^{t}\prod_{j=1}^{m_i}|\partial_{\tau_i}g_v(x,y)|. \quad \text{(B.5)}$$

For $|\partial_{\tau_i}g_v(x,y)|$, following Lemma B.1, and using the result in equation B.4, we have

$$|\partial_{\tau_i}g_v(x,y)|$$

$$= \left| \sum_{\ell'_1=0}^{\tau_{i_1}} \cdots \sum_{\ell'_d=0}^{\tau_{i_d}} \binom{\tau_{i_1}}{\ell'_1} \cdots \binom{\tau_{i_d}}{\ell'_d} \frac{\partial^{\ell'_1+\cdots+\ell'_n+\tau_{i_{d+1}}}(y-f(x))}{\partial x_1^{\ell'_1} \cdots \partial x_n^{\ell'_d} \partial y^{\tau'_{i_{d+1}}}} \frac{\partial^{(\tau_{i_1}-\ell'_1)+\cdots+(\tau_{i_d}-\ell'_d)}(\sqrt{2\pi}\,u)}{\partial x_1^{\tau_{i_1}-\ell'_1} \cdots \partial x_n^{\tau_{i_d}-\ell'_d}} \right|$$

$$\leq \sum_{\ell'_1=0}^{\tau_{i_1}} \cdots \sum_{\ell'_d=0}^{\tau_{i_d}} \binom{\tau_{i_1}}{\ell'_1} \cdots \binom{\tau_{i_d}}{\ell'_d} \left| \frac{\partial^{\ell'_1+\cdots+\ell'_n+\tau_{i_{d+1}}}(y-f(x))}{\partial x_1^{\ell'_1} \cdots \partial x_n^{\ell'_d} \partial y^{\tau'_{i_{d+1}}}} \right| \left| \frac{\partial^{(\tau_{i_1}-\ell'_1)+\cdots+(\tau_{i_d}-\ell'_d)}(\sqrt{2\pi}\,u)}{\partial x_1^{\tau_{i_1}-\ell'_1} \cdots \partial x_n^{\tau_{i_d}-\ell'_d}} \right|$$

$$\leq \sum_{\ell'_1=0}^{\tau_{i_1}} \cdots \sum_{\ell'_n=0}^{\tau_{i_n}} \binom{\tau_{i_1}}{\ell'_1} \cdots \binom{\tau_{i_n}}{\ell'_n} \frac{(|y|+B_f)(k_p!)B_{k_p}(B_g)^{k_p}}{(g'_{\min})^{k_p+1}}$$

$$\leq \frac{(|y|+B_f)(k_p!)B_{k_p}(2B_g)^{k_p}}{(g'_{\min})^{k_p+1}}.$$

For $\left| f_v^{(m)}(g_v(x,y)) \right|$, let $h_v(x) = e^x$, $e_v(x) = -\frac{1}{2}x^2$, so $f_v(x) = h_v(e_v(x))$, using Faà di Bruno's formula, we have:

$$f_v^{(m)}(x)$$

$$= \sum_{a_1+2a_2+\cdots+na_n=m} \frac{m!}{a_1!(1!)^{a_1}a_2!(2!)^{a_2}\cdots a_n!(n!)^{a_n}} \cdot h_v^{(a_1+\cdots+a_n)}(e_v(x)) \cdot \prod_{j=1}^n \left( \frac{e_v^{(j)}(x)}{j!} \right)^{a_j}$$

$$= f_v(x) \sum_{a_1+2a_2+\cdots+na_n=m} \frac{m!}{a_1!(1!)^{a_1}a_2!(2!)^{a_2}\cdots a_n!(n!)^{a_n}} \prod_{j=1}^m \left( \frac{e_v^{(j)}(x)}{j!} \right)^{a_j}$$

$$= \exp(-x^2/2) \left( (-x)^m + \sum_{a_1+2a_2=m} \frac{m!}{a_1!a_2!2^{a_2}}(-x)^{a_1}(-1/2)^{a_2} \right).$$

The first inequality holds because $\frac{d}{dx^n}(e^x) = e^x$, the second inequality holds because $\frac{e_v^{(j)}(x)}{j!} = 0$, when $j \geq 3$. So we have

$$\left| f_v^{(m)}(g_v(x,y)) \right|$$

$$= \left| f_v(g_v(x,y)) \left( (-g_v(x,y))^m + \sum_{a_1+2a_2=m} \frac{m!}{a_1!a_2!2^{a_2}}(-g_v(x,y))^{a_1}(-1/2)^{a_2} \right) \right|$$

$$\leq f_v(g_v(x,y)) \left( |g_v(x,y)|^m + \sum_{a_1+2a_2=m} \frac{m!}{a_1!a_2!2^{a_2}}|g_v(x,y)|^{a_1}(1/2)^{a_2} \right).$$

Thus implies

$$\left| f_v^{(m)}(g_v(x,y)) \prod_{i=1}^t \prod_{j=1}^{m_i} \partial_{\tau_i} g_v(x,y) \right| \tag{B.6}$$

$$\leq \left( \frac{(|y|+B_f)(k_p!)B_{k_p}(2B_g)^{k_p}}{(g'_{\min})^{k_p+1}} \right)^m \left| f_v^{(m)}(g_v(x,y)) \right|$$

$$\leq \left( \frac{(|y-f(x)|+2B_f)(k_p!)B_{k_p}(2B_g)^{k_p}}{(g'_{\min})^{k_p+1}} \right)^m \left| f_v^{(m)}(g_v(x,y)) \right|$$

$$= \left( \frac{(g(x)|g_v(x,y)|+2B_f)(k_p!)B_{k_p}(2B_g)^{k_p}}{(g'_{\min})^{k_p+1}} \right)^m \left| f_v^{(m)}(g_v(x,y)) \right|$$

$$\leq \left( \frac{(B_g|g_v(x,y)|+2B_f)(k_p!)B_{k_p}(2B_g)^{k_p}}{(g'_{\min})^{k_p+1}} \right)^m \left| f_v^{(m)}(g_v(x,y)) \right|$$

$$\leq \left| \exp\left( -\frac{1}{2}(g_v(x,y))^2 \right) F(|g_v(x,y)|; B_g, B_f, k_p, m, g'_{\min}) \right|. \tag{B.7}$$

where

$$F(x; B_g, B_f, k_p, m, g'_{\min})$$

$$= \left( \frac{(B_g|g_v(x,y)| + 2B_f)(k_p!)B_{k_p}(2B_g)^{k_p}}{(g'_{\min})^{k_p+1}} \right)^m \left( x^m + \sum_{a_1+2a_2=m} \frac{m!}{a_1!a_2!2^{a_2}} x^{a_1}(1/2)^{a_2} \right)$$

$$= \sum_{i=0}^{2m} F_i x^i,$$

is a polynomial of degree 2m, and the coefficient $F_i$ is determined by constants $B_g, B_f, k_p, m, g'_{\min}$.

We can easily show that, for every $0 \leq i \leq 2k_p$, $G_i(x) = x^i \exp(-\frac{1}{2}x^2)$ is bounded above and below by analyzing their first derivatives.

Then we define $C(B_g, B_f, k_p, g'_{\min}) = \sup_{1 \leq i \leq 2k_p} \max\{F_i|G_i(x)|\}$, so it is a constant depending on $B_g, B_f, k_p, g'_{\min}$, thus depending on $\beta_p$.

So from Equation (B.7)

$$\left| f_v^{(m)}(g_v(x,y)) \prod_{i=1}^{t} \prod_{j=1}^{m_i} \partial_{\tau_i} g_v(x,y) \right| \leq \left| \exp\left( -\frac{1}{2}(g_v(x,y))^2 \right) F(|g_v(x,y)|; B_g, B_f, k_p, m, g'_{\min}) \right|$$

$$\leq \left| \sum_{i=0}^{2m} F_i(g_v(x,y))^i \exp\left( -\frac{1}{2}(g_v(x,y))^2 \right) \right|$$

$$\leq \sum_{i=0}^{2m} \left| F_i(g_v(x,y))^i \exp\left( -\frac{1}{2}(g_v(x,y))^2 \right) \right|$$

$$\leq 2mC(B_g, B_f, k_p, g'_{\min}). \tag{B.8}$$

So combining equation B.5 and equation B.8, we have

$$\left| \frac{\partial^{(\beta_1-\ell_1)+\cdots+(\beta_d-\ell_d)+\beta_{d+1}} v(x,y)}{\partial x_1^{\beta_1-\ell_1} \cdots \partial x_d^{\beta_d-\ell_d} \partial y^{\beta_{d+1}}} \right|$$

$$\leq \sum_{\tau=m_1\tau_1+m_2\tau_2+\cdots+m_t\tau_t} M_{\{m_1\tau_1,m_2\tau_2,\cdots,m_t\tau_t\}} \left| f_v^{(m)}(g_v(x,y)) \right| \prod_{i=1}^{t} \prod_{j=1}^{m_i} |\partial_{\tau_i} g_v(x,y)|$$

$$\leq \sum_{\tau=m_1\tau_1+m_2\tau_2+\cdots+m_t\tau_t} M_{\{m_1\tau_1,m_2\tau_2,\cdots,m_t\tau_t\}} \cdot 2mC(B_g, B_f, k_p, g'_{\min})$$

$$\leq 2k_pC(B_g, B_f, k_p, g'_{\min}) \sum_{\tau=m_1\tau_1+m_2\tau_2+\cdots+m_t\tau_t} M_{\{m_1\tau_1,m_2\tau_2,\cdots,m_t\tau_t\}}$$

$$= 2k_pC(B_g, B_f, k_p, g'_{\min})B_{k_p}. \tag{B.9}$$

Combining equation B.3, equation B.4 and equation B.9, we can conclude

$$\left| \frac{\partial^{|\gamma|} p(y|x)}{\partial x_1^{\gamma_1} \cdots \partial x_d^{\gamma_d} \partial y^{\gamma_{d+1}}} \right|$$

$$\leq \sum_{\ell_1=0}^{\gamma_1} \cdots \sum_{\ell_d=0}^{\gamma_d} \binom{\gamma_1}{\ell_1} \cdots \binom{\gamma_d}{\ell_d} \left| \frac{\partial^{\ell_1+\cdots+\ell_d} u(x)}{\partial x_1^{\ell_1} \cdots \partial x_d^{\ell_d}} \right| \left| \frac{\partial^{(\gamma_1-\ell_1)+\cdots+(\gamma_d-\ell_d)+\gamma_{d+1}} v(x,y)}{\partial x_1^{\gamma_1-\ell_1} \cdots \partial x_d^{\gamma_d-\ell_d} \partial y^{\gamma_{d+1}}} \right|$$

$$\leq \frac{2k_pC(B_g, B_f, k_p, g'_{\min})(k_p!)(B_{k_p})^2(2B_g)^{k_p}}{\sqrt{2\pi}(g'_{\min})^{k_p+1}} := C'_1(B_g, B_f, k_p, g'_{\min}).$$

For the $\alpha_p$-Hölder continuity of the partial derivatives of $p(y|x)$, we begin by checking if $\frac{\partial^{\ell_1+\cdots+\ell_d} u(x)}{\partial x_1^{\ell_1} \cdots \partial x_d^{\ell_d}}$ and $\frac{\partial^{(\gamma_1-\ell_1)+\cdots+(\gamma_d-\ell_d)+\gamma_{d+1}} v(x,y)}{\partial x_1^{\gamma_1-\ell_1} \cdots \partial x_d^{\gamma_d-\ell_d} \partial y^{\gamma_{d+1}}}$ are $\alpha_p$-Hölder continuous.

We define $z = (x, y)$, when $\|z - z'\| \geq 1$, due to upper bounds we just derive for these partial derivatives, they must be $\alpha_p$-Hölder continuous, because for $p(y|x)$,

$$|D^\gamma p(y|x) - D^\gamma p(y'|x')| \leq 2C_1(B_g, B_f, k_p, g'_{\min}) \leq 2C_1(B_g, B_f, k_p, g'_{\min}) \|z - z'\|^{\alpha_p}.$$

Next, we only consider $\|z - z'\| \leq 1$

For $\frac{\partial^{\ell_1 + \cdots + \ell_d} u(x)}{\partial x_1^{\ell_1} \cdots \partial x_d^{\ell_d}}$, any $x \neq x' \in \mathbb{R}^d$, from equation B.1, we have

$$\left| \frac{\partial^{\ell_1 + \cdots + \ell_d} u(x)}{\partial x_1^{\ell_1} \cdots \partial x_d^{\ell_d}} - \frac{\partial^{\ell_1 + \cdots + \ell_d} u(x')}{\partial x_1^{\ell_1} \cdots \partial x_d^{\ell_d}} \right|$$

$$= \sum_{\tau = m_1\tau_1 + m_2\tau_2 + \cdots + m_t\tau_t} M \left| \frac{m!(-1)^m}{\sqrt{2\pi} g(x)^{m+1}} \prod_{i=1}^{t} \prod_{j=1}^{m_i} \partial_{\tau_i} g(x) - \frac{m!(-1)^m}{\sqrt{2\pi} g(x')^{m+1}} \prod_{i=1}^{t} \prod_{j=1}^{m_i} \partial_{\tau_i} g(x') \right|$$

$$\leq \sum_{\tau = m_1\tau_1 + m_2\tau_2 + \cdots + m_t\tau_t} M \underbrace{\left| \frac{m!(-1)^m}{\sqrt{2\pi} g(x)^{m+1}} \prod_{i=1}^{t} \prod_{j=1}^{m_i} \partial_{\tau_i} g(x) - \frac{m!(-1)^m}{\sqrt{2\pi} g(x')^{m+1}} \prod_{i=1}^{t} \prod_{j=1}^{m_i} \partial_{\tau_i} g(x) \right|}_{B_1}$$

$$+ M \underbrace{\left| \frac{m!(-1)^m}{\sqrt{2\pi} g(x')^{m+1}} \prod_{i=1}^{t} \prod_{j=1}^{m_i} \partial_{\tau_i} g(x) - \frac{m!(-1)^m}{\sqrt{2\pi} g(x')^{m+1}} \prod_{i=1}^{t} \prod_{j=1}^{m_i} \partial_{\tau_i} g(x') \right|}_{B_2},$$

where $M = M_{\{m_1\tau_1, m_2\tau_2, \cdots, m_t\tau_t\}}$.

For $(B_1)$, we have

$$\left| \frac{m!(-1)^m}{\sqrt{2\pi} g(x)^{m+1}} \prod_{i=1}^{t} \prod_{j=1}^{m_i} \partial_{\tau_i} g(x) - \frac{m!(-1)^m}{\sqrt{2\pi} g(x')^{m+1}} \prod_{i=1}^{t} \prod_{j=1}^{m_i} \partial_{\tau_i} g(x) \right|$$

$$= \left| \frac{m!(-1)^m}{\sqrt{2\pi}} \left( \frac{1}{g(x)^{m+1}} - \frac{1}{g(x')^{m+1}} \right) \prod_{i=1}^{t} \prod_{j=1}^{m_i} \partial_{\tau_i} g(x) \right|$$

$$= \left| \frac{m!(-1)^m}{\sqrt{2\pi}} \left( -(m+1) \xi^{-(m+2)} \cdot (g(x) - g(x')) \right) \prod_{i=1}^{t} \prod_{j=1}^{m_i} \partial_{\tau_i} g(x) \right|$$

$$\leq \frac{m!(m+1)(B_g)^m}{\sqrt{2\pi}(g'_{\min})^{k_p+2}} |g(x) - g(x')|$$

$$\leq \frac{m!(m+1)(B_g)^m}{\sqrt{2\pi}(g'_{\min})^{k_p+2}} B_g \|x - x'\|_2^{\alpha_g}$$

$$\leq \frac{(k_p)!(k_p+1)(B_g)^{k_p+1}}{\sqrt{2\pi}(g'_{\min})^{k_p+2}} \|x - x'\|_2^{\alpha_p},$$

In the second equation, we apply the mean value theorem to the function $\frac{1}{x}$, where $\xi \in [\min(g(x), g(x')), \max(g(x), g(x'))]$.

We define $C_2'(k_p, B_g, g'_{\min}) = \frac{(k_p)!(k_p+1)(B_g)^{k_p+1}}{\sqrt{2\pi}(g'_{\min})^{k_p+2}}$, then we obtain

$$\left| \frac{m!(-1)^m}{\sqrt{2\pi} g(x)^{m+1}} \prod_{i=1}^{t} \prod_{j=1}^{m_i} \partial_{\tau_i} g(x) - \frac{m!(-1)^m}{\sqrt{2\pi} g(x')^{m+1}} \prod_{i=1}^{t} \prod_{j=1}^{m_i} \partial_{\tau_i} g(x) \right|$$

$$\leq C_2'(k_p, B_g, g'_{\min}) \|x - x'\|_2^{\alpha_p}.$$

For $(B_2)$, applying the triangle inequality, we obtain

$$\left| \frac{m!(-1)^m}{\sqrt{2\pi}g(x')^{m+1}} \prod_{i=1}^{t}\prod_{j=1}^{m_i} \partial_{\tau_i} g(x) - \frac{m!(-1)^m}{\sqrt{2\pi}g(x')^{m+1}} \prod_{i=1}^{t}\prod_{j=1}^{m_i} \partial_{\tau_i} g(x') \right|$$

$$\leq \frac{m!}{\sqrt{2\pi}(g'_{\min})^{m+1}} \left| \prod_{i=1}^{t}(\partial_{\tau_i} g(x))^{m_i} - \prod_{i=1}^{t}(\partial_{\tau_i} g(x'))^{m_i} \right|.$$

For $\left| \prod_{i=1}^{t}(\partial_{\tau_i} g(x))^{m_i} - \prod_{i=1}^{t}(\partial_{\tau_i} g(x'))^{m_i} \right|$, we have

$$\left| \prod_{i=1}^{t}(\partial_{\tau_i} g(x))^{m_i} - \prod_{i=1}^{t}(\partial_{\tau_i} g(x'))^{m_i} \right|$$

$$\leq \sum_{j=0}^{t-1} \left| \prod_{i=1}^{j}(\partial_{\tau_i} g(x'))^{m_i} \prod_{i=j+1}^{t}(\partial_{\tau_i} g(x))^{m_i} - \prod_{i=1}^{j+1}(\partial_{\tau_i} g(x'))^{m_i} \prod_{i=j+2}^{t}(\partial_{\tau_i} g(x))^{m_i} \right|$$

$$= \sum_{j=0}^{t-1} \left| \prod_{i=1}^{j}(\partial_{\tau_i} g(x'))^{m_i} \prod_{i=j+2}^{t}(\partial_{\tau_i} g(x))^{m_i} \left( (\partial_{\tau_{j+1}} g(x))^{m_{j+1}} - (\partial_{\tau_{j+1}} g(x'))^{m_{j+1}} \right) \right|$$

$$\leq \sum_{j=0}^{t-1} \left| \prod_{i=1}^{j}(\partial_{\tau_i} g(x'))^{m_i} \prod_{i=j+2}^{t}(\partial_{\tau_i} g(x))^{m_i} (m_{j+1})(\xi_j^{m_{j+1}-1})(\partial_{\tau_i} g(x) - \partial_{\tau_i} g(x')) \right|$$

$$\leq m(B_g)^m t B_g \|x - x'\|_2^{\alpha_g}.$$

where $\xi_j \in [\min(\partial_{\tau_i} g(x), \partial_{\tau_i} g(x')), \max(\partial_{\tau_i} g(x), \partial_{\tau_i} g(x'))]$. In the first inequality we apply the triangle inequality, and in the second inequality we use the mean value theorem again.

Then we can conclude

$$\frac{m!}{\sqrt{2\pi}(g'_{\min})^{m+1}} \left| \prod_{i=1}^{t}(\partial_{\tau_i} g(x))^{m_i} - \prod_{i=1}^{t}(\partial_{\tau_i} g(x'))^{m_i} \right| \leq \frac{m! m(B_g)^m t}{\sqrt{2\pi}(g'_{\min})^{m+1}} B_g \|x - x'\|_2^{\alpha_g}$$

$$\leq \frac{(k_p)!(k_p)^2(B_g)^{k_p+1}}{\sqrt{2\pi}(g'_{\min})^{k_p+1}} \|z - z'\|_2^{\alpha_p}.$$

We define $C_3'(k_p, B_g, g'_{\min}) = \frac{(k_p)!(k_p)^2(B_g)^{k_p+1}}{\sqrt{2\pi}(g'_{\min})^{k_p+1}}$, so we can conclude

$$\left| \frac{m!(-1)^m}{\sqrt{2\pi}g(x')^{m+1}} \prod_{i=1}^{t}\prod_{j=1}^{m_i} \partial_{\tau_i} g(x) - \frac{m!(-1)^m}{\sqrt{2\pi}g(x')^{m+1}} \prod_{i=1}^{t}\prod_{j=1}^{m_i} \partial_{\tau_i} g(x') \right|$$

$$\leq C_3'(k_p, B_g, g'_{\min}) \|z - z'\|_2^{\alpha_p}.$$

Then we check $\frac{\partial^{(\gamma_1-\ell_1)+\cdots+(\gamma_d-\ell_d)+\gamma_{d+1}} v(x,y)}{\partial x_1^{\gamma_1-\ell_1} \cdots \partial x_d^{\gamma_d-\ell_d} \partial y^{\gamma_{d+1}}}$, define $z = (x,y), z' = (x',y')$, for any $z, z' \in \mathbb{R}^{d+1}$, from equation B.2

$$\left| \frac{\partial^{(\gamma_1-\ell_1)+\cdots+(\gamma_d-\ell_d)+\gamma_{d+1}} v(x,y)}{\partial x_1^{\gamma_1-\ell_1} \cdots \partial x_d^{\gamma_d-\ell_d} \partial y^{\gamma_{d+1}}} - \frac{\partial^{(\gamma_1-\ell_1)+\cdots+(\gamma_d-\ell_d)+\gamma_{d+1}} v(x',y')}{\partial x_1^{\gamma_1-\ell_1} \cdots \partial x_d^{\gamma_d-\ell_d} \partial y^{\gamma_{d+1}}} \right|$$

$$\leq \sum_{\tau = m_1\tau_1 + m_2\tau_2 + \cdots + m_t\tau_t} M \left| f_v^{(m)}(g_v(z)) \prod_{i=1}^{t}\prod_{j=1}^{m_i} \partial_{\tau_i} g_v(z) - f_v^{(m)}(g_v(z')) \prod_{i=1}^{t}\prod_{j=1}^{m_i} \partial_{\tau_i} g_v(z') \right|$$

$$\leq \sum_{\tau = m_1\tau_1 + m_2\tau_2 + \cdots + m_t\tau_t} M \underbrace{\left| f_v^{(m)}(g_v(z)) \prod_{i=1}^{t}\prod_{j=1}^{m_i} \partial_{\tau_i} g_v(z) - f_v^{(m)}(g_v(z')) \prod_{i=1}^{t}\prod_{j=1}^{m_i} \partial_{\tau_i} g_v(z) \right|}_{B_3}$$

$$+ M \underbrace{\left| f_v^{(m)}(g_v(z')) \prod_{i=1}^{t}\prod_{j=1}^{m_i} \partial_{\tau_i} g_v(z) - f_v^{(m)}(g_v(z')) \prod_{i=1}^{t}\prod_{j=1}^{m_i} \partial_{\tau_i} g_v(z') \right|}_{B_4}.$$

For ($B_3$), we have

$$\left| f_v^{(m)}(g_v(x,y)) \prod_{i=1}^{t} \partial_{\tau_i}(g_v(x,y))^{m_i} - f_v^{(m)}(g_v(x',y')) \prod_{i=1}^{t} \partial_{\tau_i}(g_v(x,y))^{m_i} \right|$$

$$= \prod_{i=1}^{t} \left| (\partial_{\tau_i} g_v(x,y))^{m_i} \right| \left| f_v^{(m)}(g_v(x,y)) - f_v^{(m)}(g_v(x',y')) \right|$$

$$\leq \left( \frac{(B_g |g_v(x,y)| + 2B_f)(k_p!)B_{k_p}(2B_g)^{k_p}}{(g'_{\min})^{k_p+1}} \right)^m \left| f_v^{(m)}(g_v(x,y)) - f_v^{(m)}(g_v(x',y')) \right|. \quad \text{(B.10)}$$

Since $|g_v(z) - g_v(z')| = \left| \frac{g(x')y - f(x)g(x') - g(x)y' + f(x')g(x)}{g(x)g(x')} \right|$, we obtain

$$|g_v(z) - g_v(z')|$$

$$= \left| \frac{g(x')y - g(x)y + f(x')g(x') - f(x)g(x') + g(x)y - g(x)y' + f(x')g(x) - f(x')g(x')}{g(x)g(x')} \right|$$

$$\leq \frac{|(g(x') - g(x))y| + |f(x') - f(x)||g(x')| + |g(x)||y' - y| + |f(x')||g(x) - g(x')|}{|g(x)g(x')|}$$

$$\leq \frac{B_g(1 + 2B_f + |y|) \|z - z'\|_2^{\alpha_p}}{(g'_{\min})^2}.$$

We have $|y| \leq B_y$, then $|g_v(z)| \leq \frac{B_y + B_f}{g_{\min}} := C_{fg}$, from equation B.10

$$\left| f_v^{(m)}(g_v(x,y)) \prod_{i=1}^{t} \partial_{\tau_i}(g_v(x,y))^{m_i} - f_v^{(m)}(g_v(x',y')) \prod_{i=1}^{t} \partial_{\tau_i}(g_v(x,y))^{m_i} \right|$$

$$\leq \left( \frac{((B_g C_{fg}) + 2B_f)(k_p!)B_{k_p}(2B_g)^{k_p}}{(g'_{\min})^{k_p+1}} \right)^m \left| f_v^{(m)}(g_v(x,y)) - f_v^{(m)}(g_v(x',y')) \right|$$

$$\leq \left( \frac{((B_g C_{fg}) + 2B_f)(k_p!)B_{k_p}(2B_g)^{k_p}}{(g'_{\min})^{k_p+1}} \right)^{k_p} \left| f_v^{(m+1)}(\zeta) \cdot (g_v(z) - g_v(z')) \right|$$

$$\leq \left( \frac{((B_g C_{fg}) + 2B_f)(k_p!)B_{k_p}(2B_g)^{k_p}}{(g'_{\min})^{k_p+1}} \right)^{k_p} \left( \frac{B_g(1 + 2B_f + B_y) \|z - z'\|^{\alpha_p}}{(g'_{\min})^2} \right) \left| f_v^{(m+1)}(\zeta) \right|,$$
$$\text{(B.11)}$$

where $\zeta \in [\min(g_v(z), g_v(z')), \max(g_v(z), g_v(z'))]$.

We know $f_v = \exp(-\frac{1}{2}x^2)$, its $(m+1)$-th derivative is bounded above and below, and we define $C_{k_p} = \max_{0 \leq m \leq k_p} \sup_{\zeta \in \mathbb{R}} \left| f_v^{(m+1)}(\zeta) \right|$.

Continue on equation B.11, we can conclude

$$\left| f_v^{(m)}(g_v(x,y)) \prod_{i=1}^{t} \partial_{\tau_i}(g_v(x,y))^{m_i} - f_v^{(m)}(g_v(x',y')) \prod_{i=1}^{t} \partial_{\tau_i}(g_v(x',y'))^{m_i} \right|$$

$$\leq \left( \frac{((B_g C_{fg}) + 2B_f)(k_p!)B_{k_p}(2B_g)^{k_p}}{(g'_{\min})^{k_p+1}} \right)^{k_p} \left( \frac{B_g(1 + 2B_f + B_y)C_{k_p}}{(g'_{\min})^2} \right) \|z - z'\|_2^{\alpha_p}$$

$$= C'_4(k_p, B_y, B_g, B_f, g'_{\min}) \|z - z'\|_2^{\alpha_p}.$$

where we define

$$C'_4(k_p, B_y, B_g, B_f, g'_{\min})$$

$$= \left( \frac{((B_g C_{fg}) + 2B_f)(k_p!)B_{k_p}(2B_g)^{k_p}}{(g'_{\min})^{k_p+1}} \right)^{k_p} \left( \frac{B_g(1 + 2B_f + B_y)C_{k_p}}{(g'_{\min})^2} \right).$$

For $(B_4)$, we have

$$\left| f_v^{(m)}(g_v(z')) \prod_{i=1}^{t} \prod_{j=1}^{m_i} \partial_{\tau_i} g_v(z) - f_v^{(m)}(g_v(z')) \prod_{i=1}^{t} \prod_{j=1}^{m_i} \partial_{\tau_i} g_v(z') \right|$$

$$= \left| f_v^{(m)}(g_v(z')) \left( \prod_{i=1}^{t} (\partial_{\tau_i} g_v(z))^{m_i} - \prod_{i=1}^{t} (\partial_{\tau_i} g_v(z'))^{m_i} \right) \right|$$

$$\leq C_{k_p} \left| \left( \prod_{i=1}^{t} (\partial_{\tau_i} g_v(z))^{m_i} - \prod_{i=1}^{t} (\partial_{\tau_i} g_v(z'))^{m_i} \right) \right|$$

$$\leq C_{k_p} \sum_{j=0}^{t-1} \left| \prod_{i=1}^{j} (\partial_{\tau_i} g_v(z'))^{m_i} \prod_{i=j+1}^{t} (\partial_{\tau_i} g_v(z))^{m_i} - \prod_{i=1}^{j+1} (\partial_{\tau_i} g_v(z'))^{m_i} \prod_{i=j+2}^{t} (\partial_{\tau_i} g_v(z))^{m_i} \right|$$

$$= C_{k_p} \sum_{j=0}^{t-1} \left| \prod_{i=1}^{j} (\partial_{\tau_i} g_v(z'))^{m_i} \prod_{i=j+2}^{t} (\partial_{\tau_i} g_v(z))^{m_i} \left( (\partial_{\tau_{j+1}} g_v(z'))^{m_{j+1}} - (\partial_{\tau_{j+1}} g_v(z))^{m_{j+1}} \right) \right|.$$

Again, we use mean value theorem, then

$$C_{k_p} \sum_{j=0}^{t-1} \left| \prod_{i=1}^{j} (\partial_{\tau_i} g_v(z'))^{m_i} \prod_{i=j+2}^{t} (\partial_{\tau_i} g_v(z))^{m_i} \left( (\partial_{\tau_{j+1}} g_v(z'))^{m_{j+1}} - (\partial_{\tau_{j+1}} g_v(z))^{m_{j+1}} \right) \right|$$

$$\leq C_{k_p} \sum_{j=0}^{t-1} \left| \prod_{i=1}^{j} (\partial_{\tau_i} g_v(z'))^{m_i} \prod_{i=j+2}^{t} (\partial_{\tau_i} g_v(z))^{m_i} (m_{j+1})(\xi'_j)^{m_{j+1}-1} \left( \partial_{\tau_{j+1}} g_v(z) - \partial_{\tau_{j+1}} g_v(z') \right) \right|$$

For $|\partial_{\tau_{j+1}} g_v(z) - \partial_{\tau_{j+1}} g_v(z')|$, we have

$$|\partial_{\tau_{j+1}} g_v(z) - \partial_{\tau_{j+1}} g_v(z')|$$

$$= \sqrt{2\pi} |\partial_{\tau_{j+1}}(u(x)(y - f(x))) - \partial_{\tau_{j+1}}(u(x')(y' - f(x')))|$$

$$\leq \sqrt{2\pi} |\partial_{\tau_{j+1}}(u(x)(y - f(x))) - \partial_{\tau_{j+1}}(u(x')(y - f(x)))|$$

$$+ \sqrt{2\pi} |\partial_{\tau_{j+1}}(u(x')(y - f(x))) - \partial_{\tau_{j+1}}(u(x')(y' - f(x')))|$$

$$\leq \sqrt{2\pi}(B_y + B_f)B_{k_p}(C_2 + C_3) \|z - z'\|^{\alpha_p} + \frac{(k_p!)B_{k_p}(B_g)^{k_p}}{(g'_{\min})^{k_p+1}} (B_f + 1) \|z - z'\|^{\alpha_p}$$

The second inequality holds due to equation B.10, equation B.11 and the fact

$$|y - f(x) - (y' - f(x'))| \leq (B_f + 1) \|z - z'\|^{\alpha_p}$$

Then for $(B_4)$, we have

$$B_4 \leq C'_5(k_p, B_y, B_g, B_f, g'_{\min}) \|z - z'\|^{\alpha_p}.$$

where $C'_5(k_p, B_y, B_g, B_f, g'_{\min})$ is defined as

$$C'_5(k_p, B_y, B_g, B_f, g'_{\min})$$

$$=k_pC_{k_p}\left(\frac{(B_y+B_f)(k_p!)^2(2B_g)^{k_p}}{(g'_{\min})^{k_p+1}}\right)^{k_p}$$

$$\cdot\left(\sqrt{2\pi}(B_y+B_f)B_{k_p}(C'_2+C'_3)+\frac{(k_p!)B_{k_p}(B_g)^{k_p}}{(g'_{\min})^{k_p+1}}(B_f+1)\right),$$

and $\xi'_j \in [\min(\partial_{\tau_{j+1}}g_v(z), \partial_{\tau_{j+1}}g_v(z')), \max(\partial_{\tau_{j+1}}g_v(z), \partial_{\tau_{j+1}}g_v(z'))]$.

Combining the result in $(B_1)$, $(B_2)$, $(B_3)$ and $(B_4)$, we have for

$$\left|\frac{\partial^{|\boldsymbol{\gamma}|}p(y|x)}{\partial x_1^{\gamma_1}\cdots\partial x_d^{\gamma_d}\partial y^{\gamma_{d+1}}}-\frac{\partial^{|\boldsymbol{\gamma}|}p(y'|x')}{\partial x_1^{\gamma_1}\cdots\partial x_d^{\gamma_d}\partial y^{\gamma_{d+1}}}\right|$$

$$=\sum_{\ell_1=0}^{\gamma_1}\cdots\sum_{\ell_d=0}^{\gamma_d}\binom{\gamma_1}{\ell_1}\cdots\binom{\gamma_d}{\ell_d}\left|D^{\mathbf{l}}u(x)D^{\mathbf{h}}v(x,y)-D^{\mathbf{l}}u(x')D^{\mathbf{h}}v(x',y')\right|$$

$$\leq\sum_{\ell_1=0}^{\gamma_1}\cdots\sum_{\ell_d=0}^{\gamma_d}\binom{\gamma_1}{\ell_1}\cdots\binom{\gamma_d}{\ell_d}\left|D^{\mathbf{l}}u(x)D^{\mathbf{h}}v(x,y)-D^{\mathbf{l}}u(x')D^{\mathbf{h}}v(x,y)\right|$$

$$+\left|D^{\mathbf{l}}u(x')D^{\mathbf{h}}v(x,y)-D^{\mathbf{l}}u(x')D^{\mathbf{h}}v(x',y')\right|$$

$$\leq\sum_{\ell_1=0}^{\gamma_1}\cdots\sum_{\ell_d=0}^{\gamma_d}\binom{\gamma_1}{\ell_1}\cdots\binom{\gamma_d}{\ell_d}\left(2k_pC(B_g,B_f,k_p,g'_{\min})B_{k_p}\right)B_{k_p}(C'_2+C'_3)||z-z'||^{\alpha_f}$$

$$+\frac{(k_p!)B_{k_p}(B_g)^{k_p}}{\sqrt{2\pi}(g'_{\min})^{k_p+1}}B_{k_p}(C'_4+C'_5)||z-z'||^{\alpha_f}$$

$$=\left(2k_pC(B_g,B_f,k_p,g'_{\min})B_{k_p}^2(C'_2+C'_3)+\frac{(k_p!)B_{k_p}^2(B_g)^{k_p}}{\sqrt{2\pi}(g'_{\min})^{k_p+1}}(C'_4+C'_5)\right)||z-z'||^{\alpha_f}.$$

We define

$$C_6(k_p, B_y, B_g, B_f, g'_{\min})$$

$$=2k_pC(B_g,B_f,k_p,g'_{\min})B_{k_p}^2(C'_2+C'_3)+\frac{(k_p!)B_{k_p}^2(B_g)^{k_p}}{\sqrt{2\pi}(g'_{\min})^{k_p+1}}(C'_4+C'_5),$$

so we can conclude

$$\left|\frac{\partial^{|\boldsymbol{\gamma}|}p(y|x)}{\partial x_1^{\gamma_1}\cdots\partial x_d^{\gamma_d}\partial y^{\gamma_{d+1}}}-\frac{\partial^{|\boldsymbol{\gamma}|}p(y'|x')}{\partial x_1^{\gamma_1}\cdots\partial x_d^{\gamma_d}\partial y^{\gamma_{d+1}}}\right|\leq C_6(k_p, B_y, B_g, B_f, g'_{\min})||z-z'||^{\alpha_f}$$

So $p(y|x) \in \mathcal{H}^{\beta_p}$, and $\|p(y|x)\|_{\mathcal{H}^{\beta_p}} \leq \sum_{|\gamma|\leq k_p}C_1 + \sum_{|\gamma|=k_p}C_6$, where the norm is a constant depending on $k_p, B_y, B_g, B_f, g'_{\min}$, then it depends on $\beta_f, \beta_g, B_y, B_g, B_f, g_{\min}$. To show that the conditional density $p(y|x)$ has a sub-Gaussian tail, we begin with its explicit form:

$$p(y|x) = \frac{1}{\sqrt{2\pi}g(x)}\exp\left(-\frac{1}{2g(x)^2}(y-f(x))^2\right).$$

Rewriting the exponent, we obtain:

$$p(y|x) = \frac{1}{\sqrt{2\pi}g(x)}\exp\left(-\frac{1}{2g(x)^2}\left(y^2-2f(x)y+f^2(x)\right)\right)$$

$$= \frac{1}{\sqrt{2\pi}g(x)}\exp\left(-\frac{1}{4g(x)^2}y^2\right)\exp\left(-\frac{1}{4g(x)^2}(y-2f(x))^2+\frac{1}{2g(x)^2}f^2(x)\right).$$

Using the bound $|f(x)| \leq B_f$, we get:

$$p(y|x) \leq \frac{1}{\sqrt{2\pi}g(x)}\exp\left(\frac{1}{2g(x)^2}B_f^2\right)\exp\left(-\frac{1}{4g(x)^2}y^2\right).$$

Since $g(x) \geq g_{\min}$ and $g(x) \leq B_g$, we obtain the uniform bound:

$$p(y|x) \leq \underbrace{\frac{1}{\sqrt{2\pi}g_{\min}} \exp\left(\frac{1}{2g_{\min}^2}B_f^2\right)}_{C_1} \exp\left(-\underbrace{\frac{1}{2B_g^2}}_{C_2} \cdot \frac{y^2}{2}\right) = C_1 \exp(-C_2 y^2/2).$$

where $C_1 = \frac{1}{\sqrt{2\pi}g_{\min}} \exp\left(\frac{1}{2g_{\min}^2}B_f^2\right)$, and $C_2 = \frac{1}{2B_g^2}$. $\qquad\square$

## C  Proof of Lemma 4.6

We start our proof by introducing two lemmas:

**Lemma C.1** (Truncate $y$). Under Assumption 4.2, for any $R > 1$, $x$ and $t > 0$, we have

$$\int_{|y|\geq R} p_t(y|x)\,\mathrm{d}y \lesssim R\exp(-C_2''R^2), \tag{C.1}$$

$$\int_{|y|\geq R} |\partial_y \log p_t(y|x)|^2 p_t(y|x)\,\mathrm{d}y \lesssim \frac{1}{\sigma_t^4}R^3\exp(-C_2''R^2). \tag{C.2}$$

where $C_2'' = \frac{C_2}{2\max(C_2,1)}$.

**Lemma C.2** (Truncate $p_t(y|x)$). Under Assumption 4.2, for any $R > 0$, $x$ and $\epsilon_{\text{low}} > 0$, we have

$$\int_{|y|\leq R} \mathbb{1}\{|p_t(y|x)| < \epsilon_{\text{low}}\} p_t(y|x)\,\mathrm{d}y \lesssim R\,\epsilon_{\text{low}}, \tag{C.3}$$

$$\int_{|y|\leq R} \mathbb{1}\{|p_t(y|x)| < \epsilon_{\text{low}}\} |\partial_y \log p_t(y|x)|^2 p_t(y|x)\,\mathrm{d}y \lesssim \frac{\epsilon_{\text{low}}}{\sigma_t^4}R^3. \tag{C.4}$$

These two lemmas are Lemma A.1 and Lemma A.2 in (Fu et al., 2024) when the response is a scaler.

For notational convenience, we replace $N$ with $N^{d+1}$ and accordingly redefine $C_\sigma$ and $C_\alpha$ as $(d+1)C_\sigma$ and $(d+1)C_\alpha$, respectively, so that the range $t \in [N^{-C_\sigma}, C_\alpha \log N]$ remains unchanged.

We now proceed to prove Proposition C.3, which extends Proposition A.3 from (Fu et al., 2024). Define $C_y = \sqrt{\frac{2\beta}{C_2''}}$, $R_y = C_y\sqrt{\log N}$, $s = \lfloor\beta\rfloor$, and $R_x = \sqrt{\frac{4\beta\log N}{C_{x_2}}}$.

**Proposition C.3.** (Conditional Score Approximation) Suppose Assumption 4.2 holds. We consider time $t \in [N^{-C_\sigma}, C_\alpha \log N]$ for constants $C_\sigma$ and $C_\alpha$. Given any integer $N > 0$, we constrain $(x, y) \in [-R_x, R_x]^d \times [-C_y\sqrt{\log N}, C_y\sqrt{\log N}]$. Then there exists a ReLU neural network class $\mathcal{F}(M_t, W, \kappa, L, K)$ which contains a mapping $s(x, y, t)$ satisfying

$$p_t(y|x)\|\nabla_y \log p_t(y|x) - s(x, y, t)\|_\infty \lesssim \frac{1}{\sigma_t^2}N^{-\beta}(\log N)^{\frac{s+2}{2}}, \quad \text{for any } t \in [N^{-C_\sigma}, C_\alpha \log N]. \tag{C.5}$$

Furthermore, the neural network hyperparameters satisfy

$$M_t = \mathcal{O}\left(\sqrt{\log N}/\sigma_t^2\right), \quad W = \mathcal{O}\left(N^{d+1}\log^7 N\right),$$
$$\kappa = \exp\left(\mathcal{O}(\log^4 N)\right), \quad L = \mathcal{O}(\log^4 N), \quad K = \mathcal{O}\left(N^{d+1}\log^9 N\right).$$

The proof of Proposition C.3 is provided in section C.1. Proposition C.3 is important in the proof of Lemma 4.6, it gives a uniformally approximation of score function over the domain $[-R_x, R_x]^d \times [-C_y\sqrt{\log N}, C_y\sqrt{\log N}]$ norm.

Given Lemmas C.1 and C.2 and Proposition C.3, we are ready to start our proof.

We can decompose the expectation of $L_2$ score approximation error as

$$\mathbb{E}_{x\sim P_x}\left[\mathbb{E}_{y\sim P_t(\cdot|x)}\left[|s(x, y, t) - \partial_y \log p_t(y|x)|^2\right]\right]$$

$$= \underbrace{\mathbb{E}_{x \sim P_x} \left[ \mathbb{E}_{y \sim P_t(\cdot|x)} \left[ \mathbf{1}_{\{|y| > R_y\}} |s(x,y,t) - \partial_y \log p_t(y|x)|^2 \right] \right]}_{C_1}$$

$$+ \underbrace{\mathbb{E}_{x \sim P_x} \left[ \mathbb{E}_{y \sim P_t(\cdot|x)} \left[ \mathbf{1}_{\{\|x\|_\infty > R_x, |y| \le R_y\}} |s(x,y,t) - \partial_y \log p_t(y|x)|^2 \right] \right]}_{C_2}$$

$$+ \underbrace{\mathbb{E}_{x \sim P_x} \left[ \mathbb{E}_{y \sim P_t(\cdot|x)} \left[ \mathbf{1}_{\{\|x\|_\infty \le R_x, |y| \le R_y\}} |s(x,y,t) - \partial_y \log p_t(y|x)|^2 \right] \right]}_{C_3}.$$

For these three terms, $(C_1)$ corresponds to the truncation error due to the unboundedness of $y$, while $(C_2)$ accounts for the truncation error arising from the unboundedness of $x$. The last term, $(C_3)$, captures the approximation error of $s(x,y,t)$. We will bound each of these terms one by one in the following.

**Upper Bound for $(C_1)$.** We can bound the truncation error $(C_1)$ as follows:

$$(C_1) \le \mathbb{E}_{x \sim P_x} \left[ \mathbb{E}_{y \sim P_t(\cdot|x)} \left[ \mathbf{1}_{\{|y| > R_y\}} \left( |s(x,y,t)| + |\partial_y \log p_t(y|x)| \right)^2 \right] \right]$$

$$\le 2 \int_{\mathbb{R}^d} \int_{|y| > R_y} |s(x,y,t)|^2 \, p_t(y|x) p(x) \, \mathrm{d}y \, \mathrm{d}x$$

$$+ 2 \int_{\mathbb{R}^d} \int_{|y| > R_y} |\partial_y \log p_t(y|x)|^2 \, p_t(y|x) p(x) \, \mathrm{d}y \, \mathrm{d}x$$

$$\overset{(i)}{\lesssim} 2 \left( \frac{\sqrt{\log N}}{\sigma_t^2} \right)^2 \int_{|y| > R_y} p_t(y|x) p(x) \, \mathrm{d}y \mathrm{d}x + \frac{2}{\sigma_t^4} \left( \frac{2\beta}{C_2'} \log N \right)^{3/2} N^{-2\beta}$$

$$\overset{(ii)}{=} \frac{2 \log N}{\sigma_t^4} \left( \frac{2\beta}{C_2'} \log N \right)^{1/2} N^{-2\beta} + \frac{2}{\sigma_t^4} \left( \frac{2\beta}{C_2'} \log N \right)^{3/2} N^{-2\beta}$$

$$\lesssim \frac{N^{-2\beta} \cdot (\log N)^{3/2}}{\sigma_t^4}.$$

In step $(i)$, we apply the uniform bound on the score function $|s(x,y,t)| \lesssim \frac{\sqrt{\log N}}{\sigma_t^2}$ from Proposition C.3, together with the tail integral bound equation C.2 in Lemma C.1, by taking $R = R_y$. In step $(ii)$, we make use of the tail probability estimate equation C.1 from Lemma C.1.

**Upper Bound for $(C_2)$.** For the term $(C_2)$, similar to $(C_1)$, we have

$$(C_2) \le 2 \int_{\|x\|_\infty > R_x} \int_{\mathbb{R}} \left( |s(x,y,t)|^2 + |\partial_y \log p_t(y|x)|^2 \right) p_t(y|x) p(x) \, \mathrm{d}y \, \mathrm{d}x$$

$$\le \underbrace{2 \int_{\|x\|_\infty > R_x} \int_{\mathbb{R}} |s(x,y,t)|^2 p_t(y|x) p(x) \, \mathrm{d}y \, \mathrm{d}x}_{C_4}$$

$$+ \underbrace{2 \int_{\|x\|_\infty > R_x} \int_{\mathbb{R}} |\partial_y \log p_t(y|x)|^2 p_t(y|x) p(x) \, \mathrm{d}y \, \mathrm{d}x}_{C_5}.$$

For the term $(C_4)$, we can bound it by leveraging the sub-Gaussian tail of $p(x)$. Specifically, we have

$$2 \int_{\|x\|_\infty > R_x} \int_{\mathbb{R}} |s(x,y,t)|^2 p_t(y|x) p(x) \, \mathrm{d}y \, \mathrm{d}x$$

$$\le 2 \left( \frac{\sqrt{\log N}}{\sigma_t^2} \right)^2 \int_{\|x\|_\infty > R_x} p(x) \, \mathrm{d}x$$

$$\le 2 \left( \frac{\sqrt{\log N}}{\sigma_t^2} \right)^2 \int_{\|x\| > R_x} C_{x_1} \exp \left( -C_{x_2} \|x\|^2 \right) \, \mathrm{d}x$$

$$\overset{(iii)}{\lesssim} \frac{\log N}{\sigma_t^4} \exp \left( -C_{x_2} R_x^2 / 2 \right) \tag{C.6}$$

In step $(iii)$, we apply the tail bounds for sub-gaussian distribution.

For the term $(C_5)$, we begin by introducing two lemmas:

**Lemma C.4.** (Lemma A.9 in Fu et al. (2024)) Under Lemma 4.3, there exists a constant $C_7$ such that the diffused conditional density $p_t(y|x)$ can be bounded as:

$$\frac{C_7}{\sigma_t} \exp\left(\frac{-y^2+1}{\sigma_t^2}\right) \le p_t(y|x) \le \frac{C_1}{\sqrt{\alpha_t^2 + C_2\sigma_t^2}} \exp\left(\frac{-C_2 y^2}{2(\alpha_t^2 + C_2\sigma_t^2)}\right).$$

**Lemma C.5.** (Lemma A.10 in Fu et al. (2024)) Under Lemma 4.3, there exists a constant $C_8$ such that the score function $\partial_y \log p_t(y|x)$ can be bounded as:

$$\|\partial_y \log p_t(y|x)\|_\infty \le \frac{C_8}{\sigma_t^2}(|y|+1).$$

According to Lemma C.5, we have for any $x \in \mathbb{R}^d$,

$$\mathbb{E}_{y \sim P_t(\cdot|x)}\left[\|\partial_y \log p_t(y|x)\|_2^2\right] \lesssim \mathbb{E}_{y \sim P_t(\cdot|x)}\left[\frac{(|y|+1)^2}{\sigma_t^4}\right].$$

For $\mathbb{E}_{y \sim P_t(\cdot|x)}\left[(|y|+1)^2\right]$, using Lemma C.4, we have

$$\mathbb{E}_{y \sim P_t(\cdot|x)}\left[(|y|+1)^2\right] = \int (|y|^2 + 2|y| + 1)p_t(y|x)\mathrm{d}y$$

$$\le \int (|y|^2 + 2|y| + 1)\frac{C_1}{\sqrt{\alpha_t^2 + C_2\sigma_t^2}} \exp\left(\frac{-C_2 y^2}{2(\alpha_t^2 + C_2\sigma_t^2)}\right) \mathrm{d}y$$

$$= \frac{\sqrt{2\pi}C_1(\alpha_t^2 + C_2\sigma_t^2)}{C_2\sqrt{C_2}} + \frac{2C_1\sqrt{\alpha_t^2 + C_2\sigma_t^2}}{C_2} + \frac{\sqrt{2\pi}C_1}{\sqrt{C_2}}.$$

So we have

$$\mathbb{E}_{y \sim P_t(\cdot|x)}\left[\|\partial_y \log p_t(y|x)\|_2^2\right] \lesssim \frac{1}{\sigma_t^4}.$$

Thus, we obtain that

$$2\int_{\|x\|_\infty > R_x} \int_{\mathbb{R}} |\partial_y \log p_t(y|x)|^2 p_t(y|x)p(x) \, \mathrm{d}y \, \mathrm{d}x$$

$$= \mathbb{E}_{x \sim P_x}\left[\mathbb{E}_{y \sim P_t(\cdot|x)}\left[\mathbf{1}\left\{\|x\|_\infty \ge R_x\right\}\|\partial_y \log p_t(y|x)\|^2\right]\right]$$

$$\lesssim \frac{1}{\sigma_t^4}\mathbb{E}_{x \sim P_x}\left[\mathbf{1}\left\{\|x\|_\infty \ge R_x\right\}\right]$$

$$\lesssim \frac{\exp\left(-C_{x_2} R_x^2/2\right)}{\sigma_t^4}. \tag{C.7}$$

The last inequality holds because of the sub-Gaussian tail of $p(x)$.

By combining equation C.6 and equation C.7, we can conclude

$$(C_2) \lesssim \frac{\log N \exp\left(-C_{x_2} R_x^2\right)}{\sigma_t^4} + \frac{\exp\left(-C_{x_2} R_x^2\right)}{\sigma_t^4}$$

$$\lesssim \frac{N^{-2\beta} \log N}{\sigma_t^4}.$$

**Upper Bound for** $(C_3)$. We first decompose $(C_3)$ by truncating the region where $p_t(y|x)$ is small:

$$(C_3) = \underbrace{\int_{\|x\|_\infty \le R_x} \int_{|y| \le R_y} \mathbf{1}\left\{p_t(y|x) < \epsilon_{\text{low}}\right\} |s(x,y,t) - \partial_y \log p_t(y|x)|^2 p_t(y|x)\, p(x)\, \mathrm{d}y\, \mathrm{d}x}_{(C_6)}$$

$$+ \underbrace{\int_{\|x\|_\infty \leq R_x} \int_{|y| \leq R_y} \mathbf{1}\left\{p_t(y|x) \geq \epsilon_{\text{low}}\right\} |s(x,y,t) - \partial_y \log p_t(y|x)|^2 \, p_t(y|x) \, p(x) \, \mathrm{d}y \, \mathrm{d}x}_{(C_7)}.$$

For the term $(C_6)$, this term corresponds to the truncation error arising from the region where $p_t(y|x)$ is smaller than $\epsilon_{\text{low}}$, using Lemma C.2,

$$\int_{\|x\|_\infty \leq R_x} \int_{|y| \leq R_y} \mathbf{1}\left\{p_t(y|x) < \epsilon_{\text{low}}\right\} |s(x,y,t) - \partial_y \log p_t(y|x)|^2 \, p_t(y|x) \, p(x) \, \mathrm{d}y \, \mathrm{d}x$$

$$\leq 2 \int_{\|x\|_\infty \leq R_x} \int_{|y| \leq R_y} \left( |s(x,y,t)|^2 + |\partial_y \log p_t(y|x)|^2 \right) p(x) \, \mathrm{d}y \, \mathrm{d}x$$

$$\leq 2 \int_{\|x\|_\infty \leq R_x} \int_{|y| \leq R_y} \left( \left( \frac{\sqrt{\log N}}{\sigma_t^2} \right)^2 + |\partial_y \log p_t(y|x)|^2 \right) p(x) \, \mathrm{d}y \, \mathrm{d}x$$

$$\lesssim \left( \frac{\sqrt{\log N}}{\sigma_t^2} \right)^2 R_y \epsilon_{\text{low}} + \frac{\epsilon_{\text{low}}}{\sigma_t^4} R_y^3$$

$$\lesssim \left( \frac{\sqrt{\log N}}{\sigma_t^2} \right)^2 R_y \epsilon_{\text{low}} + \frac{\epsilon_{\text{low}}}{\sigma_t^4} R_y^3$$

$$\lesssim \frac{\epsilon_{\text{low}} (\log N)^{\frac{3}{2}}}{\sigma_t^4}.$$

For the term $(C_7)$, this term is related to the approximation error of $s(x,y,t)$, using Proposition C.3, we have

$$(C_7)$$

$$= \int_{\|x\|_\infty \leq R_x} \int_{|y| \leq R_y} \mathbf{1}\left\{p_t(y|x) \geq \epsilon_{\text{low}}\right\} \left( \frac{|s(x,y,t) - \partial_y \log p_t(y|x)|^2 \, (p_t(y|x))^2}{p_t(y|x)} \right) p(x) \, \mathrm{d}y \, \mathrm{d}x$$

$$\leq \int_{\|x\|_\infty \leq R_x} \int_{|y| \leq R_y} \mathbf{1}\left\{p_t(y|x) \geq \epsilon_{\text{low}}\right\} \frac{B^2}{\sigma_t^4} N^{-2\beta} (\log N)^{s+2} \left( \frac{1}{p_t(y|x)} \right) p(x) \, \mathrm{d}y \, \mathrm{d}x$$

$$= \frac{B^2}{\sigma_t^4 \epsilon_{\text{low}}} N^{-2\beta} (\log N)^{s+2} \int_{\|x\|_\infty \leq R_x} \int_{|y| \leq R_y} \mathbf{1}\left\{p_t(y|x) \geq \epsilon_{\text{low}}\right\} \left( \frac{\epsilon_{\text{low}}}{p_t(y|x)} \right) p(x) \, \mathrm{d}y \, \mathrm{d}x$$

$$\leq \frac{B^2}{\sigma_t^4 \epsilon_{\text{low}}} N^{-2\beta} (\log N)^{s+2} \int_{\|x\|_\infty \leq R_x} \int_{|y| \leq R_y} \mathbf{1}\left\{p_t(y|x) \geq \epsilon_{\text{low}}\right\} p(x) \, \mathrm{d}y \, \mathrm{d}x$$

$$\lesssim \frac{B^2}{\sigma_t^4 \epsilon_{\text{low}}} N^{-2\beta} (\log N)^{s+2} R_y$$

$$\lesssim \frac{B^2}{\sigma_t^4 \epsilon_{\text{low}}} N^{-2\beta} (\log N)^{s+\frac{5}{2}}.$$

Then we can conclude,

$$(C_3) \lesssim \frac{\epsilon_{\text{low}} (\log N)^{\frac{3}{2}}}{\sigma_t^4} + \frac{B^2}{\sigma_t^4 \epsilon_{\text{low}}} N^{-2\beta} (\log N)^{s+\frac{5}{2}}.$$

Combining the bounds of $(C_1)$, $(C_2)$ and $(C_3)$ together, we have

$$\mathbb{E}_{x \sim P_x} \left[ \mathbb{E}_{y \sim P_t(\cdot|x)} \left[ |s(x,y,t) - \partial_y \log p_t(y|x)|^2 \right] \right]$$

$$\lesssim \frac{N^{-2\beta} (\log N)^{3/2}}{\sigma_t^4} + \frac{N^{-2\beta} \log N}{\sigma_t^4} + \frac{\epsilon_{\text{low}} (\log N)^{\frac{3}{2}}}{\sigma_t^4} + \frac{B^2}{\sigma_t^4 \epsilon_{\text{low}}} N^{-2\beta} (\log N)^{s+\frac{5}{2}}.$$

Substitute $\epsilon_{\text{low}} = C_3'' N^{-\beta} (\log N)^{\frac{1+s}{2}}$ into the display above, where $C_3''$ is a constant depending on the construction of the ReLU neural network in Proposition C.3, and then we can conclude:

$$\mathbb{E}_{x \sim P_x} \left[ \mathbb{E}_{y \sim P_t(\cdot|x)} \left[ |s(x,y,t) - \partial_y \log p_t(y|x)|^2 \right] \right] \lesssim \frac{B^2}{\sigma_t^4} N^{-\beta} (\log N)^{\frac{s}{2}+2}$$

$$\lesssim \frac{1}{\sigma_t^4} N^{-\beta} (\log N)^{\frac{\beta}{2}+2}.$$

Replacing $N$ by $N^{\frac{1}{d+1}}$, we complete the proof.

## C.1 PROOF OF PROPOSITION C.3

*Proof.* We follow the proof of Proposition A.3 in (Fu et al., 2024). We define

$$B = \|p(y|x)\|_{\mathcal{H}^\beta(\mathbb{R}^d \times (-C_y\sqrt{\log N}-1, C_y\sqrt{\log N}+1)^{d_y})}.$$

Following similar arguments as in Lemmas A.4 to A.7 of Fu et al. (2024), we can approximate both $p_t(y|x)$ and $\log p_t(y|x)$ over the domain $(x, y) \in [-R_x, R_x]^{d_x} \times [-C_y\sqrt{\log N}, C_y\sqrt{\log N}]^{d_y}$. After we approximate both $p_t(y|x)$ and $\log p_t(y|x)$, we can repeat the proof of Proposition A.3 in Fu et al. (2024) to extend the conditional score approximation to our target domain.

Due to the extension of the domain of $(x, y)$, certain details in the proofs must be adjusted. Also, note that in our setting, the label is $x$, whereas in Fu et al. (2024), the label is $y$. Therefore, to align with our notation, $x$ and $y$ should be exchanged in every step of the proof. For simplicity, when adjusting the proofs, we follow our own definitions.

**Changes in Lemma A.4** In step (i), The approximation of $p_t(y|x)$ still holds for any $x \in \mathbb{R}^{d_x}$ and $y \in \mathbb{R}^{d_y}$ by defining

$$f_2(x, y, t) = \int_{B_{y,N}} p(z|x) \frac{1}{\sigma_t^d (2\pi)^{d/2}} \exp\left(-\frac{\|\alpha_t z - y\|^2}{2\sigma_t^2}\right) dz,$$

where

$$B_{y,N} = \left[\frac{y - \sigma_t C(0,d)\sqrt{\beta \log N}}{\alpha_t}, \frac{y + \sigma_t C(0,d)\sqrt{\beta \log N}}{\alpha_t}\right]$$
$$\cap \left[-C(0,d)\sqrt{\beta \log N}, C(0,d)\sqrt{\beta \log N}\right].$$

Thus, we have

$$|f_2(x, y, t) - p_t(y|x)| \leq N^{-\beta}, \quad \text{for all } x \in \mathbb{R}^{d_x}, \ y \in \mathbb{R}^{d_y}.$$

In step (ii), for the approximation of $p(z|x)$, we redefine $f(x, y)$ as

$$f(x, y) = P\left(R_{xy}(y - 1/2) \mid R_{xy}(x - 1/2)\right), \quad \text{for } x \in [0,1]^{d_x}, \ y \in [0,1]^{d_y},$$

where $R_{xy} = \max(2L\sqrt{\log N}, 2R_x)$ and $R_x = \sqrt{\frac{4\beta \log N}{C_{x_2}}}$. Thus, $R_{xy} = \mathcal{O}(\sqrt{\log N})$.

Under Assumption 4.2, it can be verified that $f$ satisfies

$$\|f\|_{\mathcal{H}^\beta([0,1]^{d_x + d_y})} \leq B(R_{xy})^s,$$

and we can construct local polynomials $q(x, y)$ as in Lemma A.4, so that it still holds

$$|f(x, y) - q(x, y)| \leq \frac{B(R_{xy})^s (d_x + d_y)^s}{s! N^\beta}, \quad \forall x \in (0,1]^{d_x}, \ y \in [0,1]^{d_y}. \tag{C.8}$$

This inequality corresponds to (A.27) in Lemma A.4, by substituting $R$ with $R_{xy}$.

Next, we redefine $f_3$ as

$$f_3(x, y, t) = \frac{1}{\sigma_t^d (2\pi)^{d/2}} \int_{B_{y,N}} q\left(\frac{z}{R_{xy}} + \frac{1}{2}, \frac{x}{R_{xy}} + \frac{1}{2}\right) \exp\left(-\frac{\|\alpha_t z - y\|^2}{2\sigma_t^2}\right) dz.$$

Following similar arguments as in (A.28) and (A.29) of Lemma A.4, we can show that

$$|f_3(x, y, t) - f_2(x, y, t)| \leq \frac{B(R_{xy})^s}{N^\beta} \int_{\mathbb{R}^{d_x}} \frac{1}{\sigma_t^d (2\pi)^{d/2}} \exp\left(-\frac{\|\alpha_t z - y\|^2}{2\sigma_t^2}\right) dz$$

$$\lesssim \frac{BN^{-\beta}\log^{s/2} N}{\alpha_t^d},$$

and similarly,

$$|f_3(x,y,t) - f_2(x,y,t)| \lesssim \frac{BN^{-\beta}\log^{(s+d_y)/2} N}{\sigma_t^d}.$$

Taking the minimum of the two bounds, we obtain

$$|f_3(x,y,t) - f_2(x,y,t)| \leq B \min\left(\frac{1}{\sigma_t^d}, \frac{1}{\alpha_t^d}\right) N^{-\beta}\log^{(d_y+s)/2} N$$

$$\lesssim BN^{-\beta}\log^{(d_y+s)/2} N.$$

and $\log p_t(y|x)$ In step (iii), the approximation of $\exp\left(-\frac{\|y-\alpha_t z\|^2}{2\sigma_t^2}\right)$ remains the same, so we can redefine $f_1$ as

$$f_1(x,y,t)$$

$$= \frac{1}{\sigma_t^d(2\pi)^{d/2}} \int_{\mathbf{B},N} q\left(\frac{z}{R_{xy}} + \frac{1}{2}, \frac{x}{R_{xy}} + \frac{1}{2}\right) \sum_{k<p} \frac{1}{k!}\left(-\frac{|y_i - \alpha_t z_i|^2}{2\sigma_t^2}\right)^k \mathrm{d}z$$

$$= \sum_{\mathbf{w}\in[N]^{d_x}} \sum_{\mathbf{v}\in[N]^{d_y}} \sum_{\|\mathbf{n}\|_1 + \|\mathbf{n}'\|_1 < s} \frac{\mathbf{1}}{\mathbf{n}!\mathbf{n}'!} \frac{\partial^{n+n'} f}{\partial x^{\mathbf{n}}\partial y^{\mathbf{n}'}}\bigg|_{x=\frac{\mathbf{w}}{N}, y=\frac{\mathbf{v}}{N}} \left(\frac{x}{R_{xy}} + \frac{1}{2} - \frac{\mathbf{w}}{N}\right)^{\mathbf{n}'}$$

$$\times \prod_{j=1}^{d_y} \phi\left(3N\left(\frac{x_j}{R_{xy}} + \frac{1}{2} - \frac{w_j}{N}\right)\right) \prod_{i=1}^{d} \int \frac{\left(\frac{z_i}{R_{xy}} + \frac{1}{2} - \frac{v_i}{N}\right)^{n_i}}{\sigma_t(2\pi)^{1/2}} \sum_{k<p} \frac{1}{k!}\left(-\frac{|y_i - \alpha_t z_i|^2}{2\sigma_t^2}\right)^k \mathrm{d}z_i.$$

which is the approximation of $p_t(y|x)$, and it can still be represented as a linear combination of diffused local monomial $\Phi_{\mathbf{n},\mathbf{n}',\mathbf{v},\mathbf{w}}(x,y,t)$.

$$\Phi_{\mathbf{n},\mathbf{n}',\mathbf{v},\mathbf{w}}(x,y,t)$$

$$= \left(\frac{x}{R_{xy}} + \frac{1}{2} - \frac{\mathbf{w}}{N}\right)^{\mathbf{n}'} \prod_{j=1}^{d_x} \phi\left(3N\left(\frac{x_j}{R_{xy}} + \frac{1}{2} - \frac{w_j}{N}\right)\right) \prod_{i=1}^{d_y} \sum_{1\leq k<p} g(y_i, n_i, v_i, k),$$

where

$$g(y,n,v,k) = \frac{1}{\sigma_t(2\pi)^{1/2}} \int \left(\frac{z}{R_{xy}} + \frac{1}{2} - \frac{v}{N}\right)^n \frac{1}{k!}\left(-\frac{|y-\alpha_t z|^2}{2\sigma_t^2}\right)^k dz.$$

and

$$f_1(x,y,t) = \sum_{\mathbf{w}\in[N]^{d_x}, \mathbf{v}\in[N]^{d_y}} \sum_{\|\mathbf{n}\|_1 + \|\mathbf{n}'\|_1 \leq s} \frac{1}{\mathbf{n}!\mathbf{n}'!} \frac{\partial^{n+n'} f}{\partial x^{\mathbf{n}}\partial y^{\mathbf{n}'}}\bigg|_{x=\frac{\mathbf{w}}{N}, y=\frac{\mathbf{v}}{N}} \Phi_{\mathbf{n},\mathbf{n}',\mathbf{v},\mathbf{w}}(x,y,t).$$

We still have (A.34) in Lemma A.4 holds:

$$|p_t(y|x) - f_1(x,y,t)| \lesssim BN^{-\beta}\log^{\frac{d_y+s}{2}} N,$$

where $(x,y) \in [-R_x, R_x]^{d_x} \times [-C_y\sqrt{\log N}, C_y\sqrt{\log N}]^{d_y}$

Therefore, Lemma A.4 of Fu et al. (2024) has been successfully extended to the domain considered in our setting.

**Changes in Lemma A.5** Since $R_{xy} = \mathcal{O}(\sqrt{\log N})$, when constructing the ReLU neural network, the hyperparameters are still in the same order, so we just repeat the proof in Lemma A.5 to construct the similar ReLU neural network.

**Changes in Lemma A.6 and Lemma A.7**   The proof in Lemma A.6 and Lemma A.7 follows the proof in Lemma A.4 and Lemma A.5, so their changes remain the same.

After verifying these lemmas, we conclude the proof by setting $d_x = d$ and $d_y = 1$, from which the result in Proposition C.3 follows directly.

$\square$

## D   PROOF OF LEMMA 4.7

*Proof.* We follow the proof strategy of Theorem 4.1 from Fu et al. (2024).

We first revisit a few important definitions.

Recall the conditional score matching objective for a given score approximator $s$:

$$\mathcal{R}(s) = \frac{1}{T - t_0} \int_{t_0}^{T} \mathbb{E}_{x, y_t} \left\| s(x, y_t, t) - \partial_y \log p_t(y_t | x) \right\|_2^2 \, dt. \tag{D.1}$$

In Theorem 1 in Dasgupta et al. (2025), it is shown that equation D.1 differs from the following term only by a constant that depends on $s$

$$\mathcal{L}(s) = \frac{1}{T - t_0} \int_{t_0}^{T} \mathbb{E}_{x, y_0} \left[ \mathbb{E}_{y_t | y_0} \left[ \left\| s(x, y_t, t) - \partial_y \log p_t(y_t | x) \right\|_2^2 \right] \right] \, dt. \tag{D.2}$$

We now consider training the model using $n$ samples $\{(x_i, y_i)\}_{i=1}^{n}$ by minimizing the following empirical approximation of $\mathcal{L}(s)$:

$$\widehat{\mathcal{L}}(s) = \frac{1}{n} \sum_{i=1}^{n} \ell(x_i, y_i, s), \tag{D.3}$$

where the per-sample loss is defined as:

$$\ell(x, y, s) := \frac{1}{T - t_0} \int_{t_0}^{T} \mathbb{E}_{y_t | y} \left\| s(x, y_t, t) - \partial_y \log p_t(y_t \mid y) \right\|_2^2 \, dt. \tag{D.4}$$

Moreover, we define the loss function class $\mathcal{S}(R)$ as

$$\mathcal{S}(R) = \left\{ \ell(\cdot, \cdot; s) : \mathcal{D} \to \mathbb{R} \mid s \in \mathcal{F} \right\}.$$

First, note that Lemma D.1 in Fu et al. (2024) remains valid under our setting. We rewrite it for convenience.

**Lemma D.1.** (Lemma D.1 in Fu et al. (2024)) Suppose that we configure the network parameters $M_t, W, \kappa, L, K$ according to Lemma 4.6 and we denote $m_t = M_t / \sqrt{\log N}$. Then for any $s \in \mathcal{F}(M_t, W, \kappa, L, K)$ and $(x, y) \in \mathcal{D}$, we have $|\ell(s, x, y)| \lesssim \int_0^T m_t^2 dt \triangleq M$. In particular, if we take $t_0 = n^{-\mathcal{O}(1)}$ and $T = \mathcal{O}(\log n)$, we have $M = \mathcal{O}(\log t_0)$ for $m_t = \frac{1}{\sigma_t}$, and $M = \mathcal{O}\left(\frac{1}{t_0}\right)$ for $m_t = \frac{1}{\sigma_t^2}$, respectively.

Moreover, to convert our approximation guarantee to statistical theory, we need to calculate the covering number of the loss function class $S(R)$, which is defined as follows. As for Lemma D.3 in Fu et al. (2024), we restate it below with an additional assumption that bounds the feature vector $x$.

**Lemma D.2** (Extended Lemma D.3 in Fu et al. (2024)). Let $\delta > 0$. Suppose $\|x\|_\infty \le R$ and $|y| \le R$. Then, the $\delta$-covering number of the loss function class $\mathcal{S}(R)$ w.r.t. $\|\cdot\|_{L_\infty(\mathcal{D})}$ satisfies

$$\mathcal{N}\left( \delta, \mathcal{S}(R), \|\cdot\|_{L_\infty(\mathcal{D})} \right) \lesssim \left( \frac{2L^2 \left( W \max(R, T) + 2 \right) \kappa^L W^{L+1} \log N}{\delta} \right)^{2K}.$$

A key observation is that our definitions of $\mathcal{L}(s)$, $\widehat{\mathcal{L}}(s)$, and $\ell(x, y, s)$ coincide with those in Fu et al. (2024) when $\tau = \mathrm{id}$ with probability 1. Similarly, our definition of $\mathcal{R}(s)$ matches their definition of $\mathcal{R}_\star(s)$, namely $R_\star(s) = R(s)$

We define the truncated loss as

$$\ell^{\text{trunc}}(x, y, s) := \ell(x, y, s)\mathbf{1}\left\{\|x\|_\infty \le R, |y| \le R\right\}.$$

We repeat the decomposition of $\mathbb{E}_{\{x_i, y_i\}_{i=1}^n}[\mathcal{R}_\star(\widehat{s})]$.

$$\mathbb{E}_{\{x_i, y_i\}_{i=1}^n}[\mathcal{R}_\star(\widehat{s})] = \underbrace{\mathbb{E}_{\{x_i, y_i\}_{i=1}^n}\left[\mathbb{E}_{\{x_i', y_i'\}_{i=1}^n}\left[\mathcal{L}_2 - \mathcal{L}_2^{\text{trunc}}\right]\right]}_{A_2}$$

$$+ \underbrace{\mathbb{E}_{\{x_i, y_i\}_{i=1}^n}\left[\mathbb{E}_{\{x_i', y_i'\}_{i=1}^n}\left[\mathcal{L}_2^{\text{trunc}} - \mathcal{L}_1^{\text{trunc}}\right]\right]}_{B}$$

$$+ \underbrace{\mathbb{E}_{\{x_i, y_i\}_{i=1}^n}\left[\mathcal{L}_1^{\text{trunc}} - \mathcal{L}_1\right]}_{A_1} + \underbrace{\mathbb{E}_{\{x_i, y_i\}_{i=1}^n}\left[\mathcal{L}_1\right]}_{C}.$$

where

$$\mathcal{L}_1 = \frac{1}{n}\sum_{i=1}^n \left(\ell(x_i, y_i, \widehat{s}) - \ell(x_i, y_i, s^\star)\right), \quad \mathcal{L}_1^{\text{trunc}} = \frac{1}{n}\sum_{i=1}^n \left(\ell^{\text{trunc}}(x_i, y_i, \widehat{s}) - \ell^{\text{trunc}}(x_i, y_i, s^\star)\right),$$

and

$$\mathcal{L}_2 = \frac{1}{n}\sum_{i=1}^n \left(\ell(x_i', y_i', \widehat{s}) - \ell(x_i', y_i', s^\star)\right), \quad \mathcal{L}_2^{\text{trunc}} = \frac{1}{n}\sum_{i=1}^n \left(\ell^{\text{trunc}}(x_i', y_i', \widehat{s}) - \ell^{\text{trunc}}(x_i', y_i', s^\star)\right),$$

and $s^\star(x, y, t) = \partial_y \log p_t(y|x)$. $(x_i', y_i')_{i=1}^n$ are ghost samples from $P_{x,y}$.

**Upper bound of $A_1$ and $A_2$**  Define the truncated domain as

$$\mathcal{D}_R := \left\{(x, y) \in \mathbb{R}^{d+1} \mid \|x\|_\infty \le R, \ |y| \le R\right\}.$$

Since we have for any $s \in \mathcal{F}$, (s can depend on $x, y$)

$$\mathbb{E}_{x,y}\left[\left|\ell(x, y, s) - \ell^{\text{trunc}}(x, y, s)\right|\right]$$

$$= \int_{t_0}^T \frac{1}{T - t_0}\int_{R^{d+1}\backslash \mathcal{D}_R} \mathbb{E}_{y_t|y_0=y}\left[|s(x, y_t, t) - \partial_y \log \phi_t(y_t|y_0)|^2\right] p(y|x)p(x)dydxdt$$

$$\le 2\int_{t_0}^T \frac{1}{T - t_0}\int_x \int_{|y|>R} \mathbb{E}_{y_t|y_0=y}\left[|s(x, y_t, t)|^2 + |\partial_y \log \phi_t(y_t|y_0)|^2\right] p(y|x)p(x)dydxdt$$

$$+ \int_{t_0}^T \frac{1}{T - t_0}\int_{\|x\|_\infty>R} \int_y \mathbb{E}_{y_t|y_0=y}\left[|s(x, y_t, t)|^2 + |\partial_y \log \phi_t(y_t|y_0)|^2\right] p(y|x)p(x)dydxdt.$$

Using the subGaussian property of $p(y|x)$ and $p(x)$ in Lemma 4.3 and Assumption 4.4, and $\mathbb{E}_{y_t|y_0}[|\partial_y \log \phi_t(y_t|y_0)|^2] = \frac{1}{\sigma_t^2}$, we can conclude

$$\mathbb{E}_{x,y}\left[\left|\ell(x, y, s) - \ell^{\text{trunc}}(x, y, s)\right|\right]$$

$$\lesssim \int_{t_0}^T \frac{1}{\log N}\int_{|y|>R} \mathbb{E}_{x,y_t|y_0=y}\left[m_t^2 \log N + |\partial_y \log \phi_t(y_t|y_0)|^2\right] \exp\left(-C_2|y|^2/2\right) dxdt$$

$$+ \int_{t_0}^T \frac{1}{\log N}\int_{\|x\|_\infty>R} \mathbb{E}_{y,y_t|y_0=y}\left[m_t^2 \log N + |\partial_y \log \phi_t(y_t|y_0)|^2\right] \exp\left(-C_{x_2}\|x\|^2/2\right) dxdt$$

$$\lesssim \exp(-C_2 R^2)R\int_{t_0}^T m_t^2 dt + \exp(-C_2 R^2)\int_{t_0}^T \frac{1}{\sigma_t^2}dt$$

$$+ \exp(-C_{x_2}R^2)R\int_{t_0}^T m_t^2 dt + \exp(-C_{x_2}R^2)\int_{t_0}^T \frac{1}{\sigma_t^2}dt$$

$$\lesssim (\exp(-C_2 R^2) + \exp(-C_{x_2}R^2))RM.$$

Thus, term $A_1$ and $A_2$ are bounded by $\mathcal{O}((\exp(-C_2 R^2) + \exp(-C_{x_2}R^2))RM)$.

**Upper bound of $B$ and $C$**   For upper bounds of $B$ and $C$, we just repeat the proof of Theorem 4.1 in Fu et al. (2024), and finally we have

$$B \lesssim C + (\exp(-C_2 R^2) + \exp(-C_{x_2} R^2))RM + \frac{M}{n}\log\mathcal{N} + 7\delta,$$

and

$$C \leq \min_{s\in\mathcal{F}} \mathcal{R}_\star(s) = \min_{s\in\mathcal{F}} \mathcal{R}(s).$$

Combining these upper bounds together, we have

$$\mathbb{E}_{\{x_i,y_i\}_{i=1}^n}\left[\mathcal{R}(\widehat{s})\right] = \mathbb{E}_{\{x_i,y_i\}_{i=1}^n}\left[\mathcal{R}_\star(\widehat{s})\right]$$

$$\leq \min_{s\in\mathcal{F}} \frac{1}{T-t_0}\int_{t_0}^{T}\mathbb{E}_{x,y_t}\left\|s(x,y_t,t)-\partial_y\log p_t(y_t|x)\right\|_2^2\,dt$$

$$+\mathcal{O}\left(\frac{M}{n}N\log^9 N\left(\log^8 N+\log^2 N\log R+\log\frac{1}{\delta}\right)\right)$$

$$+\mathcal{O}\left((\exp(-C_2 R^2)+\exp(-C_{x_2}R^2))RM\right)+7\delta.$$

Using Lemma 4.6, and setting $R = \sqrt{\frac{(C_\sigma+2\beta)\log N}{\max(C_2,C_{x_2})(d+1)}}$, $\delta = N^{-\frac{2\beta}{d+1}}$, $N = n^{\frac{d+1}{d+1+\beta}}$, we can conclude

$$\mathbb{E}_{\{x_i,y_i\}_{i=1}^n}\left[\mathcal{R}(\widehat{s})\right] = \mathbb{E}_{\{x_i,y_i\}_{i=1}^n}\left[\mathcal{R}_\star(\widehat{s})\right]$$

$$\leq \mathcal{O}(MN^{-\frac{\beta}{d+1}}\log^{1+\frac{\beta}{2}}N)+\mathcal{O}\left(\frac{M}{n}N\log^{17}N\right)$$

$$+\mathcal{O}\left(MN^{-2\beta-C_\sigma}\log^{\frac{1}{2}}N\right)+\mathcal{O}(N^{-\frac{2\beta}{d+1}})$$

$$\lesssim \frac{1}{t_0}n^{-\frac{\beta}{d+\beta+1}}\log^{\max(17,1+\beta/2)}n.$$

Then we have

$$\mathbb{E}_{\mathcal{D}}\left[\mathcal{R}(\widehat{s})\right] = \mathcal{O}\left(\frac{1}{t_0}n^{-\frac{\beta}{d+\beta+1}}\log^{\max(17,1+\beta/2)}n\right)$$

$\square$

# E   PROOF OF LEMMA 4.8

*Proof.* We introduce a distribution shift item $\mathcal{T}(x_{\text{new}})$ defined as

$$\mathcal{T}(x_{\text{new}}) = \sup_{s\in\mathcal{F}}\frac{\sqrt{\mathbb{E}_{Y\sim P(\cdot|x_{\text{new}})}\left[\ell(x_{\text{new}},Y;s)\right]}}{\sqrt{\mathbb{E}_{X,Y}\left[\ell(X,Y;s)\right].}}$$

Given $\{x_i,y_i\}_{i=1}^n$ and $x_{\text{new}}$, we can generate an estimated conditional distribution $\widetilde{P}_{T-t_0}(\cdot|x_{\text{new}})$ using backward diffusion process.

We begin by proving that the KL divergence between $P(\cdot|x)$ and $\mathcal{N}(0,I)$ is bounded by a constant. In particular,

$$\mathrm{KL}\left(P(\cdot|x)\,\|\,\mathcal{N}(0,I)\right) = \frac{1}{2}\left(g(x)^2+f(x)^2-1-2\log g(x)\right) \leq \frac{1}{2}\left(B_g^2+B_f^2-1-2\log g_{\min}\right),$$

which is exactly bounded by a constant.

We repeat the proof of proposition 3 until Equation D.23 in Fu et al. (2024), obtaining that

$$\mathrm{TV}\left(P(\cdot|x_{\text{new}}),\widetilde{P}_{T-t_0}(\cdot|x_{\text{new}})\right)$$

$$\lesssim \sqrt{t_0}\log^{(d+1)/2}\frac{1}{t_0}+\exp(-T)$$

$$+ \sqrt{\int_{t_0}^{T} \frac{1}{2} \int p_t(y|x_{\text{new}}) \|\widehat{s}(y, x_{\text{new}}, t) - \partial_y \log p_t(y|x_{\text{new}})\|^2 dy dt}$$

$$\lesssim \sqrt{t_0} \log^{(d+1)/2} \frac{1}{t_0} + \exp(-T)$$

$$+ \frac{\sqrt{\int_{t_0}^{T} \mathbb{E}_{y_t} \left[ \|\widehat{s}(y, x_{\text{new}}, t) - \partial_y \log p_t(y_t|x_{\text{new}})\|^2 \right] dt}}{\sqrt{\int_{t_0}^{T} \mathbb{E}_{y_t,x} \left[ \|\widehat{s}(y_t, x, t) - \partial_y \log p_t(y_t|x)\|^2 \right] dt}} \cdot \sqrt{\frac{T}{2} \mathcal{R}(\widehat{s})}$$

$$\lesssim \sqrt{t_0} \log^{(d+1)/2} \frac{1}{t_0} + \exp(-T) + \mathcal{T}(x_{\text{new}}) \cdot \sqrt{\frac{T}{2} \mathcal{R}(\widehat{s})}.$$

where $\mathcal{R}(\widehat{s})$ defined by

$$\mathcal{R}(\widehat{s}) = \int_{t_0}^{T} \frac{1}{T - t_0} \mathbb{E}_{(y_t,x)} \|\widehat{s}(y_t, x, t) - \partial_y \log p_t(y_t|x)\|_2^2 dt.$$

Using Jensen's Inequality, we have:

$$\mathbb{E}_{\{x_i,y_i\}_{i=1}^{n}} \left[ \mathbf{TV} \left( P(\cdot|x_{\text{new}}), \widetilde{P}_{T-t_0}(\cdot|x_{\text{new}}) \right) \right]$$

$$\lesssim \sqrt{t_0} \log^{(d+1)/2} \frac{1}{t_0} + \exp(-T) + \mathcal{T}(x_{\text{new}}) \cdot \mathbb{E}_{\{x_i,y_i\}_{i=1}^{n}} \left[ \sqrt{\frac{T}{2} \mathcal{R}(\widehat{s})} \right]$$

$$\leq \sqrt{t_0} \log^{(d+1)/2} \frac{1}{t_0} + \exp(-T) + \mathcal{T}(x_{\text{new}}) \cdot \sqrt{\frac{T}{2} \mathbb{E}_{\{x_i,y_i\}_{i=1}^{n}} [\mathcal{R}(\widehat{s})]}$$

$$= \sqrt{t_0} \log^{(d+1)/2} \frac{1}{t_0} + \exp(-T) + \mathcal{T}(x_{\text{new}}) \cdot \sqrt{\frac{T}{2} \mathcal{O} \left( \frac{1}{t_0} \cdot n^{-\frac{\beta}{d+1+\beta}} (\log n)^{\max(17, 1+\beta/2)} \right)},$$

where the last inequality invokes Lemma 4.7.

Take early-stopping time $t_0 = n^{-\frac{\beta}{2(d+\beta+1)}}$, the terminal time $T = \frac{\beta}{4(d+\beta+1)} \log n$, so we have

$$\mathbb{E}_{\mathcal{D}} \left[ \mathbf{TV} \left( P(\cdot|x_{\text{new}}), \widetilde{P}_{T-t_0}(\cdot|x_{\text{new}}) \right) \right] = \mathcal{T}(x_{\text{new}}) \mathcal{O} \left( n^{-\frac{\beta}{4(d+\beta+1)}} (\log n)^{\max(9, 1+\frac{\beta}{4})} \right).$$

$\square$

## F    PROOF OF THEOREM 4.5

*Proof.* Let $q_{\frac{\alpha}{2}}$ and $q_{1-\frac{\alpha}{2}}$ denote the population $\frac{\alpha}{2}$- and $1-\frac{\alpha}{2}$-quantiles of the distribution $\widetilde{P}_{T-t_0}(\cdot|x)$, respectively, let $\widehat{F}$ be the CDF of $\widetilde{P}_{T-t_0}(\cdot|x)$, $F$ be the CDF of $P(\cdot|x_{\text{new}})$, and let $F_{n_1}$ be the empirical CDF based on $n_1$ i.i.d. samples from the generated distribution.

For $\mathbb{P}(Y \in [\widehat{q}_{\frac{\alpha}{2}}, \widehat{q}_{1-\frac{\alpha}{2}}])$, we have

$$\mathbb{P}(Y \in [\widehat{q}_{\frac{\alpha}{2}}, \widehat{q}_{1-\frac{\alpha}{2}}]) = F(\widehat{q}_{1-\frac{\alpha}{2}}) - F(\widehat{q}_{\frac{\alpha}{2}})$$

$$= \widehat{F}(\widehat{q}_{1-\frac{\alpha}{2}}) + F(\widehat{q}_{1-\frac{\alpha}{2}}) - \widehat{F}(\widehat{q}_{1-\frac{\alpha}{2}}) + \widehat{F}(\widehat{q}_{\frac{\alpha}{2}}) - F(\widehat{q}_{\frac{\alpha}{2}}) - \widehat{F}(\widehat{q}_{\frac{\alpha}{2}})$$

$$= F_{n_1}(\widehat{q}_{1-\frac{\alpha}{2}}) + \left( \widehat{F}(\widehat{q}_{1-\frac{\alpha}{2}}) - F_{n_1}(\widehat{q}_{1-\frac{\alpha}{2}}) \right) + \left( F(\widehat{q}_{1-\frac{\alpha}{2}}) - \widehat{F}(\widehat{q}_{1-\frac{\alpha}{2}}) \right)$$

$$+ \left( \widehat{F}(\widehat{q}_{\frac{\alpha}{2}}) - F(\widehat{q}_{\frac{\alpha}{2}}) \right) - F_{n_1}(\widehat{q}_{\frac{\alpha}{2}}) - \left( \widehat{F}(\widehat{q}_{\frac{\alpha}{2}}) - F_{n_1}(\widehat{q}_{\frac{\alpha}{2}}) \right)$$

$$= 1 - \alpha + \left( \widehat{F}(\widehat{q}_{1-\frac{\alpha}{2}}) - F_{n_1}(\widehat{q}_{1-\frac{\alpha}{2}}) \right) + \left( F(\widehat{q}_{1-\frac{\alpha}{2}}) - \widehat{F}(\widehat{q}_{1-\frac{\alpha}{2}}) \right)$$

$$+ \left( \widehat{F}(\widehat{q}_{\frac{\alpha}{2}}) - F(\widehat{q}_{\frac{\alpha}{2}}) \right) - \left( \widehat{F}(\widehat{q}_{\frac{\alpha}{2}}) - F_{n_1}(\widehat{q}_{\frac{\alpha}{2}}) \right).$$

We can decompose the expectation into two terms:

$$\mathbb{E}_{\mathcal{D}} \left[ \left| \mathbb{P}[Y \in [\widehat{q}_{\frac{\alpha}{2}}, \widehat{q}_{1-\frac{\alpha}{2}}]] - (1 - \alpha) \right| \right]$$

$$\leq \underbrace{\mathbb{E}_{\mathcal{D}} \left[ \left| \widehat{F}(\widehat{q}_{1-\frac{\alpha}{2}}) - F_{n_1}(\widehat{q}_{1-\frac{\alpha}{2}}) \right| \right]}_{E_1} + \underbrace{\mathbb{E}_{\mathcal{D}} \left[ \left| F(\widehat{q}_{1-\frac{\alpha}{2}}) - \widehat{F}(\widehat{q}_{1-\frac{\alpha}{2}}) \right| \right]}_{E_2}$$

$$+ \underbrace{\mathbb{E}_{\mathcal{D}} \left[ \left| \widehat{F}(\widehat{q}_{\frac{\alpha}{2}}) - F_{n_1}(\widehat{q}_{\frac{\alpha}{2}}) \right| \right]}_{E_3} + \underbrace{\mathbb{E}_{\mathcal{D}} \left[ \left| \widehat{F}(\widehat{q}_{\frac{\alpha}{2}}) - F(\widehat{q}_{\frac{\alpha}{2}}) \right| \right]}_{E_4}$$

**Upper Bound of $E_1$ and $E_3$**   To quantify the concentration of the empirical quantile estimates, we invoke the Dvoretzky–Kiefer–Wolfowitz (DKW) inequality, a fundamental result in empirical process theory.

**Lemma F.1** (Dvoretzky–Kiefer–Wolfowitz (DKW) Inequality)**.** Let $F$ be the cumulative distribution function (CDF) of a real-valued random variable, and let $F_n$ be the empirical CDF based on $n$ i.i.d. samples from $F$. Then, for any $\delta > 0$,

$$\mathbb{P} \left( \sup_{x \in \mathbb{R}} |F_n(x) - F(x)| > \delta \right) \leq 2 \exp(-2n\delta^2).$$

From Lemma F.1, we can conclude, for any $\epsilon > 0$,

$$\mathbb{P} \left( \left| F_{n_1}(\widehat{q}_{\frac{\alpha}{2}}) - \widehat{F}(\widehat{q}_{\frac{\alpha}{2}}) \right| > \frac{\epsilon}{2} \right) \leq 2 \exp(-n_1 \epsilon^2 / 2)$$

$$\iff \mathbb{P} \left( \left| \widehat{F}(q_{\frac{\alpha}{2}}) - \widehat{F}(\widehat{q}_{\frac{\alpha}{2}}) \right| > \frac{\epsilon}{2} \right) \leq 2 \exp(-n_1 \epsilon^2 / 2),$$

$$\text{and} \quad \mathbb{P} \left( \left| F_{n_1}(\widehat{q}_{1-\frac{\alpha}{2}}) - \widehat{F}(\widehat{q}_{1-\frac{\alpha}{2}}) \right| > \frac{\epsilon}{2} \right) \leq 2 \exp(-n_1 \epsilon^2 / 2)$$

$$\iff \mathbb{P} \left( \left| \widehat{F}(q_{1-\frac{\alpha}{2}}) - \widehat{F}(\widehat{q}_{1-\frac{\alpha}{2}}) \right| > \frac{\epsilon}{2} \right) \leq 2 \exp(-n_1 \epsilon^2 / 2),$$

which imply

$$\mathbb{P} \left( \left| \widehat{F}(q_{\frac{\alpha}{2}}) - \widehat{F}(\widehat{q}_{\frac{\alpha}{2}}) \right| \leq \frac{\epsilon}{2} \right) \geq 1 - 2 \exp(-n_1 \epsilon^2 / 2), \tag{F.1}$$

and

$$\mathbb{P} \left( \left| \widehat{F}(q_{\frac{\alpha}{2}}) - \widehat{F}(\widehat{q}_{\frac{\alpha}{2}}) \right| \leq \frac{\epsilon}{2} \right) \geq 1 - 2 \exp(-n_1 \epsilon^2 / 2). \tag{F.2}$$

We consider the event $E_{\text{joint}}$ defined as the simultaneous occurrence of two conditions:

$$E_{\text{joint}} = \left\{ \left| \widehat{F}(q_{\frac{\alpha}{2}}) - \widehat{F}(\widehat{q}_{\frac{\alpha}{2}}) \right| \leq \frac{\epsilon}{2} \quad \text{and} \quad \left| \widehat{F}(q_{1-\frac{\alpha}{2}}) - \widehat{F}(\widehat{q}_{1-\frac{\alpha}{2}}) \right| \leq \frac{\epsilon}{2} \right\}.$$

Let $E_L$ be the event $\left| \widehat{F}(q_{\frac{\alpha}{2}}) - \widehat{F}(\widehat{q}_{\frac{\alpha}{2}}) \right| \leq \frac{\epsilon}{2}$, and $E_U$ be the event $\left| \widehat{F}(q_{1-\frac{\alpha}{2}}) - \widehat{F}(\widehat{q}_{1-\frac{\alpha}{2}}) \right| \leq \frac{\epsilon}{2}$.

Thus, $E_{\text{joint}} = E_L \cap E_U$, and from equation F.1 and equation F.2,

$$\mathbb{P}(E_{\text{joint}}) = \mathbb{P}(E_L) + \mathbb{P}(E_U) - \mathbb{P}(E_L \cup E_U)$$
$$\geq \mathbb{P}(E_L) + \mathbb{P}(E_U) - 1$$
$$\geq 1 - 4 \exp(-n_1 \epsilon^2 / 2). \tag{F.3}$$

Under the events $E_{\text{joint}}$, we can easily conclude:

$$E_1 \leq \frac{\epsilon}{2}, \quad E_3 \leq \frac{\epsilon}{2}. \tag{F.4}$$

**Upper Bound of $E_2$ and $E_4$**   From the definition of total variation distance, we have

$$|F(x) - \widehat{F}(x)| \leq \text{TV} \left( \widetilde{P}_{T-t_0}(\cdot | x_{\text{new}}), P(\cdot | x_{\text{new}}) \right)$$

From Lemma 4.8, we have

$$E_2 \leq \mathcal{T}(x_{\text{new}}) \cdot \mathcal{O}\left(n^{-\frac{\beta}{4(d+\beta+1)}} (\log n)^{\max\left(9, 1+\frac{\beta}{4}\right)}\right), \tag{F.5}$$

$$E_4 \leq \mathcal{T}(x_{\text{new}}) \cdot \mathcal{O}\left(n^{-\frac{\beta}{4(d+\beta+1)}} (\log n)^{\max\left(9, 1+\frac{\beta}{4}\right)}\right). \tag{F.6}$$

Combining equation F.4 and equation F.6, we have

$$\mathbb{E}_{\mathcal{D}}\left[\left|\mathbb{P}[Y \in [\widehat{q}_{\frac{\alpha}{2}}, \widehat{q}_{1-\frac{\alpha}{2}}]] - (1-\alpha)\right|\right] \leq \epsilon + \mathcal{T}(x_{\text{new}}) \cdot \mathcal{O}\left(n^{-\frac{\beta}{4(d+\beta+1)}} (\log n)^{\max\left(9, 1+\frac{\beta}{4}\right)}\right),$$

with the probability $\mathbb{P}(E_{\text{joint}}) \geq 1 - 4\exp(-n_1\epsilon^2/2)$.

Let $\delta = 1 - \mathbb{P}(E_{\text{joint}})$, so $\delta \leq 4\exp(-n_1\epsilon^2/2)$, then

$$\frac{n_1\epsilon^2}{2} \leq \log\frac{4}{\delta} \Rightarrow \epsilon \leq \sqrt{\frac{2}{n_1}\log\frac{4}{\delta}}.$$

Then we can conclude

$$\mathbb{E}_{\mathcal{D}}\left[\left|\mathbb{P}[Y \in [\widehat{q}_{\frac{\alpha}{2}}, \widehat{q}_{1-\frac{\alpha}{2}}]] - (1-\alpha)\right|\right]$$
$$\leq \sqrt{\frac{2}{n_1}\log\frac{4}{\delta}} + \mathcal{T}(x_{\text{new}}) \cdot \mathcal{O}\left(n^{-\frac{\beta}{4(d+\beta+1)}} (\log n)^{\max\left(9, 1+\frac{\beta}{4}\right)}\right),$$

with the probability $1 - \delta$. $\qquad\square$

## G  TRAINING DETAILS

### G.1  EXPERIMENTAL SETUP

To assess our conditional diffusion-based uncertainty quantification method and compare against state-of-the-art approaches, we evaluate on ten continuous-response regression tasks drawn from the UCI Machine Learning Repository (Dua & Graff, 2019). These benchmarks span have seen extensive use in the literature on predictive uncertainty (Gal & Ghahramani, 2016a; Lakshminarayanan et al., 2017a; Dewolf et al., 2023).

In Table 3, we report the number of examples ($n$) and the feature dimension ($d$) for each task after null columns are removed.

Table 3: Sample size ($n$) and feature dimension ($d$) for each regression benchmark.

|   | Housing | Concrete | Energy | Yacht | Kin8nm | Naval | Power | Wine | Protein | Year |
|---|---------|----------|--------|-------|--------|-------|-------|------|---------|------|
| $n$ | 506 | 1030 | 768 | 308 | 8192 | 11934 | 9568 | 1599 | 45730 | 515345 |
| $d$ | 13 | 8 | 8 | 6 | 9 | 16 | 4 | 11 | 9 | 90 |

### G.2  IMPLEMENTATION DETAILS

**Data Processing**  Following *High Quality Prediction Intervals for Deep Learning* [Pearce et al., 2018] and Salem et al. (2020), we evaluate each method over 20 randomized splits (i.e., 20 trials) per dataset. The Protein dataset is trained and evaluated on five randomized splits, and the Year Dataset is trained and evaluated on one split, also following the approach in Salem et al. (2020). We train all models with the Adam optimizer following the setup in Salem et al. (2020) but share the same learning rates specified in Section G.2.2. To ensure consistency across trials, all datasets are standardized to zero mean and unit standard deviation based on the training set.

When implementing conformal methods, we reserve 10% of the training data for conformal calibration, and train predictive models on the remaining percentage. For our diffusion-based method, a 5% of the training data is held out for validation, while the remaining 75% is used for training. Prior to splitting, we removed any covariates with zero values, which eliminated several features each from the POWER and NAVAL datasets. We apply Z-score normalization to both inputs and targets, computing the mean and standard deviation on the training set only and then using those statistics to transform the calibration and test sets as well, thereby avoiding data leakage, following the processing pipeline of Salem et al. (2020). As Han et al. (2022) does, we multiply the response variable by 100 for the kin8nm and naval dataset to ensure that the values form a correct comparison.

**Model Architecture** Our Conditional Diffusion Model (CDM) extends the DDPM framework of Ho et al. (2020a) to the conditional setting by treating each scalar response $y$ as a one-dimensional signal driven by covariates $x \in \mathbb{R}^d$. The score network is the same two-layer MLP (hidden size 64, SiLU activations) with two residual blocks, each followed by dropout (rate 0.1). We adopt a 32-dimensional sinusoidal time embedding for $t$ (Vaswani et al., 2017). We implement the backward diffusion process via the `DDPMScheduler` in HuggingFace's `diffusers` library, instantiating it with the "squaredcos_cap_v2" beta schedule of Nichol & Dhariwal (2021) over 1000 timesteps. For inference, 1000 samples were generated through the DDPM reverse sampler and a corresponding 95% prediction interval ($\alpha = 5\%$) was constructed. The point estimate for this model was the mean of the samples.

### G.2.1 BASELINE METHODS

To evaluate our method, we compare it against several representative baseline algorithms for uncertainty quantification in regression. These include Split-Conformal Prediction (Split-CP), Conformal Quantile Regression (CQR), Monte Carlo Dropout (MC Dropout), and Deep Ensembles (DE). Each baseline is implemented following its original paper and used to construct prediction intervals for test inputs. We use the abbreviation Split-CP, CQR, MC Dropout, and DE throughout this section.

**Split-Conformal Prediction (Split-CP) (Papadopoulos et al., 2002; 2007)** We first learn a regression model $\widehat{f}(x)$ on the training data, then compute nonconformity scores on the independent calibration split:

$$s_i = \big|y_i - \widehat{f}(x_i)\big|.$$

The constructed $100(1-\alpha)\%$ prediction interval given $x_{\text{new}}$ is

$$C_{\text{CP}}(x_{\text{new}}) = \big[\widehat{f}(x_{\text{new}}) - q_{1-\alpha}, \widehat{f}(x_{\text{new}}) + q_{1-\alpha}\big],$$

where $q_{1-\alpha}$ is the empirical $(1-\alpha)$-quantile of $\{s_i\}$. The regression model output serves as the point estimate. In our implementation, Split-CP uses a two-layer MLP with 64 hidden units as its neural network, sharing the same architecture as CDM for evaluating baselines.

**Conformal Quantile Regression (CQR) (Romano et al., 2019)** CQR builds upon Split-Conformal by training two quantile regressors $\ell_\tau(x)$ and $u_\tau(x)$ for lower and upper quantiles. On the calibration split, we evaluate $s_i = \max\big\{\ell_\tau(x_i) - y_i, \ y_i - u_\tau(x_i)\big\}$ and let $q_{\text{CQR}}$ be its $(1-\alpha)$-quantile. The prediction interval becomes

$$C_{\text{CQR}}(x_{\text{new}}) = \big[\ell_\tau(x_{\text{new}}) - q_{\text{CQR}}, u_\tau(x_{\text{new}}) + q_{\text{CQR}}\big].$$

The CQR point estimate is the mid-point between the calibrated lower and upper quantile nets $\ell_\tau(x_{\text{new}})$ and $u_\tau(x_{\text{new}})$. In our implementation, CQR uses a two-layer MLP with 64 hidden units, sharing the same architecture as CDM for evaluating baselines.

**Monte Carlo Dropout (MC Dropout)** We follow the implementation in Gal & Ghahramani (2016b). During both training and inference, we apply dropout with probability $p$ at every layer. At test time we perform $N_{\text{MC}}$ stochastic forward passes with dropout (i.e. random weight masks $w$), yielding predictions

$$y^{(b)} = f_{w^{(b)}}(x_{\text{new}}), \quad b = 1, \dots, N_{\text{MC}}.$$

We then inject the analytic precision noise by sampling $\widehat{y}^{(b)} \sim \mathcal{N}\big(y^{(b)}, \tau^{-1}\big)$, where $\tau$ is the model precision hyperparameter in MC-Dropout. Finally, we calculate

$$\widehat{\mu}(x_{\text{new}}) = \tfrac{1}{N_{\text{MC}}} \sum_{t=1}^{N_{\text{MC}}} \widehat{y}^{(b)}, \quad C_{\text{MC-Dropout}}(x_{\text{new}}) = \big[q_{\alpha/2}\{\widehat{y}^{(b)}\}, q_{1-\alpha/2}\{\widehat{y}^{(b)}\}\big],$$

where $q$ extracts the quantile. In our experiments, we set $N_{MC}$ to 10000 samples to produce predictive intervals. Because predictive interval accuracy is dominated by score network quality, increasing samples beyond 1000 yields diminishing returns since it takes more time. For the Year Dataset, we used 1000 samples due to its size. However, using too few samples degrades the performance because the estimation of the quantiles becomes unreliable. Similarly, the uncertainty estimate for the MC dropout improves mainly by increasing the number of stochastic forward passes. We also report prediction interval coverage probability (PICP) values in Table 2, following the benchmark protocol in Han et al. (2022) to ensure direct comparability across methods.

**Deep Ensemble (DE)**   We follow the implementation in Lakshminarayanan et al. (2017c). We train 5 independently initialized networks on the same data; each network is augmented with a separate variance head (mean and log variance outputs) as in Kendall & Gal (2017b). At inference, we collect $M = 1000$ predictions, $\{(\mu_m(x_{\text{new}}), \sigma_m^2(x_{\text{new}}))\}_{m=1}^M$ and sample $y_m \sim \mathcal{N}(\mu_m, \sigma_m^2)$ to form a prediction interval based on quantiles.

### G.2.2   TRAINING

To eliminate capacity-driven bias across methods, all models share an identical network architecture and are optimized with the same hyper-parameters. Batch size is chosen solely as a function of dataset cardinality: 64 for datasets with $n < 10^3$ examples; 128 for $10^3 \leq n < 10^4$; and 1024 for $n \geq 10^4$. Each CDM was trained with an AdamW optimizer with a learning rate of $1e - 4$, except for the Blog and Energy datasets which used a learning rate of $3e - 4$.

Because smaller tasks are more prone to overfitting, in the CDM we apply a dropout rate of $0.20$ to the HOUSING, ENERGY, and CONCRETE datasets. Conversely, the largest benchmarks—YEAR, PROTEIN, and NAVAL—use a milder rate of $0.05$. All remaining datasets employ a default dropout of $0.1$. Training proceeds for at most 500 epochs with early stopping (patience $= 50$), except for the two smallest tasks—YACHT and HOUSING—which are allowed up to 1000 epochs to ensure convergence. For all baseline methods, the learning rate ($5e - 4$), drop out rate ($0.1$) are kept constant, and the number of epochs follow the same procedure as the CDM. In MC Dropout, we vary the model precision $\tau$ and the dropout rate from 0.05 to 0.25, using grid search to determine the correct parameters best parameters from the best RMSE from each split's validation set. For DEs we vary only the dropout rate in the same increments as we do in MC Dropout in each dataset.

### G.3   FURTHER EXPERIMENTS: BLOG DATASET

Apart from the ten standard UCI regression datasets from Dua & Graff (2019), we further evaluate our methods on the high-dimensional, heavy-tailed BLOG dataset, which contains 60,021 samples with 279 features. Following Dewolf et al. (2021), we apply a log-transformation to the target variable prior to training to mitigate extreme skewness. After this transformation, the Blog dataset still exhibits a relatively heavy-tailed distribution on its features: the estimated excess Kurtosis is 3.29 (Larger values indicate heavier tails and Gaussian distribution has zero excess Kurtosis). The results for the BLOG dataset are based on our own experiments, conducted with the same parameter settings described above, while adhering to the training setup outlined in Appendix G.2.

Table 4: Performance on the BLOG dataset (mean $\pm$ s.e.).

| Method | RMSE | NLL | PICP |
|---|---|---|---|
| Split-CP | 0.85±0.02 | 1.28±0.02 | 95.44±0.85 |
| CQR | 0.78±0.03 | **1.01±0.14** | 95.62±0.92 |
| CDM | **0.74±0.24** | 1.45±0.47 | 94.20±0.90 |
| Deep Ensemble | 0.83±0.01 | 4.48±0.63 | 94.16±0.89 |
| MC Dropout | 0.77±0.03 | 5.03±0.87 | **95.71±0.97** |

The results, reported in Table 4, demonstrate that our approach remains robust even under such high-dimensional and heavy-tailed conditions. The presence of these characteristics results in the relatively unstable performance. As stated in Section 5, in these situations where features are sparse given the distribution, NLL may reflect model–distribution mismatch rather than purely predictive quality, which helps explain the instability (standard error) of NLL across certain benchmarks, such as here with BLOG or with YEAR. Furthermore, the conditional distribution $p(y|x)$ may be harder to approximate with limited data, with data sparsity in high dimensions (Han et al., 2022). CDM achieves competitive performance compared to the other methods in terms of RMSE and NLL while also providing calibrated uncertainty estimates (PICP). We note that the heavy-tailed distribution is beyond our theoretical assumptions, suggesting the diffusion-based method can be applied to broad applications due to their strong modeling power.

## G.4 Computational Resources

Table 5: Total time (s) reported for each dataset and method.

| Dataset | CDM | DE | MC Dropout |
|---------|-----|-----|-----------|
| Housing | 12.39±0.05 | **11.74±0.22** | 50.93±0.72 |
| Concrete | 62.21±0.36 | **19.8±1.16** | 120.24±5.03 |
| Energy | **11.06±0.07** | 13.1±0.39 | 69.03±1.50 |
| Kin8nm | 234.65±1.73 | **161.55±0.45** | 1040.30±191.75 |
| Naval | 270.42±1.52 | **137.3±0.59** | 1607.60±122.14 |
| Power | 284.15±1.47 | **123.35±0.63** | 278.58+ 3.90 |
| Protein | 280.47±0.57 | **277.44±0.39** | 9059.40 ±44.40 |
| Wine | **94.76±0.41** | 106.9±1.81 | 193.70±16.56 |
| Yacht | 19.04±0.05 | **7.01±0.19** | 36.78±0.58 |
| Year | **1570.72±0.00** | 1682.25±0.00 | 10333.37±0.00 |
| Blog | **276.39±2.73** | 317.22±1.54 | 297.68±26.39 |

This section reports the CDM's total running times (training time plus inference time) on ten UCI-style datasets. All runs used PyTorch 2.8.0+cu126 on a single NVIDIA A100 GPU. We utilize the same CDM architecture described in the training section. For training, we record the wall steps per second, the end-to-end step rate measured with time.perf_counter, including data loading, host to device copies, optimizer updates, logging, and synchronization. With this setting, training averaged ∼180–200 steps/s across datasets; after a brief warm-up the rate was stable over epochs.

We report the total combined training and inference times in Table 5, focusing our comparison against Deep Ensembles (DE) and MC Dropout because these methods are widely adopted baselines and are also known to be computationally demanding. CDM and DE show comparable total times for running each dataset across the board. Note that we did not optimize for sampling, as we used DDPM, so the total cost of the model could be reduced with faster diffusion model sampling methods. In contrast, Deep Ensembles require training multiple models—resulting in higher overall time even though each inference call is relatively cheap. For instance, on the Year dataset, CDM is slightly faster than DE, reflecting the cumulative cost of training multiple ensemble members when dataset size scales. MC Dropout, on the other hand, is consistently the most expensive approach: because it relies on performing many stochastic forward passes per test point, its inference cost grows prohibitively large, as seen on the Protein dataset where MC Dropout requires 9059s compared to just 280s for CDM. CDM therefore demonstrates a clear efficiency advantage, achieving strong uncertainty estimates with significantly reduced computational overhead.

## G.5 Toy examples: predictive intervals and distribution shift coefficients

Figures 3 and 4 show CDM predictive intervals and true 95% intervals on two one-dimensional regression tasks. In both panels we mark an in-domain point $x_{\text{in}}$ and an OOD point $x_{\text{ood}}$, together with their estimated distribution shift coefficients $\mathcal{T}(x_{\text{in}})$ and $\mathcal{T}(x_{\text{ood}})$. For both examples we obtain $\mathcal{T}(x_{\text{in}}) \approx 1$, indicating that the CDM behaves similarly to its average in-distribution risk and that the coverage guarantee holds very well. In contrast, $\mathcal{T}(x_{\text{ood}}) \gg 1$ signals a substantial degradation of coverage at the OOD point, so the coverage guarantee can fail badly there.

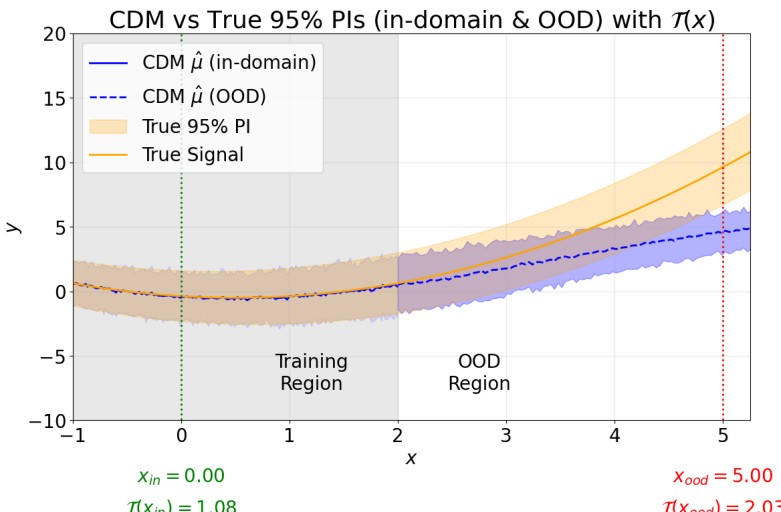

Figure 3: CDM predictive intervals and true 95% intervals on the first toy regression task.

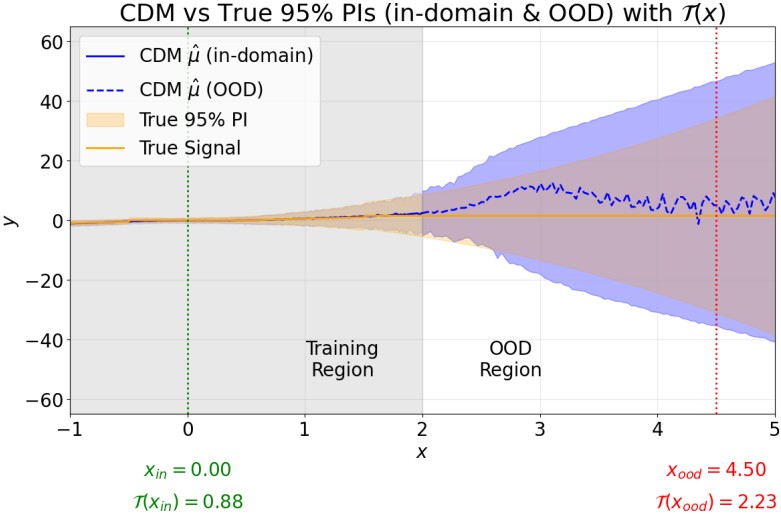

Figure 4: CDM predictive intervals and true 95% intervals on the second toy regression task.

Crucially, the two panels exhibit different qualitative patterns. Figure 3 shows that the CDM predictive band remains relatively narrow even far OOD, and the main error arises from a biased predictive mean, thus the increased risk is not obvious from visual inspection of the interval width alone. Meanwhile, Figure 4 illustrates the opposite behavior: the CDM intervals widen much more than the true 95% interval, overestimating the predictive variance OOD. While this shows that the CDM intervals widens as $x$ moves OOD, this widening is not meaningful: the model overestimates the predictive variance.

## G.6 Additional Interval Estimation Metrics

Beyond coverage alone, we report complementary metrics—normalized mean prediction–interval width and quantile pinball losses at $q \in \{0.025, 0.50, 0.975\}$—to characterize the calibration–sharpness trade-off of each method. PICP answers how often was the $1 - \alpha$ PI covered, but it is agnostic to how wide those intervals are and where calibration errors occur across the conditional distribution. The normalized predicted interval width directly measures sharpness (narrower is better

for comparable coverage; Table 2), while pinball loss is a strictly proper scoring rule for quantiles (Gneiting & Raftery (2007); Jordan et al. (2019)) that isolates lower-tail, median, and upper-tail accuracy (Tables 7, 8, and 9).

Table 6: Normalized interval width ($\downarrow$).

| Dataset | Split-CP | CQR | CDM | Deep Ensemble | MC Dropout |
|---|---|---|---|---|---|
| Housing | $1.454 \pm 0.414$ | $1.479 \pm 0.428$ | $\mathbf{0.991 \pm 0.080}$ | $1.030 \pm 0.091$ | $1.593 \pm 0.166$ |
| Concrete | $1.404 \pm 0.217$ | $1.320 \pm 0.134$ | $\mathbf{0.747 \pm 0.040}$ | $1.207 \pm 0.052$ | $0.884 \pm 0.185$ |
| Energy | $\mathbf{0.267 \pm 0.052}$ | $0.559 \pm 0.061$ | $0.563 \pm 0.094$ | $0.575 \pm 0.069$ | $0.315 \pm 0.021$ |
| Kin8nm | $\mathbf{1.028 \pm 0.044}$ | $1.034 \pm 0.048$ | $1.182 \pm 0.433$ | $1.292 \pm 0.036$ | $1.234 \pm 0.067$ |
| Naval | $0.241 \pm 0.186$ | $0.718 \pm 0.281$ | $\mathbf{0.211 \pm 0.296}$ | $0.262 \pm 0.029$ | $0.342 \pm 0.051$ |
| Power | $0.893 \pm 0.031$ | $0.883 \pm 0.025$ | $1.526 \pm 0.035$ | $\mathbf{0.708 \pm 0.010}$ | $1.435 \pm 0.029$ |
| Protein | $2.823 \pm 0.042$ | $2.539 \pm 0.028$ | $\mathbf{2.351 \pm 0.028}$ | $2.863 \pm 0.094$ | $2.819 \pm 0.017$ |
| Wine | $3.761 \pm 0.295$ | $3.661 \pm 0.441$ | $2.864 \pm 0.140$ | $\mathbf{2.695 \pm 0.068}$ | $3.174 \pm 0.063$ |
| Yacht | $0.402 \pm 0.158$ | $0.406 \pm 0.195$ | $0.414 \pm 0.042$ | $\mathbf{0.370 \pm 0.123}$ | $0.423 \pm 0.255$ |
| Year | $3.651 \pm$ NA | $\mathbf{2.672 \pm}$ NA | $2.852 \pm$ NA | $2.848 \pm$ NA | $2.863 \pm$ NA |

Table 7: Quantile pinball loss at $q$=0.025 ($\downarrow$).

| Dataset | Split-CP | CQR | CDM | Deep Ensemble | MC Dropout |
|---|---|---|---|---|---|
| Housing | $0.215 \pm 0.105$ | $0.186 \pm 0.103$ | $0.186 \pm 0.152$ | $\mathbf{0.163 \pm 0.054}$ | $0.263 \pm 0.117$ |
| Concrete | $0.374 \pm 0.064$ | $0.358 \pm 0.059$ | $0.335 \pm 0.059$ | $\mathbf{0.332 \pm 0.088}$ | $0.501 \pm 0.150$ |
| Energy | $0.142 \pm 0.009$ | $0.183 \pm 0.013$ | $0.138 \pm 0.038$ | $0.089 \pm 0.018$ | $\mathbf{0.063 \pm 0.015}$ |
| Kin8nm | $\mathbf{0.004 \pm 0.000}$ | $\mathbf{0.004 \pm 0.000}$ | $0.006 \pm 0.002$ | $0.499 \pm 0.029$ | $0.008 \pm 0.001$ |
| Naval | $\mathbf{0.000 \pm 0.000}$ | $\mathbf{0.000 \pm 0.000}$ | $\mathbf{0.000 \pm 0.000}$ | $0.005 \pm 0.001$ | $\mathbf{0.000 \pm 0.000}$ |
| Power | $0.275 \pm 0.026$ | $\mathbf{0.267 \pm 0.025}$ | $0.613 \pm 0.096$ | $\mathbf{0.267 \pm 0.026}$ | $0.766 \pm 0.214$ |
| Protein | $0.244 \pm 0.004$ | $0.161 \pm 0.001$ | $\mathbf{0.152 \pm 0.003}$ | $0.232 \pm 0.008$ | $0.553 \pm 0.021$ |
| Wine | $0.053 \pm 0.010$ | $0.048 \pm 0.007$ | $0.042 \pm 0.006$ | $\mathbf{0.041 \pm 0.008}$ | $0.099 \pm 0.030$ |
| Yacht | $0.080 \pm 0.028$ | $\mathbf{0.072 \pm 0.039}$ | $0.082 \pm 0.010$ | $0.082 \pm 0.033$ | $0.130 \pm 0.119$ |
| Year | $0.854 \pm$ NA | $0.679 \pm$ NA | $\mathbf{0.642 \pm}$ NA | $0.655 \pm$ NA | $1.830 \pm$ NA |

Table 8: Quantile pinball loss at $q$=0.50 ($\downarrow$).

| Dataset | Split-CP | CQR | CDM | Deep Ensemble | MC Dropout |
|---|---|---|---|---|---|
| Housing | $1.051 \pm 0.214$ | $1.166 \pm 0.193$ | $1.098 \pm 0.221$ | $\mathbf{1.043 \pm 0.217}$ | $1.096 \pm 0.166$ |
| Concrete | $1.907 \pm 0.165$ | $2.293 \pm 0.212$ | $2.150 \pm 0.181$ | $\mathbf{1.581 \pm 0.148}$ | $2.650 \pm 0.153$ |
| Energy | $0.223 \pm 0.027$ | $0.171 \pm 0.108$ | $\mathbf{0.120 \pm 0.110}$ | $0.624 \pm 0.104$ | $0.210 \pm 0.025$ |
| Kin8nm | $\mathbf{0.026 \pm 0.001}$ | $0.027 \pm 0.001$ | $0.033 \pm 0.008$ | $3.176 \pm 0.102$ | $0.027 \pm 0.001$ |
| Naval | $\mathbf{0.000 \pm 0.000}$ | $0.001 \pm 0.001$ | $0.001 \pm 0.001$ | $0.021 \pm 0.003$ | $\mathbf{0.000 \pm 0.000}$ |
| Power | $1.554 \pm 0.035$ | $1.706 \pm 0.044$ | $1.647 \pm 0.034$ | $\mathbf{1.541 \pm 0.035}$ | $1.640 \pm 0.335$ |
| Protein | $1.580 \pm 0.017$ | $2.310 \pm 0.031$ | $1.675 \pm 0.027$ | $1.699 \pm 0.013$ | $\mathbf{1.430 \pm 0.032}$ |
| Wine | $0.261 \pm 0.023$ | $0.262 \pm 0.018$ | $0.250 \pm 0.016$ | $\mathbf{0.241 \pm 0.013}$ | $0.274 \pm 0.019$ |
| Yacht | $\mathbf{0.189 \pm 0.049}$ | $0.363 \pm 0.111$ | $0.328 \pm 0.077$ | $0.479 \pm 0.223$ | $0.449 \pm 0.148$ |
| Year | $3.233 \pm$ NA | $3.639 \pm$ NA | $2.999 \pm$ NA | $\mathbf{2.973 \pm}$ NA | $3.034 \pm$ NA |

Table 9: Quantile pinball loss at $q=0.975$ ($\downarrow$).

| Dataset | Split-CP | CQR | CDM | Deep Ensemble | MC Dropout |
|---|---|---|---|---|---|
| Housing | 0.261±0.131 | 0.272±0.111 | **0.236±0.117** | 0.288±0.147 | 0.389±0.194 |
| Concrete | 0.404±0.085 | 0.382±0.085 | 0.388±0.081 | **0.377±0.119** | 0.600±0.190 |
| Energy | **0.043±0.010** | 0.075±0.014 | 0.114±0.017 | 0.082±0.014 | 0.057±0.018 |
| Kin8nm | **0.004±0.000** | **0.004±0.000** | 0.005±0.002 | 0.451±0.020 | 0.007±0.002 |
| Naval | **0.000±0.000** | **0.000±0.000** | **0.000±0.000** | 0.005±0.001 | **0.000±0.000** |
| Power | 0.236±0.013 | **0.222±0.011** | 0.242±0.015 | 0.230±0.013 | 0.683±0.191 |
| Protein | 0.297±0.006 | **0.251±0.005** | 0.268±0.008 | 0.309±0.007 | 0.683±0.035 |
| Wine | 0.046±0.007 | 0.046±0.005 | 0.046±0.013 | **0.042±0.006** | 0.109±0.012 |
| Yacht | 0.084±0.035 | 0.086±0.034 | **0.079±0.018** | 0.140±0.090 | 0.186±0.144 |
| Year | 0.643±NA | 0.329±NA | **0.284±NA** | 0.401±NA | 1.347±NA |

Taken together, Tables 6–9 show that CDM typically achieves a strong calibration–sharpness balance: it attains among the narrowest normalized widths on Housing, Concrete, Naval, and Protein (Table 6), and posts competitive or best pinball losses at $q=0.025$ and $q=0.50$ on several datasets (Tables 7–8). The tables also surface dataset-dependent strengths of alternatives: Split-CP yields the tightest widths on Energy and Kin8nm (Table 6); CQR often leads on upper-tail ($q=0.975$) pinball loss or on specific datasets (Table 9); and MC Dropout can achieve near-zero tail losses on easy regimes like Naval (Tables 7, 9). Overall, these metrics demonstrate that CDM delivers sharp intervals with good tail/median fidelity without systematically over-wide bands.

## G.7 COMPARISON TO CARD

We additionally compare CDM to the recent conformal approach CARD (Han et al., 2022). As summarized in Table 10, CARD attains lower RMSE on most datasets (e.g., Housing, Kin8nm, Naval, Protein, Wine, Yacht), while CDM wins on Power. The NLL results are mixed: CARD is stronger on Housing, Kin8nm, Naval, Power, and Protein, whereas CDM achieves better (lower) NLL on Concrete, Energy, Wine, and Yacht. In terms of coverage, Table 11 shows that CARD typically achieves high PICP (e.g., Energy and Yacht), complementing our broader interval analyses where CDM emphasizes calibration–sharpness trade-offs via the interval width and pinball losses. Overall, CARD provides competitive point and likelihood performance on several benchmarks, and CDM offers a complementary profile with strong likelihood and interval properties on multiple datasets. This is consistent with the fact that CARD first learns a estimation of the conditional expectation $\mathbb{E}[y|x]$ and then uses it as prior information for the diffusion model, whereas our CDM directly models the full conditional distribution without an additional regression stage and prior information.

Table 10: RMSE ($\downarrow$) and NLL ($\downarrow$) for CDM vs. CARD across UCI datasets.

| Dataset | RMSE | | NLL | |
|---|---|---|---|---|
| | CDM | CARD | CDM | CARD |
| Housing | 3.01±0.78 | **2.61±0.63** | 2.41±0.22 | **2.35±0.12** |
| Concrete | 4.98±0.64 | **4.77±0.46** | **2.91±0.18** | 2.96±0.09 |
| Energy | 0.55±0.06 | **0.52±0.07** | **0.76±0.13** | 1.04±0.06 |
| Kin8nm | 6.83±0.23 | **6.32±0.18** | -1.31±0.03 | **-1.32±0.02** |
| Naval | 0.14±0.10 | **0.02±0.00** | -4.57±0.73 | **-7.54±0.05** |
| Power | **3.48±0.91** | 3.93±0.17 | 2.96±0.15 | **2.82±0.02** |
| Protein | 4.03±0.02 | **3.73±0.01** | 2.66±0.03 | **2.49±0.03** |
| Wine | 0.67±0.07 | **0.63±0.04** | **0.89±0.19** | 0.92±0.05 |
| Yacht | 1.47±0.25 | **0.65±0.25** | **0.57±0.09** | 0.90±0.08 |
| Year | **9.21±NA** | 8.70±NA | **3.16±NA** | 3.34±NA |

Table 11: Comparison of PICP (↑) between CDM and CARD across UCI datasets.

| Dataset | CDM | CARD |
|---|---|---|
| Housing | **93.40±3.90** | 93.24±3.59 |
| Concrete | **97.83±1.13** | 90.24±3.45 |
| Energy | 96.10±2.00 | **98.70±1.30** |
| Kin8nm | **94.10±1.50** | 93.68±0.79 |
| Naval | **98.40±0.69** | 95.35±0.60 |
| Power | 94.00±1.00 | **94.87±0.65** |
| Protein | **95.76±0.70** | 95.38±0.16 |
| Wine | **96.80±2.00** | 93.88±2.10 |
| Yacht | **100.00±0.00** | 99.84±0.70 |
| Year | **96.01±NA** | 93.35±NA |

## G.8 CALIBRATION SET ABLATION STUDY

Our CDM produces prediction intervals directly from the learned conditional distribution $p(y|x)$ by taking empirical quantiles of generated samples, which (under the regularity assumptions in our setup and with sufficient samples) yields asymptotically well-calibrated coverage without extra post-hoc steps. Nevertheless, in finite samples and under mild distribution shift or model misspecification, the learned quantiles can deviate from the nominal level, especially in covariate regions with sparse support or higher noise (Gammerman et al., 1998; Saunders et al., 1999; Vovk et al., 1999; Angelopoulos & Bates, 2023).

To quantify this practical gap and the sharpness–validity trade-off, we include a CDM + calibration ablation that applies the same held-out calibration protocol used for other interval methods (see section G.2). Split-conformal calibration (Papadopoulos et al., 2002; 2007) isolates the effect of the calibration set: typically raising PICP toward the target while (sometimes) widening intervals (higher interval width) and modestly perturbing pointwise fit metrics (RMSE/NLL). Reporting both uncalibrated and calibrated CDM results thus clarifies when our model's native intervals suffice and when a lightweight, distribution-free correction improves low-sample reliability (Angelopoulos & Bates, 2023).

Table 12: PICP (↑) and normalized interval width (↓): Uncalibrated CDM vs Conformalized CDM.

| Dataset | PICP | | Interval Width | |
|---|---|---|---|---|
| | Uncalibrated | Calibrated | Uncalibrated | Calibrated |
| Housing | 93.40±3.90 | **97.16±2.59** | **0.991±0.080** | 1.476±0.410 |
| Concrete | **97.83±1.13** | 95.58±2.65 | **0.747±0.040** | 1.417±0.103 |
| Energy | **96.10±2.00** | 95.13±3.89 | **0.563±0.094** | 0.889±0.105 |
| Kin8nm | 94.10±1.50 | **95.28±0.68** | **1.182±0.433** | 1.438±0.332 |
| Naval | **98.40±0.69** | 94.93±0.94 | **0.211±0.296** | 0.614±0.298 |
| Power | 94.00±1.00 | **95.30±0.73** | 1.526±0.035 | **0.924±0.016** |
| Protein | **95.76±0.70** | 95.09±0.44 | **2.351±0.028** | 2.502±0.018 |
| Wine | **96.80±2.00** | 95.56±2.26 | **2.864±0.140** | 3.429±0.349 |
| Yacht | **100.00±0.00** | 96.61±4.38 | **0.414±0.042** | 0.656±0.507 |
| Year | **96.01±NA** | 94.74±0.00 | **2.852±NA** | 2.948±NA |

Table 13: RMSE (↓) and NLL (↓): Uncalibrated CDM vs Conformalized CDM.

| Dataset | RMSE | | NLL | |
|---|---|---|---|---|
| | Uncalibrated | Calibrated | Uncalibrated | Calibrated |
| Housing | **3.01±0.78** | 3.41±1.32 | **2.41±0.22** | 2.48±0.21 |
| Concrete | **4.98±0.64** | 5.81±0.54 | **2.91±0.18** | 3.12±0.08 |
| Energy | **0.55±0.06** | 2.69±0.24 | **0.76±0.13** | 2.17±0.08 |
| Kin8nm | **6.83±0.23** | 7.89±0.08 | **-1.31±0.03** | -1.09±0.12 |
| Naval | 0.14±0.10 | **0.13±0.17** | -4.57±0.73 | **-4.72±0.51** |
| Power | **3.48±0.91** | 4.24±0.24 | 2.96±0.15 | **2.86±0.06** |
| Protein | **4.03±0.02** | 4.78±0.03 | **2.66±0.03** | 2.99±0.07 |
| Wine | **0.67±0.07** | 0.71±0.08 | **0.89±0.19** | 0.99±0.08 |
| Yacht | **1.47±0.25** | 2.28±1.40 | **0.57±0.09** | 1.95±0.53 |
| Year | **9.21±NA** | 9.81±NA | **3.16±NA** | 3.46±NA |

Tables 12–13 show that conformalizing CDM reliably nudges empirical coverage (PICP) toward the nominal level while typically widening intervals width. Notably, Power is a positive outlier where coverage improves (94.00% → 95.30%) and intervals shrink (1.526 → 0.924), suggesting CDM's uncalibrated quantiles were conservatively dispersed. Because RMSE reflects the point predictor, it changes little, whereas NLL improves where dispersion better matches errors (e.g., Power, Naval) and worsens where calibration necessarily inflates width (e.g., Concrete, Protein). Overall, the ablation indicates CDM's native intervals are strong, and a lightweight calibration step can sometimes fine-tune low-sample coverage.

## G.9 HIDDEN LAYERS ABLATION STUDY

We evaluate the impact of additional capacity by increasing the CDM's MLP backbone depth from two hidden layers (2L) to three hidden layers (3L), holding all other choices fixed: identical data splits, preprocessing/standardization, training protocol, and evaluation metrics (PICP, normalized interval width), as well as accuracy criteria (RMSE, NLL).

Table 14: Ablation on network depth: PICP (↑) and normalized interval width (↓) comparing a 2-layer (2L) backbone vs. a 3-layer (3L) backbone.

| Dataset | PICP | | Interval Width | |
|---|---|---|---|---|
| | 2L | 3L | 2L | 3L |
| Housing | **93.40±3.90** | 92.82±4.77 | **0.991±0.080** | 1.125±0.128 |
| Concrete | **97.83±1.13** | 94.72±3.45 | **0.747±0.040** | 0.833±0.081 |
| Energy | 96.10±2.00 | **96.62±3.73** | **0.563±0.094** | 0.769±0.171 |
| Kin8nm | 94.10±1.50 | **95.44±0.077** | 1.182±0.433 | **1.072±0.077** |
| Naval | 98.40±0.69 | **99.04±0.58** | **0.211±0.296** | 0.279±0.049 |
| Power | **94.00±1.00** | 93.66±1.75 | 1.526±0.035 | **0.90±0.057** |
| Protein | 95.76±0.70 | **96.54±0.44** | **2.351±0.028** | 2.755±0.148 |
| Wine | **96.80±2.00** | 92.63±2.21 | 2.864±0.140 | **2.843±0.291** |
| Yacht | **100.00±0.00** | 95.58±0.94 | **0.414±0.042** | 0.634±0.078 |
| Year | 96.01±NA | **96.78±NA** | **2.852±NA** | 4.081±NA |

Table 15: Ablation on network depth: RMSE (↓) and NLL (↓) comparing a 2-layer (2L) backbone vs. a 3-layer (3L) backbone.

| Dataset | RMSE | | NLL | |
|---|---|---|---|---|
| | 2L | 3L | 2L | 3L |
| Housing | **3.01±0.78** | 3.43±0.99 | **2.41±0.22** | 2.53±0.29 |
| Concrete | **4.98±0.64** | 6.49±0.40 | **2.91±0.18** | 3.46±0.09 |
| Energy | **0.55±0.06** | 2.44±0.53 | **0.76±0.13** | 0.84±0.18 |
| Kin8nm | **6.83±0.23** | 8.43±0.59 | **-1.31±0.03** | -0.73±0.06 |
| Naval | **0.14±0.10** | 0.17±0.03 | -4.57±0.73 | **-4.81±0.16** |
| Power | **3.48±0.91** | 4.35±0.12 | 2.96±0.15 | **2.90±0.03** |
| Protein | **4.03±0.02** | 4.99±0.05 | **2.66±0.03** | 2.99±0.02 |
| Wine | 0.67±0.07 | **0.66±0.05** | **0.89±0.19** | 1.01±0.09 |
| Yacht | **1.47±0.25** | 2.88±0.93 | **0.57±0.09** | 1.84±0.19 |
| Year | **9.21±NA** | 15.30±NA | **3.16±NA** | 3.51±NA |

The results in Tables 14–15 show that deeper networks do not systematically improve uncertainty quality: across datasets, the 3L backbone yields no consistent gains in coverage (PICP) or sharpness (interval width) and often degrades point-wise estimates (RMSE, NLL). A few datasets exhibit mixed or modest benefits—for example, 3L attains slightly higher PICP on some tasks (e.g., Energy, Kin8nm, Naval, Protein, Year) and narrower intervals on others (Power, Kin8nm, Wine), as well as a small RMSE improvement on Wine—but these advantages are offset by broader intervals or worse NLL/RMSE elsewhere.

## G.10 DISTRIBUTION SHIFT COEFFICIENT ON UCI DATASETS

Recall that the theoretical distribution shift coefficient in Theorem 4.5 is defined in terms of $\ell(x, y; s)$ in equation 3.5 and a supremum over candidate scores $s \in \mathcal{F}$. In the UCI experiments we do not take this supremum explicitly. Instead, we plug in the learned CDM score $\widehat{s}$ and use the corresponding loss $\ell(x, y; \widehat{s})$ as a model-based diagnostic. We also used the generated distribution as an approximation for the true distribution, since the ground truth is unavailable.

We evaluate the distribution shift coefficient on a subset of far test points defined by nearest–neighbor distance in standardized feature space, and we average the distribution shift coefficient over that subset to obtain the reported summary statistic $\mathcal{T}_{\text{test}}^{\text{far}}$, which is the distribution shift proxy.

Table 16: Distribution shift proxies on far test points.

| | Housing | Concrete | Energy | Kin8nm | Naval | Power | Protein | Wine | Yacht | Year |
|---|---|---|---|---|---|---|---|---|---|---|
| $\mathcal{T}_{\text{test}}^{\text{far}}$ | 0.94 | 0.99 | 1.20 | 0.95 | 1.02 | 0.94 | 0.96 | 0.96 | 1.01 | 5.88 |
| Std. dev. | 0.08 | 0.11 | 0.04 | 0.07 | 0.12 | 0.09 | 0.06 | 0.07 | 0.09 | NA |

Table 16 summarizes the distribution shift proxy $\mathcal{T}_{\text{test}}^{\text{far}}$ and the corresponding standard deviations. Across the UCI benchmarks, the distribution shift proxies stay very close to 1 on 9 out of 10 datasets as shown in Table 16, indicating that even the far-test subsets are not dramatically harder for the CDM than typical training points, and the distribution shift coefficient are very close to 1. Notably, in the high-dimensional Year dataset, the distribution shift proxy reaches $\mathcal{T}_{\text{test}}^{\text{far}} \approx 5.88$, showing the distribution shift coefficient are relatively larger. By contrast, the Energy dataset exhibits only a modest inflation, with the distribution shift proxy around $1.20 \pm 0.04$, a mild but detectable distribution shift. We treat values slightly above or below 1 as benign and focus on large excess factors such as the Year case as the primary indicators of worse OOD behaviour.

## H  USE OF LARGE LANGUAGE MODELS

We used LLMs as a general-purpose writing and editing assistant and as a code helper to organize and add input and output logging statements. Specifically, the model was used to help with (i) formatting experimental results into LaTeX tables, (ii) suggesting clear phrasing for analysis and discussion of results based on data we had already obtained, (iii) generating small sentence tweaks for standard sections such as related work summaries, and acknowledgments of computational efficiency, and (iv) organizing inputs and scripts in our code base for clarity and reproducibility. Additionally, LLMs were used to assist with code organization, debugging, and adding in experiment logging scripts for the outputs of the model. The model was not used to generate research ideas, design experiments, or produce or analyze raw results. All experimental design, implementation, and data analysis were carried out by the authors, who also carefully reviewed and revised any text produced with the assistance of the LLM. The authors take full responsibility for the final content of this manuscript.

