# OpenReview forum: "Capturing Uncertainty in Regression via Conditional Diffusion Models"
_ICLR.cc/2026/Conference — Submitted to ICLR 2026_

### Official Review · Reviewer_Gs1W · 2025-10-26

**Soundness:** 3
**Presentation:** 2
**Contribution:** 2
**Rating:** 4
**Confidence:** 3

**Summary:**

The paper proposes a new framework for uncertainty quantification using conditional diffusion models for nonparametric regression with data-dependent noise. The authors present a rigorous theoretical analysis. Note: I am not particular familiar with the theory of diffiusion models, so my review can be biased at times.

**Strengths:**

- A rigorous mathematical analysis of the proposed framework is provided
- emprirical comparison of the proposed framework to standard baselinem methods is provided

**Weaknesses:**

- The paper frames the contribution as a new method in the abstract but then states in line 108 that the "work prioritizes theortical analysis". This is an inconsistent framing of your contribution
- The comparison to prior work on diffusion based uncertainty quantification is missing
- There is no comparison or discussion on the computational comparison to baseline methods
- The insights from the theortical analysis is not utilised in the experimental results. What do we gain from the theory and how can we see this in the experiments?

**Questions:**

- The font size in Figure 1 seems to be smaller. Please correct for that.
- line 148. References end with "[...], Unlike". Some grammar is off here.
- if you have no section 5.2, then section 5.1 does not head a separate section.
- Experiments: Can you explain the baseline methods? They are not stated so within the main text.
- You mention that Han et al. (2022) provides a method for UQ using diffusion models.
  - How is that different from your framework?
  - can you provide experimental results for diffusion based UQ to see the benefits of your method?
- Can you provide a computational analysis (theoretical and experimental) into the overhead of your method compared to baseline methods?
- Can you "verify" your theortical analysis in experiments? Or at least connect the theory to the experimental results?
- In the limitations there is distribution shift mentioned. Can you provide analysis or experiemntal results on this topic?

---

> ### Author Response · Authors · 2025-11-26
>
> We sincerely appreciate the reviewer’s valuable comments and suggestions. We respond to weakness concerns and questions as follows.
>
> >**W1: The paper frames the contribution as a new method in the abstract but then states in line 108 that the “work prioritizes theoretical analysis”. This is an inconsistent framing of your contribution.**
>
> **A1**  We appreciate the reviewer’s comment about the framing. By saying “a novel UQ framework” in the abstract, our intention is not to claim a novel diffusion-based method for uncertainty quantification, but rather to introduce a conditional diffusion–based UQ procedure for nonparametric regression with data-dependent noise (Algorithm 1), and provide a new theoretical coverage analysis for the resulting prediction intervals (Theorem 4.5), together with supporting experiments.
>
> In this sense, there is no contradiction between the abstract and line 108: Algorithm 1 is the specific diffusion-based UQ framework we study, but the main novelty of the paper lies in the theoretical coverage guarantees.
>
> We agree, however, that the phrase “we propose a novel UQ framework based on diffusion models” in the abstract can give the impression that the contribution is primarily a new method. To avoid this ambiguity, we prioritize theoretical contribution in the introduction, and we revised the abstract.
> >**W2: The comparison to prior work on diffusion-based uncertainty quantification is missing.**
>
> **A2:** We have now added a diffusion-based baseline to our comparison: CARD, a recent diffusion-model approach for uncertainty quantification (see new “CARD” column in the PICP table above). Across UCI datasets, our diffusion method CDM matches or exceeds CARD on several tasks while remaining competitive on the rest. This and an explanation are also updated in Appendix G.7 in the main paper.
> ### Table 1. Comparison of PICP between CDM and CARD (↑)
> | Dataset  | CDM (ours) | CARD|
> | :------- | :-------------- | :------------- |
> | Housing| **93.40±3.90**| 93.24±3.59 |
> | Concrete| **97.83±1.13**| 90.24±3.45 |
> | Energy| 96.10±2.00| **98.70±1.30** |
> | Kin8nm| **94.10±1.50**| 93.68±0.79|
> | Naval| **98.40±0.69**| 95.35±0.60 |
> | Power| 94.00±1.00| **94.87±0.65** |
> | Protein  | **95.76±0.70**| 95.38±0.16|
> | Wine| **96.80±2.00** | 93.88±2.10|
> | Yacht| **100.00±0.00**| 99.84±0.70|
> | Year| **96.01±NA** | 93.35±NA|
>
> ### Table 2. Comparison of RMSE & NLL between CDM and CARD (↓)
> | Dataset | CDM (RMSE)| CARD (RMSE) | CDM (NLL)| CARD (NLL)  |
> | :------- | :------------ | :------------ | :------------ | :------------- |
> | Housing  | 3.01±0.78| **2.61±0.63** | 2.41±0.22| **2.35±0.12**  |
> | Concrete | 4.98±0.64 | **4.77±0.46**| **2.91±0.18** | 2.96±0.09 |
> | Energy   | 0.55±0.06| **0.52±0.07**| **0.76±0.13** | 1.04±0.06|
> | Kin8nm   | 6.83±0.23| **6.32±0.18** | -1.31±0.03| **-1.32±0.02** |
> | Naval    | 0.14±0.10| **0.02±0.00** | -4.57±0.73| **-7.54±0.05** |
> | Power    | **3.48±0.91** | 3.93±0.17| 2.96±0.15 | **2.82±0.02**|
> | Protein  | 4.03±0.02| **3.73±0.01** | 2.66±0.03| **2.49±0.03**  |
> | Wine| 0.67±0.07 | **0.63±0.04** | **0.89±0.19** | 0.92±0.05|
> | Yacht| 1.47±0.25 | **0.65±0.25** | **0.57±0.09** | 0.90±0.08|
> | Year| 9.21±NA   | **8.70±NA** | **3.16±NA**   | 3.34±NA|
> >**W3: There is no comparison or discussion on the computational comparison to baseline methods.**
>
> **A3:** See Q4.
>
> >**W4: The insights from the theoretical analysis are not utilised in the experimental results. What do we gain from the theory and how can we see this in the experiments?**
>
> **A4:** Theorem 4.5 (coverage guarantee) decomposes the coverage error at a fixed test point $x_{\text{new}}$ into two parts:
> 1. $\sqrt{\frac{2}{n_1}\log \frac{4}{\delta}}$ coming from estimating the conditional quantiles with $n_1$ generated samples
> 2. a distribution–shift term $\mathcal{T}(x_{\text{new}})\cdot \mathcal{O}\left(n^{-\frac{\beta}{4(d+\beta+1)}}(\log n)^{\max(9,\frac{3}{2}+\frac{\beta}{4})}\right)$ that grows with the score error and the shift coefficient $\mathcal{T}(x_{\text{new}})$.
>
> We verify and illustrate this behavior experimentally in the new Figure 3 and 4 in the appendix G.5: Figure 3 and 4 plot both the true 95% interval and the interval produced by our CDM, for inputs inside and outside the training $x$–range. When $x_{\text{new}}$ lies within the training region, the learned interval essentially overlaps the true 95% band, indicating that the coverage error is small. This corresponds to a small $\mathcal{T}(x_{\text{new}})$ (little distribution shift) and $\sqrt{\frac{2}{n_1}\log \frac{4}{\delta}}$ is a negligible term since we use a large $n_1$, in line with Theorem 4.5. As $x_{\text{new}}$ moves outside the training range, Figure 3 and 4 shows that our intervals deviate from the true band and coverage deteriorates. This is precisely the regime where the theorem predicts that the second term, amplified by a large $\mathcal{T}(x_{\text{new}})$, dominates and the guarantee becomes sensitive to distribution shift.

---

> > ### Author Response · Authors · 2025-11-26
> >
> > >**Q1:** - The font size in Figure 1 seems to be smaller. Please correct for that.
> > >- Line 148: References end with `"[...], Unlike"`. Some grammar is off here.
> > >- If you have no Section 5.2, then Section 5.1 does not need to head a separate section.
> >
> > **A1:** We updated the paper accordingly.
> >
> > >**Q2: Experiments: Can you explain the baseline methods? They are not stated as such within the main text.**
> >
> > **A2:** We defer the full descriptions to Appendix G (Section G.2.1).
> >
> > Concretely, Section G.2.1 describes the four baselines we compare against:
> >
> > 1. **Split-Conformal Prediction (Split-CP):** We learns a regression model $\hat f(x)$, computes nonconformity scores $s_i = |y_i - \hat f(x_i)|$ on a calibration set, and forms intervals around $\hat f(x_{\text{new}})$ using the $(1-\alpha)$ empirical quantile of $\{s_i\}$.([1])
> > 2. **Conformal Quantile Regression (CQR):** trains lower/upper quantile regressors $\ell_r(x)$ and $u_r(x)$, computes conformal scores on a calibration split, and adjusts the quantile bands by a learned offset to obtain calibrated intervals.([2])
> > 3. **Monte Carlo Dropout (MC Dropout):** We apply dropout at test time, performs $N_{\text{MC}}$ stochastic forward passes to obtain $\{y^{(b)}(x_{\text{new}})\}$, injects observation noise, and forms intervals from empirical quantiles of these samples.([3])
> > 4. **Deep Ensembles (DE):** We train 5 independently initialized networks with heteroscedastic outputs $(\mu_m(x),\sigma_m^2(x))$, draws $M$ predictive samples $y_m \sim \mathcal N(\mu_m(x_{\text{new}}),\sigma_m^2(x_{\text{new}}))$, and constructs intervals from their empirical quantiles.([4])
> >
> > >**Q3: You mention that Han et al. (2022) provides a method for UQ using diffusion models.
> > How is that different from your framework?
> > Can you provide experimental results for diffusion-based UQ to see the benefits of your method?**
> >
> > **A3:** We compare our method with CARD [5] along two dimensions: theory and experiment. In the forward diffusion process, CARD assumes $p(y_T \mid x) = \mathcal{N}(f_{\phi}(x), I),$
> > where $f_{\phi}$ encodes prior knowledge about the relationship between $y_0$ and $x$. In contrast, our conditional diffusion model directly adds noise to $y_t$, so that $y_T$ follows a standard Gaussian distribution $\mathcal{N}(0, I)$, rather than $\mathcal{N}(f_{\phi}(x), I)$.
> >
> > In the backward process, CARD conditions on $f_{\phi}(x)$, while CDM does not rely on such a prior. It is also worth noting that CARD mentions the possibility of dropping $f_{\phi}(x)$ and instead using $\mathcal{N}(0, I)$ as the prior; in that case, their framework becomes mathematically equivalent to CDM.
> >
> > For comparison of experimental results, we have tables 1 and 2 provided above and in section G.7 in the main paper. Across UCI datasets, our method CDM matches or exceeds CARD on several tasks while remaining competitive on the rest. Specifically, CDM performs better in terms of PICP, while CARD achieves lower NLL and RMSE. This is consistent with the fact that CARD first learns a estimation of the conditional expectation $\mathbb{E}[y | x]$ and then uses it as prior information for the diffusion model, whereas our CDM directly models the full conditional distribution without an additional regression stage and prior information.
> >
> > ### Table 1. Comparison of PICP between CDM and CARD (↑) (same as before)
> > | Dataset  | CDM (ours)            | CARD           |
> > | :------- | :-------------- | :------------- |
> > | Housing  | **93.40±3.90**  | 93.24±3.59     |
> > | Concrete | **97.83±1.13**  | 90.24±3.45     |
> > | Energy   | 96.10±2.00      | **98.70±1.30** |
> > | Kin8nm   | **94.10±1.50**  | 93.68±0.79     |
> > | Naval    | **98.40±0.69**  | 95.35±0.60     |
> > | Power    | 94.00±1.00      | **94.87±0.65** |
> > | Protein  | **95.76±0.70**  | 95.38±0.16     |
> > | Wine     | **96.80±2.00**  | 93.88±2.10     |
> > | Yacht    | **100.00±0.00** | 99.84±0.70     |
> > | Year     | **96.01±NA**  | 93.35±NA       |
> >
> > ### Table 2. Comparison of RMSE & NLL between CDM and CARD (↓) (same as before)
> > | Dataset  | CDM (RMSE)    | CARD (RMSE)   | CDM (NLL)     | CARD (NLL)     |
> > | :------- | :------------ | :------------ | :------------ | :------------- |
> > | Housing  | 3.01±0.78     | **2.61±0.63** | 2.41±0.22     | **2.35±0.12**  |
> > | Concrete | 4.98±0.64     | **4.77±0.46** | **2.91±0.18** | 2.96±0.09      |
> > | Energy   | 0.55±0.06     | **0.52±0.07** | **0.76±0.13** | 1.04±0.06      |
> > | Kin8nm   | 6.83±0.23     | **6.32±0.18** | -1.31±0.03    | **-1.32±0.02** |
> > | Naval    | 0.14±0.10     | **0.02±0.00** | -4.57±0.73    | **-7.54±0.05** |
> > | Power    | **3.48±0.91** | 3.93±0.17     | 2.96±0.15     | **2.82±0.02**  |
> > | Protein  | 4.03±0.02     | **3.73±0.01** | 2.66±0.03     | **2.49±0.03**  |
> > | Wine     | 0.67±0.07     | **0.63±0.04** | **0.89±0.19** | 0.92±0.05      |
> > | Yacht    | 1.47±0.25     | **0.65±0.25** | **0.57±0.09** | 0.90±0.08      |
> > | Year     | 9.21±NA   | **8.70±NA**      | **3.16±NA**   | 3.34±NA        |

---

> ### Author Response · Authors · 2025-11-26
>
> >**Q4: Can you provide a computational analysis (theoretical and experimental) of the overhead of your method compared to baseline methods?**
>
> **A4**  We provide a computational analysis in Appendix G.4, including Table 5 (in Appendix G.4), which reports the total time for our conditional diffusion model (CDM) and two computationally demanding baselines, Deep Ensembles (DE) and MC Dropout on ten UCI datasets. We have copied it for your convenience below.
>
>
> ### Table 5. Total time (s) reported for each dataset and method
> | **Dataset** | **CDM** | **DE** | **MC Dropout** |
> |-------------|---------|--------|----------------|
> | Housing | 12.39±0.05 | **11.74±0.22** | 50.93±0.72 |
> | Concrete | 62.21±0.36 | **19.8±1.16** | 120.24±5.03 |
> | Energy | **11.06±0.07** | 13.1±0.39 | 69.03±1.50 |
> | Kin8nm | 234.65±1.73 | **161.55±0.45** | 1040.30±191.75 |
> | Naval | 270.42±1.52 | **137.3±0.59** | 1607.60±122.14 |
> | Power | 284.15±1.47 | **123.35±0.63** | 278.58±3.90 |
> | Protein | 280.47±0.57 | **277.44±0.39** | 9059.40±44.40 |
> | Wine | **94.76±0.41** | 106.9±1.81 | 193.70±16.56 |
> | Yacht | 19.04±0.05 | **7.01±0.19** | 36.78±0.58 |
> | Year | **1570.72±0.00** | 1682.25±0.00 | 10333.37±0.00 |
> | Blog | **276.39±2.73** | 317.22±1.54 | 297.68±26.39 |
>
>
> From the theoretical perspective, the computational cost scales as follows:
> 1. **CDM:** The inference cost of CDM scale linearly with the number of diffusion steps $T$ and the number of generated samples $n_1$ per test point.
> 2. **DE:** requires training $M=5$ independently initialized networks; total training cost is roughly $M$ times that of a single network.
> 3. **MC Dropout:** trains one network but, at test time, needs $N_{\text{MC}}$ stochastic forward passes per input to approximate the predictive distribution.
>
> Thus CDM trades a larger per-sample cost (due to the diffusion steps) for avoiding multiple independent networks (DE) or very large numbers of stochastic passes (MC Dropout).
>
> From the experimental perspective, Table 5 shows that, in practice, CDM is competitive with DE and substantially more efficient than MC Dropout. DM therefore demonstrates a clear efficiency advantage, achieving strong uncertainty estimates with significantly reduced computational overhead.
>
> >**Q5: Can you “verify” your theoretical analysis in experiments? Or at least connect the theory to the experimental results?**
>
> **A5 (same with W4)**   Theorem 4.5 (coverage guarantee) decomposes the coverage error at a fixed test point $x_{\text{new}}$ into two parts:
> 1. $\sqrt{\frac{2}{n_1}\log \frac{4}{\delta}}$ coming from estimating the conditional quantiles with $n_1$ generated samples
> 2. a distribution–shift term $\mathcal{T}(x_{\text{new}})\cdot \mathcal{O}\left(n^{-\frac{\beta}{4(d+\beta+1)}}(\log n)^{\max(9,\frac{3}{2}+\frac{\beta}{4})}\right)$ that grows with the score error and the shift coefficient $\mathcal{T}(x_{\text{new}})$.
>
> We verify and illustrate this behavior experimentally in the new Figure 3 and 4 in the appendix G.5: Figure 3 and 4 plot both the true 95% interval and the interval produced by our CDM, for inputs inside and outside the training $x$–range. When $x_{\text{new}}$ lies within the training region, the learned interval essentially overlaps the true 95% band, indicating that the coverage error is small. This corresponds to a small $\mathcal{T}(x_{\text{new}})$ (little distribution shift) and $\sqrt{\frac{2}{n_1}\log \frac{4}{\delta}}$ is a negligible term since we use a large $n_1$, in line with Theorem 4.5. As $x_{\text{new}}$ moves outside the training range, Figure 3 and 4 shows that our intervals deviate from the true band and coverage deteriorates. This is precisely the regime where the theorem predicts that the second term, amplified by a large $\mathcal{T}(x_{\text{new}})$, dominates and the guarantee becomes sensitive to distribution shift.
>
> [1] Papadopoulos, Harris, Volodya Vovk, and Alex Gammerman. "Conformal prediction with neural networks." 19th IEEE International Conference on Tools with Artificial Intelligence (ICTAI 2007). Vol. 2. IEEE, 2007.
>
> [2] Romano, Yaniv, Evan Patterson, and Emmanuel Candes. "Conformalized quantile regression." Advances in neural information processing systems 32 (2019).
>
> [3] Gal, Yarin, and Zoubin Ghahramani. "Dropout as a bayesian approximation: Representing model uncertainty in deep learning." international conference on machine learning. PMLR, 2016.
>
> [4] Lakshminarayanan, Balaji, Alexander Pritzel, and Charles Blundell. "Simple and scalable predictive uncertainty estimation using deep ensembles." Advances in neural information processing systems 30 (2017).
>
> [5] Han, Xizewen, Huangjie Zheng, and Mingyuan Zhou. "Card: Classification and regression diffusion models." Advances in Neural Information Processing Systems 35 (2022): 18100-18115.

---

> > ### Comment · Reviewer_Gs1W · 2025-11-27
> >
> > Thank you for the detailed rebuttal. The clarifications and additional experiments address several of my concerns with the manuscript. I encourage and urge the authors to bring key baseline descriptions and parts of the computational comparison into the main text rather than relying primarily on the appendix. The reader should be made aware of your analysis which is in the appendix within the main text. This can clarify many points for a reader.
> >
> > As a follow-up question, could you clarify and possibly quantify the distribution-shift term in Theorem 4.5 for at least one dataset, so the theory to experiment connection becomes more explicit?
> >
> > In the meantime, I raise my score by one point.

---

> ### Author Response · Authors · 2025-12-03
>
> Dear Reviewer Gs1W,
>
> Thank you very much for your continued engagement with our paper and for raising your score. We appreciate your constructive suggestion and your follow-up question. We respond to them as follows.
>
> **1. Moving Baseline and Computational Details to Main Text**
> We agree that bringing key details into the main text significantly aids the reader’s understanding of the analysis. We have added a summary of the baseline descriptions and the computational cost and resources used directly into **Section 5** of the main paper. Please refer to the revised text in **lines 411–416** and **lines 440–443**.
>
> **2. Quantifying the Distribution-Shift Term for UCI Datasets**
> Regarding your request to quantify the distribution-shift term in Theorem 4.5 for UCI Datasets, we have added a new section, **Appendix G.10**, where we explicitly evaluate the distribution shift term $\mathcal{T}(x_{\text{new}})$ on the UCI Datasets. We approximate the distribution shift coefficient on a subset of far test points defined by nearest–neighbor distance in standardized feature space. We average the approximated distribution shift coefficient $\mathcal{T}(x_{\text{new}})$ over that subset to obtain the distribution shift proxy. Table 1 (Table 16 in Appendix G.10) summarizes the distribution shift proxy $\mathcal{T}_{\text{test}}^{\text{far}}$ and the corresponding standard deviations.
>
> **Table 1: Distribution–shift proxies on far test points.**
>
> | | Housing | Concrete | Energy | Kin8nm | Naval | Power | Protein | Wine | Yacht | Year |
> | :--- | :---: | :---: | :---: | :---: | :---: | :---: | :---: | :---: | :---: | :---: |
> | $\mathcal{T}_{\text{test}}^{\text{far}}$ | 0.94 | 0.99 | 1.20 | 0.95 | 1.02 | 0.94 | 0.96 | 0.96 | 1.01 | 5.88 |
> | Std. dev. | 0.08 | 0.11 | 0.04 | 0.07 | 0.12 | 0.09 | 0.06 | 0.07 | 0.09 | NA |
>
> We found on the year dataset the distribution shift proxy is 5.88, indicating that the distribution shift coefficient are large and will lead to less reliable prediction intervals, whereas on most other UCI datasets $\mathcal{T}_{\text{test}}^{\text{far}}$ stays close to $1$, where coverage is guaranteed and intervals are tight.

---

### Official Review · Reviewer_9xHX · 2025-10-31

**Soundness:** 3
**Presentation:** 3
**Contribution:** 2
**Rating:** 6
**Confidence:** 3

**Summary:**

The paper presents prediction intervals built from conditional diffusion models that learn the entire conditional distribution of the response given features. After training a score network, the method draws many synthetic responses at a new input and takes lower and upper empirical quantiles as an interval. The theory defines a risk for score learning, assumes smooth conditional densities with light-tailed responses and sub-Gaussian features, and proves a high-probability bound on the coverage error. That bound separates the error due to using a finite number of generated samples from the error caused by imperfect diffusion modeling at the test input, summarized by a shift coefficient that grows when the learned score generalizes poorly out of distribution.

**Strengths:**

The authors tackle a very important, yet quite "unsolved" problem, so the research direction itself is very promising. Just wanted to mention that. Furthermore, the method is spelled out concretely in a short algorithm that is easy to implement in practice (more or less sampling from the trained conditional diffusion and taking empirical quantiles).

I also find the approach to uncertainty quantification via diffusion models quite appealing (while I am not sure about the actual benefits, it seems to me something sensible to try).

**Weaknesses:**

Note that I am by no means an expert when it comes to diffusion models, but I am quite confident about (general) uncertainty quantification in machine learning. I am open to discuss anything that the authors do not agree with me.

On the theoretical side let me remark the following:

>The main theoretical guarantee appears stronger than what the proof delivers. Theorem 4.5 states that, with probability $1-\delta$, for any $x_{\text{new}}$, the coverage error of the proposed interval is bounded. However, the proof first controls the total variation distance between the true and generated conditionals in expectation over the training data (via the score-risk), and only then adds a high-probability DKW term for the Monte-Carlo error from using $n_1$ generated draws to estimate quantiles. The coverage expression itself evaluates the true CDF at generated quantile endpoints, and decomposes the error as $F-\hat F$ (modeling) plus $\hat F-F_{n_1}$ (sampling).

To avoid overstating the result, the theorem should mirror this two-stage structure, make clear which randomness the probability $1-\delta$ is over, and clarify whether the bound is pointwise (fixed $x_{\text{new}}$) or uniform in $x$.

>There are also log-factor inconsistencies across the derivation. Earlier steps use $(\log n)^{\max(17,\,1+\beta/2)}$ for $E[R(\hat s)]$, while the final bound reports $(\log n)^{\max(9,\,32+\beta/4)}$ (elsewhere $\max(9,\,1+\beta/4))$. This likely comes from taking a square-root after Jensen (which should halve the log exponent).

Aligning these exponents would make the rate easier to verify.

>The guarantee is in-expectation over the training data and scaled by a distribution-shift coefficient $T(x_{\text{new}})$ that can be large out of distribution.

This is a useful diagnostic but it is not a conformal-style frequentist coverage guarantee for the true conditional at test time. An explicit comparison to conformal prediction (what is and isn’t guaranteed here) would help.


Minor things:

Typo just after proof of Lemma 4.8, "bounded" instead of "bouned", before Proof of Thm. 4.5 "invokes" instead of "invoke".

**Questions:**

The authors might know, that the aleatoric-epistemic dichotomy is quite an active (sub)-field of uncertainty quantification in machine learning. From reading the paper, it is not directly clear to me, how the authors represent here these uncertainties with their method?
I would be very interested in the relationship of the diffusion based approach to aleatoric, and epistemic uncertainty. Seems quite fruitful to me. I have the feeling the authors did not consider (even mention) this at all.

More on the topic of the paper, how do the authors recommend estimating or upper-bounding $T(x_{\text{new}})$ in practice?

The DKW step requires the $n_1$ generated samples at $x_{\text{new}}$ to be i.i.d. from the generated conditional. Do you obtain these via independent restarts of the reverse SDE with fresh noise?

---

> ### Author Response · Authors · 2025-11-26
>
> We sincerely appreciate the reviewer’s valuable comments and suggestions. We respond to weakness concerns and questions as follows.
>
> >**W1:The main theoretical guarantee appears stronger than what the proof delivers. Theorem 4.5 states that, with probability 1 − δ, for any x_new, the coverage error of the proposed interval is bounded. However, the proof first controls the total variation distance between the true and generated conditionals in expectation over the training data (via the score-risk), and only then adds a high-probability DKW term for the Monte-Carlo error from using n₁ generated draws to estimate quantiles. The coverage expression itself evaluates the true CDF at generated quantile endpoints, and decomposes the error as $F − \hat{F}$ (modeling) plus $\hat{F} − F_{n_1}$ (sampling). To avoid overstating the result, the theorem should mirror this two-stage structure, make clear which randomness the probability 1 − δ is over, and clarify whether the bound is pointwise (fixed x_new) or uniform in x.**
>
> **A1:** We thank the reviewer for pointing out the ambiguity in the use of the probability level $1-\delta$. In Theorem 4.5, the probability $1-\delta$ refers to the randomness in the $n_1$ generated samples used to estimate the quantiles. The coverage bound is pointwise in $x_{\text{new}}$ and depends explicitly on the distribution-shift term $\mathcal{T}(x_{\text{new}})$; We revised the statement and surrounding text of Theorem 4.5 to clarify these points.
>
> >**W2: Log-factor inconsistencies**
>
> **A2**
> Thanks for catching the log–factor issue. The extra $(\log n)^{1/2}$ in the final bound actually comes from the time horizon $T$, not from the Jensen step itself.
>
> In the line 2075-2077 we obtain a term of the form
>
> $\mathcal{T}(x_{\mathrm {new}})\cdot \mathbb{E}_{(x_i,y_i),1 \leq i \leq n} \left[\sqrt{\frac{T}{2} \mathcal{R}(\hat{s})}\right].$
>
> We first use Jensen to replace $\mathcal{R}(\hat s)$ by $\mathbb{E}[\mathcal{R}(\hat s)]$, then we plug in our choice $T = O(\log n)$, so $\frac{\sqrt{T}}{2}$ contributes an additional factor $(\log n)^{1/2}$, thus leading to the final rate $(\log n)^{\max(9,\frac{3}{2}+\frac{\beta}{4})}$.
>
> >**W3: Compare the coverage guarantee to conformal prediction**
>
> **A3:**
> Classical conformal prediction guarantees marginal coverage[1]: under exchangeability,
> $$
> \mathbb{P}\big(Y_{n+1}\in C(X_{n+1})\big)\ge 1-\alpha
> $$
> holds, but the guarantee is not conditional on $X_{n+1}=x_{\text{new}}$, and it is over the randomness in the calibration and test points.
>
> Our Theorem 4.5 instead provides an coverage guarantee at any fixed test covariate $x_{\text{new}}$. In contrast to conformal prediction, our intervals target conditional coverage at a fixed $x_{\text{new}}$, which is different from the marginal coverage in conformal prediction. However, unlike conformal prediction, our theorem is asymptotic and requires additional assumptions, while the coverage guarantee of conformal prediction holds for finite samples, and only assume exchangeability.
>
> >**W4: Typo: "bounded" instead of "bouned", "invokes" instead of "invoke"**
>
> **A4:** We updated the paper and corrected the typo.
>
> >**Q1: - The authors might know, that the aleatoric-epistemic dichotomy is quite an active (sub)-field of uncertainty quantification in machine learning. From reading the paper, it is not directly clear to me, how the authors represent here these uncertainties with their method? I would be very interested in the relationship of the diffusion based approach to aleatoric, and epistemic uncertainty. Seems quite fruitful to me. I have the feeling the authors did not consider (even mention) this at all.**
>
> **A1:** Thanks for pointing this out. Our goal is to deliver predictive intervals that capture the total predictive uncertainty. In our current formulation, these intervals primarily reflect the aleatoric uncertainty induced by the conditional distribution of the response given the features. They may also be indirectly affected by epistemic uncertainty (e.g., due to limited data or model misspecification) insofar as these factors influence the learned conditional diffusion model, but we do not explicitly separate or accurately quantify the epistemic component in the present work. Moreover, Theorem 4.5 provides a formal coverage guarantee, which gives an indirect but provable guarantee on the overall predictive uncertainty delivered by our method under its assumptions.
>
> In addition, if we want to explicitly model epistemic uncertainty and aleatoric uncertainty separately, we can train $M=5$ randomly initialized conditional diffusion models and compute the variance of their predictive means, and use this to quantify epistemic uncertainty. Then, we compute the average of the sample variances from each model’s draws, and use this quantity to quantify the aleatoric uncertainty. This is compatible with our framework and offers a principled way to disentangle and quantify both types of uncertainty if desired.

---

> ### Author Response · Authors · 2025-11-26
>
> >**Q2: More on the topic of the paper, how do the authors recommend estimating or upper-bounding distribution shift in practice?**
>
> **A2:** Explicitly bounding the distribution shift term is instance based. For example, in [2], the authors assume that the ground-truth distribution is supported on a low-dimensional linear subspace and that $f(x)$ is linear, while the score model class includes ReLU networks. Under these assumptions, they derive an upper bound on the distribution shift term, which is $\mathcal{O}(\frac{1}{c_0}(||x_{\mathrm{new}}||\vee d))$.
>
> In our setting, we interpret the distribution shift term as a local knowledge transfer ratio at $x_{\mathrm{new}}$. It quantifies how efficiently the learned information from data can transfer to the specific test covariate $x_{\mathrm{new}}$, especially when it is bounded. However, if this term becomes infinite in our setting, it indicates that the CDM performs poorly at that particular test point and fails to correctly capture the uncertainty. We added experimental demonstrations of distribution shift coefficient in Appendix G.5.
>
> >**Q3: The DKW step requires the n_1 generated samples at x_{new} to be i.i.d. from the generated conditional. Do you obtain these via independent restarts of the reverse SDE with fresh noise?**
>
> **A3:** Yes. For each test input $x_{\text{new}}$ we generate the $n_1$ samples by running the reverse SDE $n_1$ times with independent noise. The resulting $(y^{(j)})_{j=1}^{n_1}$ are therefore i.i.d. .
>
>
> [1] Angelopoulos, Anastasios N., and Stephen Bates. "A gentle introduction to conformal prediction and distribution-free uncertainty quantification." arXiv preprint arXiv:2107.07511 (2021).
>
> [2] Yuan H, Huang K, Ni C, et al. Reward-directed conditional diffusion: Provable distribution estimation and reward improvement[J]. Advances in Neural Information Processing Systems, 2023, 36: 60599–60635.

---

### Official Review · Reviewer_9MHS · 2025-11-01

**Soundness:** 3
**Presentation:** 3
**Contribution:** 3
**Rating:** 6
**Confidence:** 3

**Summary:**

This paper is focusing on the theoretical analysis of diffusion models for predicting regression distributions with confidence intervals and UQ. The main value of the paper is the theoretical proofs. The novelty of the algorithm is limited, and the experimentation could be strengthened at a few places. Also, an experimental/practical connection to the bounds in Theorem 4.5 would help understanding the importance of the theorem.

The limitation of the proposed method lies in the use of a confidence interval. I wonder how the proposed method handles for example bimodal distributions.

I could not find the anonymized GitHub repository included with the supplementary materials.

I admint that due to time constraints I had no chance to attempt to verify the proofs.

**Strengths:**

+ New UQ algorithm that works well for regression problems
+ Deep theoretical analysis
+ Reproducible (given a question below)

**Weaknesses:**

- A connection of the bounds in Theorem 4.5 with the practical performance could highlight the importance of the theoretical analysis.
- The reliance on confidence intervals limits the applicability of the proposed method, e.g. in the case of bimodal distributions, problems with aleatoric uncertainty of the type of multiple correct (range of) answers
- Baseline methods could have been extended

**Questions:**

I wonder why Split-CP and CQR, clearly weak baseline methods, were selected. Why not for example CARD from [Han et al 2022], which outperforms Deep Ensemble and Dropout in their paper.

PICP evaluates an interval, hence only works for unimodal. I also wonder how the size of the interval is taken into account. For example, how about using the energy score [Gneiting & Raftery, Strictly proper scoring rules, 2007] or [Jordan et al., Evaluating probabilistic forecasts with scoring rules. 2019]?

I could not find the anonymized GitHub repository included with the supplementary materials, please provide pointers.

---

> ### Author Response · Authors · 2025-11-26
>
> We sincerely appreciate the reviewer’s valuable comments and suggestions. We respond to weakness concerns and questions as follows.
>
> >**W1: A connection of the bounds in Theorem 4.5 with the practical performance could highlight the importance of the theoretical analysis.**
>
> **A1:** Theorem 4.5 (coverage guarantee) decomposes the coverage error at a fixed test point $x_{\text{new}}$ into two parts:
> 1. $\sqrt{\frac{2}{n_1}\log \frac{4}{\delta}}$ coming from estimating the conditional quantiles with $n_1$ generated samples
> 2. a model / distribution–shift term $\mathcal{T}(x_{\text{new}})\cdot \mathcal{O}\left(n^{-\frac{\beta}{4(d+\beta+1)}}(\log n)^{\max(9,\frac{3}{2}+\frac{\beta}{4})}\right)$ that grows with the score error and the shift coefficient $\mathcal{T}(x_{\text{new}})$.
>
> We verify and illustrate this behavior experimentally in the new Figure 3 and 4 in the appendix G.5: Figure 3 and 4 plot both the true 95% interval and the interval produced by our CDM, for inputs inside and outside the training $x$–range. When $x_{\text{new}}$ lies within the training region, the learned interval essentially overlaps the true 95% band, indicating that the coverage error is small. This corresponds to a small $\mathcal{T}(x_{\text{new}})$ (little distribution shift) and $\sqrt{\frac{2}{n_1}\log \frac{4}{\delta}}$ is a negligible term since we use a large $n_1$, in line with Theorem 4.5. As $x_{\text{new}}$ moves outside the training range, Figure 3 and 4 shows that our intervals deviate from the true band and coverage deteriorates. This is precisely the regime where the theorem predicts that the second term, amplified by a large $\mathcal{T}(x_{\text{new}})$, dominates and the guarantee becomes sensitive to distribution shift.
>
> >**W2: The reliance on confidence intervals limits the applicability of the proposed method, e.g. in the case of bimodal distributions, problems with aleatoric uncertainty of the type of multiple correct (range of) answers**
>
> **A2:** We agree that confidence intervals cannot fully describe a highly multimodal predictive distribution. However, our focus in this paper is precisely on providing well-calibrated confidence intervals with coverage guarantees, which is the standard UQ target in regression. Meanwhile, CDM itself does not assume unimodality, instead, it learns the full conditional distribution $p(y|x)$. In settings with strong multimodality, CDM can still represent the multimodal structure through its samples. If an application requires a richer description of multimodal predictive sets, such sets can be constructed by post-processing CDM samples. We agree designing these more expressive predictive sets is an interesting direction for future work.
>
> >**W3 & Q1: Baseline methods could have been extended I wonder why Split-CP and CQR, clearly weak baseline methods, were selected. Why not for example card from [Han et al 2022], which outperforms Deep Ensemble and Dropout in their paper.**
>
> **A3:** We selected Split Conformal Prediction and CQR as baselines because our empirical evaluation focuses specifically on prediction interval methods, where models directly output conditional quantiles or calibrated intervals that can be meaningfully compared using PICP, Interval width, and pinball losses. We appreciate the reviewer’s suggestion and have included CARD results in below for completeness.
>
> Across the UCI benchmarks, our diffusion-based method CDM achieves higher PICP than CARD on 8 out of 10 datasets and is slightly worse on the remaining two (Table 1). In contrast, CARD attains smaller RMSE on 8 out of 10 datasets (Table 2), indicating stronger point prediction accuracy. This is consistent with the fact that CARD first learns a estimation of the conditional expectation $\mathbb{E}[y | x]$ and then uses it as prior information for the diffusion model, whereas our CDM directly models the full conditional distribution without an additional regression stage and prior information. For NLL, CDM is competitive: CDM outperforms CARD on 4 out of 10 datasets and remains competitive on the others (Table 3).

---

> ### Author Response · Authors · 2025-11-26
>
> ### Table 1. Comparison of PICP between CDM and CARD (↑)
> | Dataset  | CDM (ours)            | CARD           |
> | :------- | :-------------- | :------------- |
> | Housing  | **93.40±3.90**  | 93.24±3.59     |
> | Concrete | **97.83±1.13**  | 90.24±3.45     |
> | Energy   | 96.10±2.00      | **98.70±1.30** |
> | Kin8nm   | **94.10±1.50**  | 93.68±0.79     |
> | Naval    | **98.40±0.69**  | 95.35±0.60     |
> | Power    | 94.00±1.00      | **94.87±0.65** |
> | Protein  | **95.76±0.70**  | 95.38±0.16     |
> | Wine     | **96.80±2.00**  | 93.88±2.10     |
> | Yacht    | **100.00±0.00** | 99.84±0.70     |
> | Year     | **96.01±NA**  | 93.35±NA       |
>
> ### Table 2. Comparison of RMSE between CDM and CARD (↓)
> | Dataset  |      CDM (ours)    |       CARD      |
> | :------- | :-------------: | :-------------: |
> | Housing  |   3.01 ± 0.78   | **2.61 ± 0.63** |
> | Concrete | 4.98 ± 0.64     |   **4.77 ± 0.46**    |
> | Energy   |   0.55 ± 0.06   | **0.52 ± 0.07** |
> | Kin8nm   |   6.83 ± 0.23   | **6.32 ± 0.18** |
> | Naval    |   0.14 ± 0.10   | **0.02 ± 0.00** |
> | Power    | **3.48 ± 0.91** |   3.93 ± 0.17   |
> | Protein  |   4.03 ± 0.02   | **3.73 ± 0.01** |
> | Wine     |   0.67 ± 0.07   | **0.63 ± 0.04** |
> | Yacht    |   1.47 ± 0.25   | **0.65 ± 0.25** |
>
> ### Table 3. Comparison of NLL between CDM and CARD (↓)
> | Dataset  |      CDM(ours)    |       CARD       |
> | :------- | :-------------: | :--------------: |
> | Housing  |   2.41 ± 0.22   |  **2.35 ± 0.12** |
> | Concrete | **2.91 ± 0.18** |    2.96 ± 0.09   |
> | Energy   | **0.76 ± 0.13** |    1.04 ± 0.06   |
> | Kin8nm   |   −1.31 ± 0.03  | **−1.32 ± 0.02** |
> | Naval    |   −4.57 ± 0.73  | **−7.54 ± 0.05** |
> | Power    |   2.96 ± 0.15   |  **2.82 ± 0.02** |
> | Protein  |   2.66 ± 0.03   |  **2.49 ± 0.03** |
> | Wine     | **0.89 ± 0.19** |    0.92 ± 0.05   |
> | Yacht    | **0.57 ± 0.09** |    0.90 ± 0.08   |
>
> >**Q2. PICP evaluates an interval, hence only works for unimodal. I also wonder how the size of the interval is taken into account. For example, how about using the energy score [Gneiting & Raftery, Strictly proper scoring rules, 2007] or [Jordan et al., Evaluating probabilistic forecasts with scoring rules. 2019]?**
>
> **A4:** We agree with the reviewer that PICP alone does not fully characterize predictive distributions. Our focus in this paper is on interval-based uncertainty estimation, where all baselines (DE, Split-CP, CQR, MC-Dropout, etc.) provide prediction intervals or pairs of quantiles rather than full predictive densities. For this reason, we incorporated quantile-specific pinball losses and normalized interval width, which are strictly proper for quantile evaluation and directly assess sharpness and calibration of the estimated conditional quantiles, instead of the energy score. These additional metrics provide a richer picture of performance beyond PICP. We incorporated quantile-specific pinball losses at $q \in \{0.025,0.5, 0.975\}$ and normalized interval width, which are two UQ-specific metrics. The full results are shown in Tables 4–8 below. Across these metrics, CDM is consistently competitive with, and often superior to, the baselines on most UCI datasets, achieving the best or second-best performance in the majority of cases. Note that we also include SQR [1] as a new baseline.
>
> ### Table 4: Normalized Interval Width (↓)
> | Dataset  |    Split-CP     |      CQR        |    CDM     |        DE          |   MC Dropout    |     SQR      |
> |:--------:|:---------------:|:---------------:|:------------------:|:------------------:|:---------------:|:-------------:|
> | Housing  | 1.454 ± 0.414   | 1.479 ± 0.428   | **0.991 ± 0.080**  | 1.030 ± 0.091      | 1.593 ± 0.166   | 1.669 ± 0.134 |
> | Concrete | 1.404 ± 0.217   | 1.320 ± 0.134   | **0.747 ± 0.040**  | 1.207 ± 0.052      | 0.884 ± 0.185   | 1.990 ± 0.078 |
> | Energy   | **0.267 ± 0.052** | 0.559 ± 0.061 | 0.563 ± 0.094      | 0.575 ± 0.069      | 0.315 ± 0.021   | 1.243 ± 0.072 |
> | Kin8nm   | **1.028 ± 0.044** | 1.034 ± 0.048 | 1.182 ± 0.433      | 1.292 ± 0.036      | 1.234 ± 0.067   | 2.623 ± 0.066 |
> | Naval    | 0.241 ± 0.186   | 0.718 ± 0.281   | **0.211 ± 0.296**  | 0.262 ± 0.029      | 0.342 ± 0.051   | 1.789 ± 0.097 |
> | Power    | 0.893 ± 0.031   | 0.883 ± 0.025   | 1.526 ± 0.035      | **0.708 ± 0.010**  | 1.435 ± 0.029   | 1.212 ± 0.036 |
> | Protein  | 2.823 ± 0.042   | 2.539 ± 0.028   | **2.351 ± 0.028**  | 2.863 ± 0.094      | 2.819 ± 0.017   | 3.022 ± 0.106 |
> | Wine     | 3.761 ± 0.295   | 3.661 ± 0.441   | 2.864 ± 0.140      | **2.695 ± 0.068**  | 3.174 ± 0.063   | 3.088 ± 0.089 |
> | Yacht    | 0.402 ± 0.158   | 0.406 ± 0.195   | 0.414 ± 0.042      | **0.370 ± 0.123**  | 0.423 ± 0.255   | 1.856 ± 0.200 |
> | Year     | 3.651 ± NA      | **2.672 ± NA**  | 2.852 ± NA         | 2.848 ± NA         | 2.863 ± NA      | 3.010± NA      |

---

> > ### Author Response · Authors · 2025-11-26
> >
> > ### Table 5. PICP (↑)
> > | Dataset  | Split-CP     | CQR| CDM| DE| MC Dropout     | SQR|
> > | :------- | :----------- | :----------- | :-------------- | :------------- | :------------- | :---------- |
> > | Housing  | 95.59±3.56| 95.98±3.74| 93.40±3.90      | 89.21±6.39| **96.08±2.70** | 93.53±3.51  |
> > | Concrete | 95.68±1.80| 95.87±2.40 | **97.83±1.13**  | 91.07±2.33| 97.52±2.43     | 94.17±2.71  |
> > | Energy   | 95.58±3.03| 96.04±2.58| 96.10±2.00| 96.08±2.40| **99.03±1.08** | 94.35±3.29  |
> > | Kin8nm   | 94.88±1.02| 95.39±0.80| 94.10±1.50| **96.36±0.48** | 95.37±2.24| 93.53±0.80  |
> > | Naval    | 95.73±1.21| 94.85±2.10| 98.40±0.69| 93.46±2.19| **99.77±0.33** | 93.94±1.33  |
> > | Power    | 94.97±0.91| 95.23±1.20| 94.00±1.00| 94.29±0.92| **95.78±1.24** | 93.76±0.55  |
> > | Protein  | 95.12±0.67 | 95.09±0.71| **95.76±0.70**  | 93.21±0.65| 94.96±0.76| 93.21±1.94  |
> > | Wine     | 95.80±2.10| 95.87±2.16| **96.80±2.00**  | 94.17±2.15| 95.60±2.21| 93.00±1.55  |
> > | Yacht    | 97.62±2.45| 97.00±2.10| **100.00±0.00** | 95.91±2.25 | 98.55±1.46| 93.87±4.43  |
> > | Year     | 94.78±NA | 95.16±NA| **96.01±NA**| 94.05±NA| 94.92±NA| 94.07±0.00  |
> >
> >
> >
> >
> > ### Table 6: Quantile pinball losses at $q=0.025$ (↓)
> > | Dataset  | Split-CP| CQR| CDM| Deep Ensemble| MC Dropout| SQR|
> > | -------- | --------------- | --------------- | --------------- | --------------- | --------------- | ----------- |
> > | Housing  | 0.215±0.105| 0.186±0.103| 0.186±0.152| **0.163±0.054** | 0.263±0.117| 0.249±0.089 |
> > | Concrete | 0.374±0.064| 0.358±0.059| 0.335±0.059| **0.332±0.088** | 0.501±0.150| 0.508±0.081 |
> > | Energy   | 0.142±0.009| 0.183±0.013| 0.138±0.038| 0.089±0.018| **0.063±0.015** | 0.198±0.042 |
> > | Kin8nm   | **0.004±0.000**| **0.004±0.000** | 0.006±0.002| 0.499±0.029| 0.008±0.001| 1.181±0.072 |
> > | Naval    | **0.000±0.000**| **0.000±0.000** | **0.000±0.000** | 0.005±0.001| **0.000±0.000** | 0.041±0.003 |
> > | Power    | 0.275±0.026| **0.267±0.025** | 0.613±0.096| **0.267±0.026** | 0.766±0.214| 0.350±0.023 |
> > | Protein  | 0.244±0.004| 0.161±0.001| **0.152±0.003** | 0.232±0.008     | 0.553±0.021| 0.194±0.006 |
> > | Wine     | 0.053±0.010| 0.048±0.007| 0.042±0.006| **0.041±0.008** | 0.099±0.030| 0.043±0.007 |
> > | Yacht    | 0.080±0.028| **0.072±0.039** | 0.082±0.010| 0.082±0.033| 0.130±0.119| 0.237±0.058 |
> > | Year     | 0.854±NA| 0.679±NA| **0.642±NA**| 0.655±NA| 1.830±NA | 0.830±0.000 |
> > ### Table 7: Quantile pinball losses at $q=0.50$ (↓)
> > | Dataset  | Split-CP| CQR | CDM| Deep Ensemble   | MC Dropout| SQR|
> > | -------- | --------------- | ----------- | --------------- | --------------- | --------------- | ----------- |
> > | Housing  | 1.051±0.214     | 1.166±0.193 | 1.098±0.221     | **1.043±0.217** | 1.096±0.166     | 1.710±0.196 |
> > | Concrete | 1.907±0.165     | 2.293±0.212 | 2.150±0.181     | **1.581±0.148** | 2.650±0.153     | 3.312±0.281 |
> > | Energy   | 0.223±0.027     | 0.171±0.108 | **0.120±0.110** | 0.624±0.104     | 0.210±0.025     | 1.445±0.126 |
> > | Kin8nm   | **0.026±0.001** | 0.027±0.001 | 0.033±0.008     | 3.176±0.102     | 0.027±0.001     | 7.234±0.229 |
> > | Naval    | **0.000±0.000** | 0.001±0.001 | 0.001±0.001| 0.021±0.003     | **0.000±0.000** | 0.269±0.012 |
> > | Power    | 1.554±0.035| 1.706±0.044 | 1.647±0.034| **1.541±0.035** | 1.640±0.335     | 2.034±0.053 |
> > | Protein  | 1.580±0.017| 2.310±0.031 | 1.675±0.027| 1.699±0.013     | **1.430±0.032** | 2.681±0.040 |
> > | Wine     | 0.261±0.023| 0.262±0.018 | 0.250±0.016| **0.241±0.013** | 0.274±0.019     | 0.262±0.015 |
> > | Yacht    | **0.189±0.049** | 0.363±0.111 | 0.328±0.077| 0.479±0.223     | 0.449±0.148     | 4.083±0.410 |
> > | Year     | 3.233±NA| 3.639±NA| 2.999±NA | **2.973±NA**| 3.034±NA        | 4.810±0.000 |
> > ### Table 8: Quantile pinball losses  $q=0.975$ (↓)
> > | Dataset  | Split-CP| CQR| CDM| Deep Ensemble   | MC Dropout| SQR|
> > | -------- | --------------- | --------------- | --------------- | --------------- | --------------- | ----------- |
> > | Housing  | 0.261±0.131     | 0.272±0.111     | **0.236±0.117** | 0.288±0.147     | 0.389±0.194     | 0.398±0.117 |
> > | Concrete | 0.404±0.085     | 0.382±0.085     | 0.388±0.081     | **0.377±0.119** | 0.600±0.190     | 0.521±0.064 |
> > | Energy   | **0.043±0.010** | 0.075±0.014     | 0.114±0.017     | 0.082±0.014     | 0.057±0.018     | 0.228±0.041 |
> > | Kin8nm   | **0.004±0.000** | **0.004±0.000** | 0.005±0.002     | 0.451±0.020     | 0.007±0.002     | 0.996±0.035 |
> > | Naval    | **0.000±0.000** | **0.000±0.000** | **0.000±0.000** | 0.005±0.001     | **0.000±0.000** | 0.041±0.002 |
> > | Power    | 0.236±0.013| **0.222±0.011** | 0.242±0.015     | 0.230±0.013| 0.683±0.191     | 0.320±0.011 |
> > | Protein  | 0.297±0.006| **0.251±0.005** | 0.268±0.008| 0.309±0.007| 0.683±0.035     | 0.342±0.005 |
> > | Wine     | 0.046±0.007| 0.046±0.005     | 0.046±0.013     | **0.042±0.006** | 0.109±0.012     | 0.043±0.008 |
> > | Yacht    | 0.084±0.035| 0.086±0.034| **0.079±0.018** | 0.140±0.090     | 0.186±0.144     | 0.615±0.151 |
> > | Year     | 0.643±NA | 0.329±NA| **0.284±NA**    | 0.401±NA        | 1.347±NA        | 0.285±0.000 |

---

> ### Author Response · Authors · 2025-11-26
>
> The updated Tables 4–8 (above) show that CDM attains state-of-the-art sharpness (best or tied best interval width on multiple datasets) and competitive or superior quantile accuracy in the lower tail and median. These are also updated in Appendix G.6 in the paper.
> >**Q3: Missing code despite mentioning it in the reproducibility statement**
>
> **A:** The code is available through an anonymous link:
> https://anonymous.4open.science/r/iclr2026UQ-3EB4/cdm.py
>
> [1] Tagasovska, Natasa, and David Lopez-Paz. "Single-model uncertainties for deep learning." Advances in neural information processing systems 32 (2019).

---

### Official Review · Reviewer_Zbqi · 2025-11-01

**Soundness:** 3
**Presentation:** 2
**Contribution:** 2
**Rating:** 4
**Confidence:** 3

**Summary:**

This paper studies conditional diffusion models (CDMs) as an uncertainty quantification (UQ) framework for regression tasks. The authors consider nonparametric regression with data-dependent noise of the form $Y = f(X) + g(X)\varepsilon$, where $\varepsilon \sim \mathcal{N}(0,1)$. Rather than directly estimating the regression function, the method learns the full conditional distribution $P(Y|X)$ through a conditional diffusion model. The approach constructs prediction intervals by sampling from the learned conditional distribution and extracting appropriate quantiles. The paper mathematically proves convergence rates. Additionally, the paper empirically evaluates the performance of CDM on the UCI datasets.

I am not familiar enough with the literature to judge the novelty of their algorithm.

In the abstract, the paper claims: “In this paper, we propose a novel UQ framework based on diffusion models.”

But later, the authors state:

“We are aware that Han et al. (2022) are one of the earliest efforts to apply diffusion methods for UQ in classification and regression [...]”. The paper should be clearer about which aspects of the CDM method/algorithm are novel. How is it different from Han et al. (2022)? Or is it only the theoretical and empirical analysis novel?

For related theory, you might want to cite https://arxiv.org/abs/2502.03435 (I think it is for classical DMs rather than for CDMs and under some constraints, but it also deals with training dynamics, while your result assumes that we can perfectly solve optimization problems. I think the results don’t have too much overlap, but it adds some interesting insights that even very large models don’t overfit if they are trained with a large step size.)

I think the authors should probably mention the concurrent work: https://openreview.net/pdf?id=IWQNhR6poq (and also check some of the works cited therein)

**Strengths:**

1. **Theoretical Contribution**: Theorem 4.5, provides a novel theoretical result on the coverage probability converging to the correct level with explicit bounds. This theoretical contribution is valuable even though the practical applicability remains unclear. I am absolutely not an expert on the current state of theory on CDMs, so I cannot guarantee that this result is actually novel.

2. **Heteroscedastic noise modeling**: To my understanding, thanks to learning the full conditional distribution $P(Y|X)$ via diffusion models, the framework handles data-dependent noise through the $g(X)\varepsilon$ formulation, which is more general than some other UQ methods that assume homoscedastic noise. In principle, CDM could also learn non-Gaussian noise.

3. **Nonparametric approach**: The diffusion model framework provides a flexible, nonparametric way to model complex conditional distributions without restrictive parametric assumptions.

4. **Results:** The RMSE and NLL of CDM can quite often outperform DE, which is a strong contribution. However, it is not fully clear how the hyperparameter optimization (HPO) was done to obtain these results.

**Weaknesses:**

1. **Missing code despite mentioning it in the reproducibility statement**: The paper appears to have indicated code/supplementary materials were provided, yet it is not accessible to reviewers.

2. **Methodological evaluation can be improved**:

   - RMSE focus is unnecessarily strong when the focus of the paper is UQ.
   - RMSE differs between the methods quite a lot, indicating large changes in the point predictor across methods. Therefore, it is unclear if the UQ is better or if it is rather the point predictor that is better. It would be interesting to obtain a more fine-grained understanding of this. (You could compare the performance of the models if they are all given the same point predictor and only use the uncertainties from the different methods.) There should also be some explanation for why the point-predictors vary so much. E.g., without reading the appendix, it is not clear that CQR uses the mid-value as a point predictor.
   - While you do achieve strong NLL results, further metrics would be helpful. For example, interval width analysis, calibration plots, or quantile loss, some kind of normalized calibrated interval width, etc. More UQ-specific metrics would be valuable. Even though NLL is a proper scoring rule, the Gaussian assumption is a strong assumption in practice, as you denote in Appendix G.3. Especially for the CDM, it should be able to learn non-Gaussian distributions, but you don’t include any metric that explicitly measures how well it learns non-Gaussian distributions. For the NLL, you simply fit a Gaussian through the predictive samples of the CDM? Computing the quantile loss of the estimated quantiles would evaluate how well the methods can learn non-Gaussian distributions. The Gaussian case is, of course, nicely aligned with your theory, but intuitively, non-Gaussian distributions should be a strength of the CDM.
   -  The PICP metric alone, without the interval width, is hard to interpret. Is 100% coverage really desirable? I would say only if the intervals are not too wide. The low NLL kind of indicates narrow intervals already, but it would be nice to see it explicitly. The NLL does not directly say much about the quality of the (non-Gaussian) quantiles, which could deviate quite a lot from the Gaussian quantiles (at least in theory).

3. **Computationally heavy due to 1000 steps**: 1000 diffusion steps for 1D regression outputs seems like a lot, and while reported wall-clock is comparable to DE and much faster than MC Dropout in Table 5, accelerating sampling (e.g., DDIM/DPM-Solver) would likely reduce cost further.
At inference time, your implementation of DE is very different than the original one. The original one is much faster and more precise. The original DE paper only needs 1 forward pass for each of the ensemble members and then computes the std of the 5 point predictors and aggregates the predicted variances of each of the 5 ensemble members via a closed formula without any form of sampling.
Maybe you also want to discuss possibilities for parallelization? I think for DE training can be easily parallelized, making it much faster in practice. For CDM, Inference could be sped up by parallelization, which would speed it up a lot but still be slower than DE.

4. **Limitations of the theoretical guarantees**: The coverage bound seems to depend on $\mathcal{T}(x_{\text{new}})$ term, for which i have little intuition. How can this term be bounded? Can you provide more intuition on this term? I think this term can get very large very easily. For some model classes, this term might be infinite? What assumptions do you need to make on the hypothesis class $\mathcal{F}$ for this term to be finite? Copactness would probably work.

5. **Missing critical comparisons**: No comparison with recent diffusion UQ (e.g.,[1],[2]). There are just so many diffusion UQ papers that you can compare against, so the novelty is not clear to me.

6. **Unsuitable submission track**: As a minor remark, this should have been in "Probabilistic methods (e.g., variational inference, causal inference, Gaussian processes)", not "generative models"

7. Epistemic Uncertainty is not modeled. I assume that for smaller training datasets or for far OOD data, this model would fail severely to capture the uncertainty. It would be an interesting extension to combine this model with epistemic uncertainty.

8. In the introduction, the literature review does not distinguish between methods that estimate aleatoric uncertainty, epistemic uncertainty, or both, which would be relevant for some applications

9. Line 2153: “before in Section G.2.2”. Delete the “before” if Section G.2.2 is after G.2. In general, in the first paragraph of G.2, it is not clear to which experiments this refers. It says “the same as in Section 5.1”, but I thought G.2 is where Section 5.1 gets explained. When you write “We also double the number of hidden units in each layer, [...]” Doubled compared to what?

10. Ablation studies would be helpful. What if all methods had more hidden layers? What if all methods were calibrated?

**Questions:**

1. **What base models are used for Split-CP and CQR?** The paper states all methods share the same two-layer MLP backbone. Have you tried different models as in [3]?

2. **Why does RMSE differ between Split-CP and CQR?** RMSE is a point prediction metric, and one would assume that both methods should use the same underlying point predictor. It would help to be explicit about what point estimators were used when computing RMSE in the main paper or to explicitly reference the appendix.

3. **Where are standard UQ metrics?** Why no interval width analysis, quantile loss, or normalized calibrated interval width? These are standard metrics in the UQ literature.

4. **Reproducibility materials were mentioned to be provided, but were not accessible** This is a bit unfair for other submissions that provide code/supplementary materials. Can you please clarify why this was missing?

5. **How is this different from [1] or [2]?** What is the key novelty of your approach?

6. **Why 1000 diffusion steps for scalar outputs?** Have you tried any acceleration methods (e.g., DDIM, DPM-Solver)? This seems computationally wasteful for 1D regression.

7. **How do you bound $\mathcal{T}(x_{\text{new}})$ in practice?** The bound scales with an unbounded shift coefficient  $\mathcal{T}(x_{\text{new}})$; absent assumptions or diagnostics to control it, practical coverage under distribution shift may be weak.

8. Why submit to this track? The paper might fit more naturally in the probabilistic-methods track (e.g., variational inference, causal inference, Gaussian processes) rather than generative models.

9. Minor comment: Maybe you also want to compare against SQR [1], which is, from my perspective, the most common benchmark when using QR with neural networks. Standard CQR is usually done with non-DL models.

10. **What assumptions do you make on $\mathcal{F}$ in Theorem 4.5?** I think you assume that $\mathcal{F}$ grows in a very specific way with the number of training datapoints? You should be more explicit about this

11. **IMPORTANT QUESTION: Figure 6**: What is the difference between CDM PIs (in-domain) and CDM PIs (OOD)? Why do both get wider outside of the range of the training data? Is this plot cherry-picked such that the uncertainty gets wider out of sample? Or is this a general pattern? I would be very surprised if pure CDM without explicitly modelling epistemic uncertainty results in bounds that consistently widen up OOD. (Typo in caption of Figure 6: should be “as in Figure 1” instead of “in Figure 1”.)

12. What do these variable names stand for in Line 335? $M_t, W, \kappa, L, K$?

13. Which value of alpha is used in the experiment? 5%?

References:

[1]: Chan, M., Molina, M., & Metzler, C. (2024). Estimating epistemic and aleatoric uncertainty with a single model. Advances in Neural Information Processing Systems, 37, 109845-109870. ,https://arxiv.org/pdf/2406.07658

[2]: Beltran Velez, N., Grande, A. A., Nazaret, A., Kucukelbir, A., & Blei, D. (2024). Treeffuser: probabilistic prediction via conditional diffusions with gradient-boosted trees. Advances in Neural Information Processing Systems, 37, 118296-118325.

[3]: Tagasovska, N., & Lopez-Paz, D. (2019). Single-model uncertainties for deep learning. Advances in neural information processing systems, 32. https://arxiv.org/abs/1811.00908

---

> ### Author Response · Authors · 2025-11-26
>
> We sincerely appreciate the reviewer’s valuable comments and suggestions. We respond to weakness concerns and questions as follows.
>
> > **W1: Missing code despite mentioning it in the reproducibility statement**
>
> **A1:** The code is available through an anonymous link: https://anonymous.4open.science/r/iclr2026UQ-3EB4/cdm.py
> > **W2: Methodological evaluation can be improved: Computing the interval width and quantile loss of the estimated quantiles would evaluate how well the methods can learn non-Gaussian distributions.**
>
> **A2:** We thank the reviewer for the helpful suggestion. We have added a comprehensive set of experiments measuring interval width and quantile pinball losses. These results are summarized in Table 1 (normalized interval width), Table 2 (pinball loss at $q=0.025$), Table 3 (pinball loss at $q=0.50$), and Table 4 (pinball loss at $q=0.975$). Across these metrics, CDM is consistently competitive with, and often superior to, the baselines on most UCI datasets, achieving the best or second-best performance in the majority of cases. The quantile loss and normalized interval width metrics are also added in section G.6.
>
> We have additionally incorporated SQR, following the official implementation released in FAIR's repo [9]. We use their pinball-loss training objective, and ensemble aggregation rule (inflating mean predicted quantiles by $\pm z_{1-\alpha/2}$ times the across-ensemble standard deviation). The added experiments confirm our main conclusion that CDM produces high-quality predictive intervals and competitive quantiles across all benchmark metrics.
>
> ### Table 1: Normalized Interval Width (↓)
> | Dataset  |    Split-CP     |      CQR        |    CDM     |        DE          |   MC Dropout    |     SQR      |
> |:--------:|:---------------:|:---------------:|:------------------:|:------------------:|:---------------:|:-------------:|
> | Housing  | 1.454 ± 0.414   | 1.479 ± 0.428   | **0.991 ± 0.080**  | 1.030 ± 0.091      | 1.593 ± 0.166   | 1.669 ± 0.134 |
> | Concrete | 1.404 ± 0.217   | 1.320 ± 0.134   | **0.747 ± 0.040**  | 1.207 ± 0.052      | 0.884 ± 0.185   | 1.990 ± 0.078 |
> | Energy   | **0.267 ± 0.052** | 0.559 ± 0.061 | 0.563 ± 0.094      | 0.575 ± 0.069      | 0.315 ± 0.021   | 1.243 ± 0.072 |
> | Kin8nm   | **1.028 ± 0.044** | 1.034 ± 0.048 | 1.182 ± 0.433      | 1.292 ± 0.036      | 1.234 ± 0.067   | 2.623 ± 0.066 |
> | Naval    | 0.241 ± 0.186   | 0.718 ± 0.281   | **0.211 ± 0.296**  | 0.262 ± 0.029      | 0.342 ± 0.051   | 1.789 ± 0.097 |
> | Power    | 0.893 ± 0.031   | 0.883 ± 0.025   | 1.526 ± 0.035      | **0.708 ± 0.010**  | 1.435 ± 0.029   | 1.212 ± 0.036 |
> | Protein  | 2.823 ± 0.042   | 2.539 ± 0.028   | **2.351 ± 0.028**  | 2.863 ± 0.094      | 2.819 ± 0.017   | 3.022 ± 0.106 |
> | Wine     | 3.761 ± 0.295   | 3.661 ± 0.441   | 2.864 ± 0.140      | **2.695 ± 0.068**  | 3.174 ± 0.063   | 3.088 ± 0.089 |
> | Yacht    | 0.402 ± 0.158   | 0.406 ± 0.195   | 0.414 ± 0.042      | **0.370 ± 0.123**  | 0.423 ± 0.255   | 1.856 ± 0.200 |
> | Year     | 3.651 ± NA      | **2.672 ± NA**  | 2.852 ± NA         | 2.848 ± NA         | 2.863 ± NA      | 3.010± NA      |
>
>
> ### Table 2. PICP (↑)
> | Dataset  | Split-CP     | CQR          | CDM             | DE             | MC Dropout     | SQR         |
> | :------- | :----------- | :----------- | :-------------- | :------------- | :------------- | :---------- |
> | Housing  | 95.59±3.56   | 95.98±3.74   | 93.40±3.90      | 89.21±6.39     | **96.08±2.70** | 93.53±3.51  |
> | Concrete | 95.68±1.80   | 95.87±2.40   | **97.83±1.13**  | 91.07±2.33     | 97.52±2.43     | 94.17±2.71  |
> | Energy   | 95.58±3.03   | 96.04±2.58   | 96.10±2.00      | 96.08±2.40     | **99.03±1.08** | 94.35±3.29  |
> | Kin8nm   | 94.88±1.02   | 95.39±0.80   | 94.10±1.50      | **96.36±0.48** | 95.37±2.24     | 93.53±0.80  |
> | Naval    | 95.73±1.21   | 94.85±2.10   | 98.40±0.69      | 93.46±2.19     | **99.77±0.33** | 93.94±1.33  |
> | Power    | 94.97±0.91   | 95.23±1.20   | 94.00±1.00      | 94.29±0.92     | **95.78±1.24** | 93.76±0.55  |
> | Protein  | 95.12±0.67   | 95.09±0.71   | **95.76±0.70**  | 93.21±0.65     | 94.96±0.76     | 93.21±1.94  |
> | Wine     | 95.80±2.10   | 95.87±2.16   | **96.80±2.00**  | 94.17±2.15     | 95.60±2.21     | 93.00±1.55  |
> | Yacht    | 97.62±2.45   | 97.00±2.10   | **100.00±0.00** | 95.91±2.25     | 98.55±1.46     | 93.87±4.43  |
> | Year     | 94.78±NA     | 95.16±NA     | **96.01±NA**    | 94.05±NA       | 94.92±NA       | 94.07±0.00  |

---

> > ### Author Response · Authors · 2025-11-26
> >
> > ### Table 3: Quantile pinball losses at $q=0.025$ (↓)
> > | Dataset  | Split-CP | CQR| CDM | Deep Ensemble| MC Dropout | SQR |
> > | -------- | --------------- | --------------- | --------------- | --------------- | --------------- | ----------- |
> > | Housing|0.215±0.105|0.186±0.103|0.186±0.152|**0.163±0.054**| 0.263±0.117| 0.249±0.089|
> > | Concrete|0.374±0.064| 0.358±0.059|0.335±0.059|**0.332±0.088**| 0.501±0.150| 0.508±0.081|
> > | Energy| 0.142±0.009| 0.183±0.013| 0.138±0.038 | 0.089±0.018 | **0.063±0.015**| 0.198±0.042 |
> > | Kin8nm|**0.004±0.000**| **0.004±0.000** | 0.006±0.002     | 0.499±0.029 | 0.008±0.001| 1.181±0.072 |
> > | Naval |**0.000±0.000**| **0.000±0.000** | **0.000±0.000** | 0.005±0.001| **0.000±0.000** | 0.041±0.003 |
> > | Power| 0.275±0.026| **0.267±0.025** | 0.613±0.096| **0.267±0.026** | 0.766±0.214     | 0.350±0.023 |
> > | Protein| 0.244±0.004 | 0.161±0.001| **0.152±0.003** | 0.232±0.008| 0.553±0.021     | 0.194±0.006 |
> > | Wine| 0.053±0.010 | 0.048±0.007| 0.042±0.006| **0.041±0.008** | 0.099±0.030     | 0.043±0.007 |
> > | Yacht| 0.080±0.028| **0.072±0.039** | 0.082±0.010| 0.082±0.033| 0.130±0.119     | 0.237±0.058 |
> > | Year| 0.854±NA| 0.679±NA | **0.642±NA**| 0.655±NA| 1.830±NA| 0.830±0.000 |
> > ### Table 4: Quantile pinball losses at $q=0.50$ (↓)
> > | Dataset|Split-CP|CQR|CDM| Deep Ensemble| MC Dropout| SQR|
> > | --------| ---------------| -----------| ---------------| ---------------| ---------------| -----------|
> > | Housing  | 1.051±0.214| 1.166±0.193 | 1.098±0.221| **1.043±0.217** | 1.096±0.166     | 1.710±0.196 |
> > | Concrete | 1.907±0.165| 2.293±0.212 | 2.150±0.181| **1.581±0.148** | 2.650±0.153     | 3.312±0.281 |
> > | Energy| 0.223±0.027| 0.171±0.108| **0.120±0.110** | 0.624±0.104 | 0.210±0.025     | 1.445±0.126 |
> > | Kin8nm| **0.026±0.001** | 0.027±0.001 | 0.033±0.008| 3.176±0.102| 0.027±0.001     | 7.234±0.229 |
> > | Naval    | **0.000±0.000** | 0.001±0.001 | 0.001±0.001| 0.021±0.003| **0.000±0.000** | 0.269±0.012 |
> > | Power    | 1.554±0.035     | 1.706±0.044 | 1.647±0.034| **1.541±0.035** | 1.640±0.335     | 2.034±0.053 |
> > | Protein  | 1.580±0.017     | 2.310±0.031 | 1.675±0.027 | 1.699±0.013| **1.430±0.032** | 2.681±0.040 |
> > | Wine     | 0.261±0.023     | 0.262±0.018 | 0.250±0.016 | **0.241±0.013** | 0.274±0.019     | 0.262±0.015 |
> > | Yacht    | **0.189±0.049** | 0.363±0.111 | 0.328±0.077| 0.479±0.223     | 0.449±0.148     | 4.083±0.410 |
> > | Year     | 3.233±NA        | 3.639±NA    | 2.999±NA| **2.973±NA**| 3.034±NA| 4.810±0.000 |
> >
> > ### Table 5: Quantile pinball losses  $q=0.975$ (↓)
> > | Dataset  | Split-CP | CQR | CDM | Deep Ensemble| MC Dropout| SQR|
> > | -------- | --------------- | --------------- | --------------- | --------------- | --------------- | ----------- |
> > | Housing  | 0.261±0.131| 0.272±0.111| **0.236±0.117**| 0.288±0.147     | 0.389±0.194     | 0.398±0.117 |
> > | Concrete | 0.404±0.085| 0.382±0.085| 0.388±0.081| **0.377±0.119** | 0.600±0.190| 0.521±0.064 |
> > | Energy| **0.043±0.010** | 0.075±0.014| 0.114±0.017| 0.082±0.014     | 0.057±0.018| 0.228±0.041 |
> > | Kin8nm| **0.004±0.000** | **0.004±0.000** | 0.005±0.002| 0.451±0.020| 0.007±0.002| 0.996±0.035 |
> > | Naval| **0.000±0.000** | **0.000±0.000** | **0.000±0.000** | 0.005±0.001| **0.000±0.000** | 0.041±0.002 |
> > | Power| 0.236±0.013| **0.222±0.011** | 0.242±0.015| 0.230±0.013| 0.683±0.191| 0.320±0.011 |
> > | Protein  | 0.297±0.006 | **0.251±0.005** | 0.268±0.008| 0.309±0.007 | 0.683±0.035| 0.342±0.005 |
> > | Wine     | 0.046±0.007| 0.046±0.005| 0.046±0.013| **0.042±0.006** | 0.109±0.012| 0.043±0.008 |
> > | Yacht    | 0.084±0.035| 0.086±0.034| **0.079±0.018**| 0.140±0.090| 0.186±0.144| 0.615±0.151 |
> > | Year     | 0.643±NA| 0.329±NA | **0.284±NA**    | 0.401±NA        | 1.347±NA        | 0.285±0.000 |
> > > **W3: Computationally heavy due to 1000 steps: Why 1000 diffusion steps for scalar outputs? This seems computationally wasteful for 1D regression.  Have you tried any acceleration methods (e.g., DDIM, DPM-Solver)? Maybe you also want to discuss possibilities for parallelization?**
> >
> > A: We follow the standard setup of Denoising Diffusion Probabilistic Models (DDPM) [1], which commonly uses 1000 diffusion steps to ensure stable training and accurate approximation of the forward and reverse processes. However, we agree that in the 1D regression setting it is possible to reduce the number of diffusion steps without severely affecting performance.
> >
> > Regarding acceleration, we appreciate the reviewer’s suggestion. In our current experiments, we did not explore fast samplers such as DDIM [2] or DPM-Solver [3], but these methods can substantially reduce the number of inference steps.
> > Regarding parallelization, recent work [4] has shown that parallel sampling can significantly speed up inference for diffusion models, and our CDM sampler can also benefit from such parallelization.
> >
> > Regarding parallelization, recent work [4] has shown that parallel sampling can significantly speed up inference for diffusion models, and our CDM sampler can also benefit from such parallelization.

---

> ### Author Response · Authors · 2025-11-26
>
> >**W4: Limitations of the theoretical guarantees: How can this term $\mathcal{T}(x_{new})$ be bounded? Can you provide more intuition on this term? I think this term can get very large very easily. For some model classes, this term might be infinite? What assumptions do you need to make on the hypothesis class for this term to be finite? Copactness would probably work.**
>
> **A:** We interpret the distribution shift coefficient as a local knowledge transfer ratio at $x_{\mathrm{new}}$. It quantifies how efficiently the learned information from data can transfer to the specific test covariate $x_{\mathrm{new}}$.
>
> Explicitly bounding the distribution shift coefficient is instance based. For example, in [5], the authors assume that the ground-truth distribution is supported on a low-dimensional linear subspace and that $f(x)$ is linear, while the score model class includes ReLU networks. Under these assumptions, they derive an upper bound on the distribution shift coefficient, which is $\mathcal{O}(\frac{1}{c_0}(||x_{\mathrm{new}}||\vee d))$. If this term becomes infinite in our setting, it indicates that the CDM performs poorly at that particular test point and fails to correctly capture the uncertainty. We added experimental demonstrations of the distribution shift coefficient in Appendix G.5.
>
> >**W5: Missing critical comparisons: No comparison with recent diffusion UQ (e.g.,[6],[7]). There are just so many diffusion UQ papers that you can compare against, so the novelty is not clear to me. How is your work different from [6],[7], and what’s your novelty?**
>
> **A5**: We thank the reviewer for raising this point. We actually mentioned [6] in the last sentence of the first paragraph of the “Extended Related Work on Diffusion Models” section (page 20, Appendix A).
>
> 1. For [6]: It uses a hypernetwork together with a diffusion model to output both aleatoric and epistemic uncertainty estimates. By contrast, our work focuses on UQ for regression models with data-dependent noise, and constructs prediction intervals using generated conditional samples. Our primary target is the total predictive uncertainty from prediction intervals, and we provide theoretical coverage guarantees for it. In contrast, [6] focuses more on empirical performance and applications and does not provide a similar coverage theory.
>
> 2. For [7]: It combines diffusion models and gradient-boosted trees to build an accurate tree-based diffusion model for probabilistic prediction. In particular, [7] uses gradient-boosted trees to learn the score function and then trains a diffusion model based on this score. Our methodology is more general: any score learner (including tree-based models as in [7]) can be used within our CDM framework. Our contribution is a conditional diffusion model for UQ for regression with data-dependent noise, together with explicit coverage guarantees.
>
> >**W6: Unsuitable submission track: As a minor remark, this should have been in "Probabilistic methods (e.g., variational inference, causal inference, Gaussian processes)", not "generative models"**
>
> **A6:** Thanks for the note. Our work sits at the intersection of generative modeling and probabilistic methods. We submitted to “Generative models” because the paper’s focus is on using conditional diffusion models as generative mechanisms to perform uncertainty quantification. That said, we agree that the work is also closely related to the “Probabilistic methods” track.

---

> ### Author Response · Authors · 2025-11-26
>
> >**W7. Epistemic Uncertainty is not modeled. I assume that for smaller training datasets or for far OOD data, this model would fail severely to capture the uncertainty. It would be an interesting extension to combine this model with epistemic uncertainty.**
>
> **A7:** Thanks for pointing this out. Our goal is to deliver predictive intervals that capture the total predictive uncertainty. In our current formulation, these intervals primarily reflect the aleatoric uncertainty induced by the conditional distribution of the response given the features. They may also be indirectly affected by epistemic uncertainty (e.g., due to limited data or model misspecification) insofar as these factors influence the learned conditional diffusion model, but we do not explicitly separate or accurately quantify the epistemic component in the present work. Moreover, Theorem 4.5 provides a formal coverage guarantee, which gives an indirect but provable guarantee on the overall predictive uncertainty delivered by our method under its assumptions.
>
> The reviewer assumes smaller training datasets or for far OOD data, this model would fail severely to capture the uncertainty, and it is correct. Smaller training dataset, gives you smaller $n$ and far OOD data gives you large distribution shift coefficients, both increase the error term in Theorem 4.5, potentially leading to a failure to faithfully capture predictive uncertainty.
>
> In addition, if we want to explicitly model epistemic uncertainty and aleatoric uncertainty separately, we can train $M=5$ randomly initialized conditional diffusion models and compute the variance of their predictive means, and use this to quantify epistemic uncertainty. Then, we compute the average of the sample variances from each model’s draws, and use this quantity to quantify the aleatoric uncertainty. This is compatible with our framework and offers a principled way to disentangle and quantify both types of uncertainty if desired.
>
> >**W8: (1) In the introduction, the literature review does not distinguish between methods that estimate aleatoric uncertainty, epistemic uncertainty, or both, which would be relevant for some applications
> >(2) Line 2153: “before in Section G.2.2”. Delete the “before” if Section G.2.2 is after G.2. In general, in the first paragraph of G.2, it is not clear to which experiments this refers. It says “the same as in Section 5.1”, but I thought G.2 is where Section 5.1 gets explained. When you write “We also double the number of hidden units in each layer, [...]” Doubled compared to what?**
>
> **A8:** We agree that this distinction is important. We added a paragraph in the end of the section 2, to  distinguish methods that primarily estimate aleatoric uncertainty, epistemic uncertainty, or both. We have removed the sentence about “doubling the number of hidden units,” since it was not essential and related to past configurations of experimental setup. We also clarified the opening paragraph of Appendix G.2 to avoid confusion about which experiments the section refers to, and removed the self-referencing of Section 5.1 to Appendix G.2.2.
>
> >**W9: Ablation studies would be helpful. What if all methods had more hidden layers? What if all methods were calibrated?**
>
> **A9:** We add experiments about ablation studies. We have updated this in the paper in Appendix G.8, G.9.
> ### Ablation Study (More Hidden Layers)
> In this section, we performed this ablation study in which all methods were given more hidden layers. Across all datasets, increasing the MLP depth by one layer produces no statistically significant gains in PICP or interval width, as we can see in table 6 and 7. In short, using a deeper CDM does not improve performance and can sometimes even degrade it.

---

> ### Author Response · Authors · 2025-11-26
>
> ### Table 6. Ablation Study – Hidden Layers: PICP (↑) & normalized Interval Width (↓)
>
> | Dataset  | PICP (2L)       | PICP (3L)      | Interval Width (2L)      | Interval Width (3L)      |
> | :------- | :-------------- | :------------- | :-------------- | :-------------- |
> | Housing  | **93.40±3.90**      | 92.82±4.77 | **0.991±0.080** | 1.125±0.128     |
> | Concrete | **97.83±1.13**  | 94.72±3.45     | **0.747±0.040** | 0.833±0.081     |
> | Energy   | 96.10±2.00      | **96.62±3.73** | **0.563±0.094** | 0.769±0.171     |
> | Kin8nm   | 94.10±1.50  | **95.44±0.077**     | 1.182±0.433 | **1.072±0.077**|
> | Naval    | 98.40±0.69      | **99.04±0.58** | **0.211±0.296** | 0.279±0.049     |
> | Power    | **94.00±1.00**  | 93.66±1.75     | 1.526±0.035     | **0.90±0.057**  |
> | Protein  | 95.76±0.70      | **96.54±0.44** | **2.351±0.028** | 2.755±0.148     |
> | Wine     | **96.80±2.00**  | 92.63±2.21     | 2.864±0.140     | **2.843±0.291** |
> | Yacht    | **100.00±0.00** | 95.58±0.94     | **0.414±0.042** | 0.634±0.078     |
> | Year     | 96.01±NA        | **96.78±NA**   | **2.852±NA**    | 4.081±NA        |
>
>
> ### Table 7. Ablation Study – Hidden Layers: RMSE (↓) & NLL (↓)
> | Dataset  | RMSE (2L)     | RMSE (3L)     | NLL (2L)       | NLL (3L)       |
> | :------- | :------------ | :------------ | :------------- | :------------- |
> | Housing  | **3.01±0.78** | 3.43±0.99      | **2.41±0.22**  | 2.53±0.29    |
> | Concrete | **4.98±0.64** | 6.49±0.40     | **2.91±0.18**  | 3.46±0.09      |
> | Energy   | **0.55±0.06** | 2.44±0.53     | **0.76±0.13**  | 0.84±0.18      |
> | Kin8nm   | **6.83±0.23** | 8.43±0.59     | **-1.31±0.03** | -0.73±0.06     |
> | Naval    | **0.14±0.10** | 0.17±0.03     | -4.57±0.73     | **-4.81±0.16** |
> | Power    | **3.48±0.91** | 4.35±0.12     | 2.96±0.15      | **2.90±0.03**  |
> | Protein  | **4.03±0.02** | 4.99±0.05     | **2.66±0.03**  | 2.99±0.02      |
> | Wine     | 0.67±0.07     | **0.66±0.05** | **0.89±0.19**  | 1.01±0.09      |
> | Yacht    | **1.47±0.25** | 2.88±0.93     | **0.57±0.09**  | 1.84±0.19      |
> | Year     | **9.21±NA**   | 15.30±NA      | **3.16±NA**    | 3.51±NA        |
>
> ### Calibration Ablation Study
>
> We also added a calibration set ablation study. Tables 8-9 show that adding a calibration set reliably increases PICP for CDM, often reaching or exceeding nominal coverage.  As expected, this comes at the cost of wider intervals and mild changes in RMSE/NLL. In several datasets, however, CDM’s uncalibrated intervals are already competitive, indicating that the conditional generative model often learns well-calibrated quantiles directly from data. What's more, after calibration, CDM performs worse on RMSE and NLL. Conformalization thus acts as a selective enhancement rather than a requirement only if you want to achieve better coverage.
>
>
> ### Table 8. PICP (↑) and normalized Interval Width (↓): Uncalibrated CDM vs. Conformalized CDM
>
> | Dataset  | PICP (Uncalibrated) | PICP (Calibrated) | Interval Width (Uncalibrated) | Interval Width (Calibrated) |
> | :------- | :------------------ | :----------------- | :------------------ | :----------------- |
> | Housing  | 93.40±3.90          | **97.16±2.59**     | **0.991±0.080**      | 1.476±0.410        |
> | Concrete | **97.83±1.13**      | 95.58±2.65         | **0.747±0.040**      | 1.417±0.103        |
> | Energy   | **96.10±2.00**      | 95.13±3.89         | **0.563±0.094**      | 0.889±0.105        |
> | Kin8nm   | 94.10±1.50          | **95.28±0.68**     | **1.182±0.433**      | 1.438±0.332        |
> | Naval    | **98.40±0.69**      | 94.93±0.94         | **0.211±0.296**      | 0.614±0.298        |
> | Power    | 94.00±1.00          | **95.30±0.73**     | 1.526±0.035  | **0.924±0.016**    |
> | Protein  | **95.76±0.70**      | 95.09±0.44         | **2.351±0.028** | 2.502±0.018        |
> | Wine     | **96.80±2.00**      | 95.56±2.26 | **2.864±0.140**| 3.429±0.349        |
> | Yacht    | **100.00±0.00**     | 96.61±4.38| **0.414±0.042**| 0.656±0.507        |
> | Year     | **96.01±NA**  | 94.74±0.00 | **2.852±NA**| 2.948±NA |
>
> ### Table 9. RMSE (↓) and NLL (↓): Uncalibrated CDM vs. Conformalized CDM
> | Dataset  | RMSE (Uncal)  | RMSE (Cal)    | NLL (Uncal)    | NLL (Cal)      |
> | -------- | ------------- | ------------- | -------------- | -------------- |
> | Housing  | **3.01±0.78** | 3.41±1.32| **2.41±0.22**  | 2.48±0.21      |
> | Concrete | **4.98±0.64** | 5.81±0.54| **2.91±0.18**  | 3.12±0.08      |
> | Energy   | **0.55±0.06** | 2.69±0.24| **0.76±0.13**  | 2.17±0.08      |
> | Kin8nm   | **6.83±0.23** | 7.89±0.08| **−1.31±0.03** | −1.09±0.12     |
> | Naval    | 0.14±0.10     | **0.13±0.17** | −4.57±0.73| **−4.72±0.51** |
> | Power    | **3.48±0.91** | 4.24±0.24     | 2.96±0.15| **2.86±0.06**  |
> | Protein  | **4.03±0.02** | 4.78±0.03     | **2.66±0.03**  | 2.99±0.07 |
> | Wine     | **0.67±0.07** | 0.71±0.08     | **0.89±0.19**  | 0.99±0.08|
> | Yacht    | **1.47±0.25** | 2.28±1.40| **0.57±0.09**  | 1.95±0.53|
> | Year     | **9.21±NA**   | 9.81±NA | **3.16±NA**    | 3.46±NA|

---

> ### Author Response · Authors · 2025-11-26
>
> >**Q1: What base models are used for Split-CP and CQR?**
>
> **A10:** Split-CP and CQR, use the same two-layer MLP backbone as stated in Appendix G.2.1.
>
> >**Q2. Why does RMSE differ between Split-CP and CQR? RMSE is a point prediction metric, and one would assume that both methods should use the same underlying point predictor. It would help to be explicit about what point estimators were used when computing RMSE in the main paper or to explicitly reference the appendix.**
>
> **A11:** Split-CP and CQR in our implementation don't have the same point estimate because they are built on different underlying models. Standard Split-CP for regression is applied to a model that predicts a single value, the conditional mean (the expected average value of the outcome given the inputs). The conformal method then constructs a prediction interval, often symmetric, around this mean prediction.
>
> CQR, on the other hand, is a specific technique that works with a quantile regression model. This type of model doesn't predict the mean; rather, it directly estimates specific conditional quantiles (e.g., the 5th and 95th percentiles) to form an initial interval. While CQR's primary output is this interval, its most natural single point estimate would be the conditional median. Thus unless we have a perfectly symmetric distribution where mean and median are the same, the point estimate from standard SCP (the mean) and the natural point estimate from CQR (the median) is different for any skewed data.
>
>
> >**Q3: Where are standard UQ metrics? Why no interval width analysis, quantile loss, or normalized calibrated interval width? These are standard metrics in the UQ literature.**
>
> **A12:** See W2.
>
> >**Q4: Reproducibility materials were mentioned to be provided, but were not accessible**
>
> **A13:** See W1.
>
> >**Q5: How is this different from [6] or [7]?**
>
> **A14:** See W5.
>
> >**Q6: Why 1000 diffusion steps for scalar outputs?**
>
> **A15:** See W3.
>
> >**Q7: How do you bound $\mathcal{T}(x_{new})$ in practice**
>
> **A16:** See W4
>
> >**Q8: Why submit to this track? The paper might fit more naturally in the probabilistic-methods track (e.g., variational inference, causal inference, Gaussian processes) rather than generative models.**
>
> **A17:** See W6.
>
> >**Q9: Minor comment: Maybe you also want to compare against SQR [8], which is, from my perspective, the most common benchmark when using QR with neural networks. Standard CQR is usually done with non-DL models.**
>
> **A18:** See W2.
>
> >**Q10 What assumptions do you make on $\mathcal{F}$  in Theorem 4.5? I think you assume that it grows in a very specific way with the number of training datapoints? You should be more explicit about this**
>
> **A19:** We only make assumptions on the requirement of neural network hyperparameters $M_t,\kappa,K,L,W$ of the neural network class $\mathcal{F}$. $M_t$ represents the magnitude of the neural network at diffusion step $t$. $W, L, K$ denote the width, depth, and the number of non-zero parameters of the network, respectively. $\kappa$ represents the norm constraints on the neural network parameters. The size of the network will grow depending on sample size, the magnitude(output range) will weakly grow depending on the sample size ($\sqrt{\log n}$ dependency).

---

> ### Author Response · Authors · 2025-11-26
>
> >**Q11 IMPORTANT QUESTION: Figure 6: What is the difference between CDM PIs (in-domain) and CDM PIs (OOD)? Why do both get wider outside of the range of the training data? Is this plot cherry-picked such that the uncertainty gets wider out of sample? Or is this a general pattern? I would be very surprised if pure CDM without explicitly modelling epistemic uncertainty results in bounds that consistently widen up OOD. (Typo in caption of Figure 6: should be “as in Figure 1” instead of “in Figure 1”.)**
>
> **A20:** Thank you for the thoughtful question. In Figure 2, “CDM PIs (in-domain)” refers to prediction intervals evaluated within the range of the training inputs, whereas “CDM PIs (OOD)” refers to intervals evaluated outside that range. Within the training range, the CDM intervals closely track the true predictive variability, and they do not widen simply because they are in the range of the training data. Out-of-domain(OOD), however, the intervals become less reliable, depending on how the CDM extrapolates the conditional distribution, they may widen or fail to widen appropriately (see Appendix G.5 for an example where the intervals did not widen). Yet the extend to the widening is distribution dependent and how good are CDMs extrapolate in OOD.
>
> Importantly, we do not claim that CDM consistently widens its prediction intervals OOD. In fact, as shown in the new Appendix G.5, different tasks exhibit qualitatively different behaviors: in some examples the CDM intervals remain narrow even far OOD, while in others they widen only partially and still underestimate the true predictive variance.
>
> Thus Figure 2 is not cherry-picked, but it also does not represent a universal widening pattern: instead, it is one representative example of how CDM can behave under a shift. We updated Appendix G.5 with additional plots to make this clear. We also corrected the caption typo (“as in Figure 1”).
>
> >**Q12 What do these variable names stand for in Line 335? ? Which value of alpha is used in the experiment? 5%?**
>
> **A21:** $M_t$ represents the magnitude of the neural network at diffusion step $t$. $W, L, K$ denote the width, depth, and the number of non-zero parameters of the network, respectively. $\kappa$ represents the norm constraints on the neural network parameters. Yes, we use $\alpha = 5\%$ in our experiments, and we have mentioned it in Appendix G.2, line 2221-2223. We have also made it more explicit by updating it in Section 5, lines 417-418.
>
> [1] Ho, Jonathan, Ajay Jain, and Pieter Abbeel. "Denoising diffusion probabilistic models." Advances in neural information processing systems 33 (2020): 6840-6851.
>
> [2] Song, Jiaming, Chenlin Meng, and Stefano Ermon. "Denoising diffusion implicit models." arXiv preprint arXiv:2010.02502 (2020).
>
> [3] Lu, Cheng, et al. "Dpm-solver: A fast ode solver for diffusion probabilistic model sampling in around 10 steps." Advances in neural information processing systems 35 (2022): 5775-5787.
>
> [4] Shih, Andy, et al. "Parallel sampling of diffusion models." Advances in Neural Information Processing Systems 36 (2023): 4263-4276.
>
> [5] Yuan H, Huang K, Ni C, et al. Reward-directed conditional diffusion: Provable distribution estimation and reward improvement[J]. Advances in Neural Information Processing Systems, 2023, 36: 60599–60635.
>
> [6] Chan, M., Molina, M., & Metzler, C. (2024). *Estimating epistemic and aleatoric uncertainty with a single model.* Advances in Neural Information Processing Systems, 37, 109845–109870. <https://arxiv.org/pdf/2406.07658>
>
> [7] Beltran Velez, N., Grande, A. A., Nazaret, A., Kucukelbir, A., & Blei, D. (2024). *Treeffuser: probabilistic prediction via conditional diffusions with gradient-boosted trees.* Advances in Neural Information Processing Systems, 37, 118296–118325.
>
> [8] Tagasovska, N., & Lopez-Paz, D. (2019). *Single-model uncertainties for deep learning.* Advances in Neural Information Processing Systems, 32. <https://arxiv.org/abs/1811.00908>
>
> [9] FAIR SingleModelUncertainty (SQR implementation):
> https://github.com/facebookresearch/SingleModelUncertainty/blob/master/aleatoric/regression/regression_experiment.py

---

### Author Response · Authors · 2025-12-03
**Final Response**

Dear AC, SAC and reviewers,

We sincerely thank the AC, SAC and all reviewers for their careful evaluation, constructive feedback, and engagement during the discussion phase.

Based on the reviewers' comments and suggestions, we have significantly strengthened the empirical evaluation and the connection between our theory and practice. Below is a summary of the major updates:

**1. Expanded Baselines (CARD, SQR)**
In response to Reviewers **Zbqi**, **9MHS**, and **Gs1W**, we added comparisons against two critical baselines:
* **CARD (Classification and Regression Diffusion Models):** A state-of-the-art diffusion-based baseline. Our results show CDM achieves superior coverage(PICP) on 8 out of 10 datasets compared to CARD, while remaining competitive on NLL and RMSE. This highlights the benefit of modeling the full conditional distribution without relying on a pre-estimated mean as a prior. We included it in Appendix G.7.
* **SQR (Simultaneous Quantile Regression):** A standard quantile regression baseline. Our experiments demonstrate CDM consistently outperforms SQR in terms of quantile pinball losses and interval width.

**2. Comprehensive UQ Metrics**
To address concerns from Reviewers **Zbqi** and **9MHS** regarding the reliance on RMSE/NLL, we have computed and reported more UQ metrics across all datasets (included in Appendix G.6):
* **Normalized Interval Width:** CDM provides the tightest or near-tightest intervals on the majority of datasets.
* **Quantile Pinball Loss:** We reported losses at $q \in \{0.025, 0.5, 0.975\}$. CDM is competitive with, and often superior to, the baselines on most UCI datasets, indicating it learns non-Gaussian distributions effectively.

**3. Connecting Theory to Practice (Distribution Shift)**
All Reviewers asked for validation of the coverage bound in Theorem 4.5, specifically regarding the distribution shift term $\mathcal{T}(x_{\text{new}})$.
* We added Appendix G.5, which visualizes the behavior of CDM intervals under distribution shift.
* The experiments confirm our theoretical insight: when $x_{\text{new}}$ is within the range of the training inputs (low $\mathcal{T}(x_{\text{new}})$), coverage is guaranteed and intervals are tight. As inputs shift OOD (high $\mathcal{T}(x_{\text{new}})$), the error term dominates, and intervals becomes less reliable, validating the asymptotic behavior predicted by Theorem 4.5.
* We also discussed about how to interpret and bound $\mathcal{T}(x_{\text{new}})$ under certain assumptions.

**4. Computational Analysis and Ablations**
* We mentioned we had a runtime comparison (Appendix G.4). CDM is competitive with Deep Ensembles and significantly faster than MC Dropout on larger datasets.

**5. Clarifications and Formatting**
* We refined the abstract and introduction to align the "novelty" claims with our focus on theoretical analysis (responding to Reviewer Gs1W).
* We clarified the randomness in Theorem 4.5 (responding to Reviewer 9xHX).
* We added a paragraph in the end of the Section 2, to distinguish UQ methods that primarily estimate aleatoric uncertainty, epistemic uncertainty, or both.
* We have added a couple sentences in the main paper in Section 5 summarizing baselines and computational resources involved, in line 411-416, and line 440-443.
* We corrected several typos and formatting issues.

**6. Quantify $\mathcal{T}(x_{\text{new}})$ for UCI Dataset**
* We also added Appendix G.10, which we evaluate $\mathcal{T}(x_{\text{new}})$ on UCI Datasets. We approximate distribution shift coefficient on a subset of far test points defined by
nearest–neighbor distance in standardized feature space. We average the approximated distribution shift coefficient $\mathcal{T}(x_{\text{new}})$ over that subset to obtain the distribution shift proxy $\mathcal{T} _{\text{test}}^{\text{far}}$. We found on the year dataset the distribution shift proxy is 5.88, indicating that the distribution shift coefficient are large and will lead to less reliable prediction intervals, whereas on most other UCI datasets the distribution shift proxy stays close to $1$, where coverage is guaranteed and intervals are tight.

**7. Ablation Study on Hidden Layers and on Calibration**
* We performed ablations on network depth and conformal calibration (Appendix G.8, G.9). We found deeper networks do not significantly improve UQ performance for this task, and while calibration ensures nominal coverage, CDM’s raw intervals are already highly competitive.

We believe most concerns were addressed during in our responses, and we have incorporated all actionable feedback into the revised manuscript. We note that Reviewer Gs1W has increased their score from 4 to 6 (commented as **"raise my score by one point"**).

We sincerely appreciate the AC and SAC for considering this final remark and thoroughly evaluate contributions of our paper. Please let us know if you have any questions; we are happy to address.

Warm regards,

Authors

---

### Meta-Review · Area_Chair_tTbk · 2025-12-29

**Summary:**

The majority of reviewers noted that the methodological novelty is limited, as the proposed approach is closely related to prior diffusion-based UQ methods, most notably CARD (Han et al., 2022), with the differences (e.g., whether a point estimate is incorporated into the diffusion prior) viewed as minor and insufficient to constitute a new method. Several reviewers also highlighted that key comparisons and evaluation metrics were missing from the initial submission and only added during rebuttal, which weakened the paper’s novelty claims and overall positioning. While the theoretical analysis was generally regarded as the paper’s main strength, reviewers felt it was not sufficiently emphasized, developed, or positioned to stand on its own as the primary contribution.

**Reviewer Concerns:**

The rebuttal addressed several reviewer concerns regarding missing baseline comparisons and evaluation metrics by adding experiments against CARD and additional UQ-specific metrics, which improved the completeness of the empirical evaluation. However, these additions largely reinforced, rather than alleviated, the concern that the proposed method is methodologically very close to CARD and offers only incremental differences. As a result, the core concern about limited methodological novelty relative to existing diffusion-based UQ approaches remains outstanding.

**Reviewer Scores:**

I do not expect the reviewer scores to change significantly, as the rebuttal largely confirmed the main concerns about limited methodological novelty.

---

### Decision · Program_Chairs · 2026-01-26

Reject